# Symmetries, Flat Minima and the Conserved Quantities of Gradient Flow

**Bo Zhao**[*][†]
University of California, San Diego
bozhao@ucsd.edu

**Iordan Ganev**[*]
Radboud University
iganev@cs.ru.nl

**Robin Walters**
Northeastern University
r.walters@northeastern.edu

**Rose Yu**
University of California, San Diego
roseyu@ucsd.edu

**Nima Dehmamy**
IBM Research
nima.dehmamy@ibm.com

## Abstract

Empirical studies of the loss landscape of deep networks have revealed that many local minima are connected through low-loss valleys. Yet, little is known about the theoretical origin of such valleys. We present a general framework for finding continuous symmetries in the parameter space, which carve out low-loss valleys. Our framework uses equivariances of the activation functions and can be applied to different layer architectures. To generalize this framework to nonlinear neural networks, we introduce a novel set of nonlinear, data-dependent symmetries. These symmetries can transform a trained model such that it performs similarly on new samples, which allows ensemble building that improves robustness under certain adversarial attacks. We then show that conserved quantities associated with linear symmetries can be used to define coordinates along low-loss valleys. The conserved quantities help reveal that using common initialization methods, gradient flow only explores a small part of the global minimum. By relating conserved quantities to convergence rate and sharpness of the minimum, we provide insights on how initialization impacts convergence and generalizability.

## 1 Introduction

Training deep neural networks (NNs) is a highly non-convex optimization problem. The loss landscape of a NN, which is shaped by the model architecture and the dataset, is generally very rugged, with the number of local minima growing rapidly with model size (Bray & Dean, 2007; Şimşek et al., 2021). Despite this complexity, recent work has revealed many interesting structures in the loss landscape. For example, NN loss landscapes often contain approximately flat directions along which the loss does not change significantly (Freeman & Bruna, 2017; Garipov et al., 2018). Flat minima have been used to build ensemble or mixture models by sampling different parameter configurations that yield similar loss values (Garipov et al., 2018; Benton et al., 2021). However, finding such flat directions is mostly done empirically, with few theoretical results.

One source of flat directions is parameter transformations that keep the loss invariant (i.e. symmetries). Specifically, moving in the parameter space from a minimum in the direction of a symmetry takes us to another minimum. Motivated by the fact that continuous symmetries of the loss result in flat directions in local minima, we derive a general class of such symmetries in this paper.

---

[*]Equal contribution.
[†]Work done during an internship at IBM.

Our key insight is to focus on *equivariances of the nonlinear activation functions*; most known continuous symmetries can be derived using this framework. Models related by exact equivalence cannot behave differently on different inputs. Hence, for ensembling or robustness tasks, we need to find *data-dependent symmetries*. Indeed, aside from the familiar "linear symmetries" of NN, the framework of equivariance allows us to introduce a novel class of symmetries which act *nonlinearly* on the parameters and are data-dependent. These nonlinear symmetries cover a much larger class of continuous symmetries than their linear counterparts, as they apply for almost any activation function. We provide preliminary experimental evidence that ensembles using these nonlinear symmetries are more robust to adversarial attacks.

Extended flat minima arise frequently in the loss landscape of NNs; we show that symmetry-induced flat minima can be parametrized using *conserved quantities*. Furthermore, we provide a method of deriving explicit conserved quantities (CQ) for different continuous symmetries of NN parameter spaces. CQ had previously been

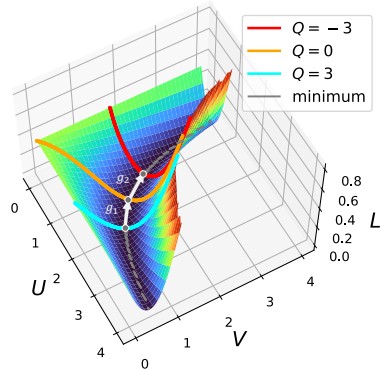

Figure 1: Visualization of the extended minimum in a 2-layer linear network with loss $\mathcal{L} = \|Y - UVX\|^2$. Points along the minima are related to each other by scaling symmetry $U \to Ug^{-1}$ and $V \to gV$. Conserved quantities, $Q$, associated with scaling symmetry parametrize points along the minimum.

derived from symmetries for one-parameter groups (Kunin et al., 2021; Tanaka & Kunin, 2021). Using a similar approach we derive the CQ for general continuous symmetries. This approach *fails* to find CQ for rotational symmetries. Nevertheless, we find the conservation law resulting from the symmetry implies a *cancellation of angular momenta* between layers. To summarize, our contributions are:

1. A general framework based on **equivariance** for finding symmetries in NN loss landscapes.

2. A derivation of the **dimensions of minima** induced by symmetries.

3. A new class of **nonlinear, data-dependent symmetries** of NN parameter spaces.

4. An expansion of prior work on **deriving conserved quantities** (CQ) associated with symmetries, and a discussion of its failure for rotation symmetries.

5. A **cancellation of angular momenta** result for between layers for rotation symmetries.

6. A **parameterization** of symmetry-induced flat minima via the associated CQ.

This paper is organized as follows. First, we review existing literature on flat minima, continuous symmetries of parameter space, and conserved quantities. In Section 3, we define continuous symmetries and flat minima, and show how linear symmetries lead to extended minima. We illustrate our constructions through examples of linear symmetries of NN parameter spaces. In Section 4, we define nonlinear, data-dependent symmetries. In Section 5, we use infinitesimal symmetries to derive conserved quantities for parameter space symmetries, extending the results in Kunin et al. (2021) to larger groups and more activation functions. Additionally, we show how CQ can be used to define coordinates along flat minima. We close with experiments involving nonlinear symmetries, conserved quantities and a discussion of potential use cases.

## 2   RELATED WORK

**Continuous symmetry in parameter space.** Overparametrization in neural networks leads to symmetries in the parameter space (Głuch & Urbanke, 2021). Continuous symmetry has been identified in fully-connected linear networks (Tarmoun et al., 2021), homogeneous neural networks (Badrinarayanan et al., 2015; Du et al., 2018), radial neural networks (Ganev et al., 2022), and softmax and batchnorm functions (Kunin et al., 2021). We provide a unified framework that generalizes previous findings, and identify nonlinear group actions that have not been studied before.

**Conserved quantities.** The imbalance between layers in linear or homogeneous networks is known to be invariant during gradient flow and related to convergence rate (Saxe et al., 2014; Du et al.,

2018; Arora et al., 2018a;b; Tarmoun et al., 2021; Min et al., 2021). Huh (2020) discovered similar conservation laws in natural gradient descents. Kunin et al. (2021) develop a more general approach for finding conserved quantities for certain one-parameter symmetry groups. Tanaka & Kunin (2021) relate continuous symmetries to dynamics of conserved quantities using an approach similar to Noether's theorem (Noether, 1918). We develop a procedure that determines conserved quantities from infinitesimal symmetries, which is closely related to Noether's theorem.

**Topology of minimum.** The global minimum of overparametrized neural networks are connected spaces instead of isolated points. We show that parameter space symmetries lead to extended flat minima. Previously, Cooper (2018) proved that the global minima is usually a manifold with dimension equal to the number of parameters subtracted by the number of data points. We derive the dimensionality of the symmetry-induced flat minima and show they are related to the number of infinitesimal symmetry generators and dimension of weight matrices. Şimşek et al. (2021) study permutation symmetry and show that in certain overparametrized networks, the minimum related by permutations are connected. Entezari et al. (2022) hypothesize that SGD solutions can be permuted to points on the same connected minima. Ainsworth et al. (2023) develop algorithms that find such permutations. Additional discussion on mode connectivity, sharpness of minima, and the role of symmetry in optimization can be found in Appendix A.

## 3 CONTINUOUS SYMMETRIES IN DEEP LEARNING

In this section, we first summarize our notation for basic neural network constructions (see Appendix C for more details). Then we consider transformations on the parameter space that leave the loss invariant and demonstrate how they lead to extended flat minima.

### 3.1 THE PARAMETER SPACE AND LOSS FUNCTION

The parameters of a neural network consist of weights[1] $W_i \in \mathbb{R}^{n_i \times m_i}$ for each layer $i$, where $n_i$ and $m_i$ are the layer output and input dimensions, respectively. For feedforward networks, successive output and input dimensions match: $m_i = n_{i-1}$. We group the widths into a tuple $\mathbf{n} = (n_L, \ldots, n_1, n_0)$, and the parameter space becomes $\mathbf{Param} = \mathbb{R}^{n_L \times n_{i-1}} \times \cdots \times \mathbb{R}^{n_1 \times n_0}$. We denote an element therein as a tuple of matrices $\boldsymbol{\theta} = (W_i \in \mathbb{R}^{n_i \times n_{i-1}})_{i=1}^{L}$. The activation of the $i$-th layer is a piecewise differentiable function $\sigma_i : \mathbb{R}^{n_i} \to \mathbb{R}^{n_i}$, which may or may not be pointwise. For $\boldsymbol{\theta} \in \mathbf{Param}$ and input $x \in \mathbb{R}^{n_0}$, the feature vector of the $i$th layer in feedforward network is $Z_{i+1}(x) = W_{i+1}\sigma(Z_i(x))$, where the juxtaposition '$W\sigma(Z)$' denotes an arbitrary linear operation depending on the context; for example, matrix product, convolution, etc. For simplicity, we largely focus on the case of multilayer perceptrons (MLPs). We denote the final output by $F_{\boldsymbol{\theta}} : \mathbb{R}^{n_0} \to \mathbb{R}^{n_L}$, defined as $F_{\boldsymbol{\theta}}(x) = \sigma_L(Z_L(x))$. The "loss function" $\mathcal{L}$ of our model is defined as:

$$\mathcal{L} : \mathbf{Param} \times \mathbf{Data} \to \mathbb{R}, \qquad \mathcal{L}(\boldsymbol{\theta}, (x, y)) = \mathrm{Cost}(y, F_{\boldsymbol{\theta}}(x)). \tag{1}$$

where $\mathbf{Data} = \mathbb{R}^{n_0} \times \mathbb{R}^{n_L}$ is the space of data and $\mathrm{Cost} : \mathbb{R}^{n_L} \times \mathbb{R}^{n_L} \to \mathbb{R}$ is a differentiable cost function, such as mean square error or cross-entropy. In the case of multiple samples, we have matrices $X \in \mathbb{R}^{n_0 \times k}$ and $Y \in \mathbb{R}^{n_L \times k}$ whose columns are the $k$ samples[2], and retain the same notation for the feedforward function, namely, $F_{\boldsymbol{\theta}} : \mathbb{R}^{n_0 \times k} \to \mathbb{R}^{n_L \times k}$. Most of our results concern properties of $\mathcal{L}$ that hold for any training data. Hence, unless specified otherwise, we take a fixed batch of data $\{(x_i, y_i)\}_{i=1}^{k} \subseteq \mathbf{Data}$, and consider the loss as a function of the parameters only.

**Example 3.1. Two-layer network with MSE** Consider a network with $\mathbf{n} = (n, h, m)$, the identity output activation ($\sigma_L(x) = (x)$), and no biases. The parameter space is $\mathbf{Param}(\mathbf{n}) = \mathbb{R}^{n \times h} \times \mathbb{R}^{h \times m}$ and we denote an element as $\boldsymbol{\theta} = (U, V)$. Taking the mean square error cost function, the loss function for data $(X, Y) \in \mathbb{R}^{n \times k} \times \mathbb{R}^{m \times k}$ takes the form $\mathcal{L}(\boldsymbol{\theta}, (X, Y)) = \frac{1}{k}\|Y - U\sigma(VX)\|^2$.

### 3.2 ACTION OF CONTINUOUS GROUPS AND FLAT MINIMA

Let $G$ be a group. An action of $G$ on the parameter space $\mathbf{Param}$ is a function $\cdot : G \times \mathbf{Param} \to \mathbf{Param}$, written as $g \cdot \boldsymbol{\theta}$, that satisfies the unit and multiplication axioms of the group, meaning $\mathrm{id} \cdot \boldsymbol{\theta} = \boldsymbol{\theta}$ where id is the identity of $G$, and $g_1 \cdot (g_2 \cdot \boldsymbol{\theta}) = (g_1 g_2) \cdot \boldsymbol{\theta}$ for all $g_1, g_2 \in G$.

---

[1] For clarity, we suppress the bias vectors; all results can be extended to include bias; see appendix C.

[2] We use capital letters for matrix data and small letters for individual samples.

**Definition 3.1** (Parameter space symmetry). *The action $G \times \textbf{Param} \to \textbf{Param}$ is a symmetry of $\mathcal{L}$ if it leaves the loss function invariant, that is:*

$$\mathcal{L}(g \cdot \boldsymbol{\theta}) = \mathcal{L}(\boldsymbol{\theta}), \qquad\qquad \forall \boldsymbol{\theta} \in \textbf{Param}, \quad g \in G. \qquad\qquad (2)$$

We describe examples of parameter space symmetries in the next section. Before doing so, we show how a parameter space symmetry leads to flat minima (see Appendix C.6):

**Proposition 3.2.** *Suppose $G \times \textbf{Param} \to \textbf{Param}$ is a symmetry of $\mathcal{L}$. If $\boldsymbol{\theta}^*$ is a critical point (resp. local minimum) of $\mathcal{L}$, then so is $g \cdot \boldsymbol{\theta}^*$ for any $g \in G$.*

The proof of this result relies on using the differential of the action of $g$ to relate the gradient of $\mathcal{L}$ at $\boldsymbol{\theta}^*$ with the gradient at $g \cdot \boldsymbol{\theta}^*$. We see that, if $\boldsymbol{\theta}^*$ is a local minimum, then so is every element of the set $\{g \cdot \boldsymbol{\theta}^* \mid g \in G\}$. This set is known as the *orbit* of $\boldsymbol{\theta}^*$ under the action of $G$. The orbits of different parameter values may be of different dimensions. However, in many cases, there is a "generic" or most common dimension, which is the orbit dimension of any randomly chosen $\boldsymbol{\theta}$.

### 3.3 Equivariance of the activation function

In this section, we describe a large class of linear symmetries of $\mathcal{L}$ using an *equivariance* property of the activations between layers. For accessibility, we focus on the example of two layers with output $F(x) = U\sigma(Vx)$ for $(U, V) \in \textbf{Param} = \mathbb{R}^{m \times h} \times \mathbb{R}^{h \times n}$ and $x \in \mathbb{R}^n$. All results generalize to multiple layers by letting $U = W_i$ and $V = W_{i-1}$ be weights of two successive layers in a deep neural network (see Appendix C.5). Let $G \subseteq \mathrm{GL}_h(\mathbb{R})$ be a subgroup of the general linear group, and let $\pi : G \to \mathrm{GL}_h(\mathbb{R})$ a representation (the simplest example is $\pi(g) = g$). We consider the following action of the group $G$ on the parameter space **Param**:

$$g \cdot U = U\pi(g^{-1}), \qquad g \cdot V = gV \qquad\qquad (3)$$

This action becomes a symmetry of $\mathcal{L}$ if and only if the following identity holds:

$$\sigma(gz) = \pi(g)\sigma(z) \qquad\qquad \forall g \in G, \quad \forall z \in \mathbb{R}^h \qquad\qquad (4)$$

We now turn our attention to examples. To ease notation, we write $\mathrm{GL}_h$ instead of $\mathrm{GL}_h(\mathbb{R})$.

**Example 3.2. Linear networks** A simple example of (4) is that of linear networks, where $\sigma$ is the identity function: $\sigma(x) = x$. One can take $\pi(g) = g$ and $G = \mathrm{GL}_h$.

**Example 3.3. Homogeneous activations** Suppose the activation $\sigma : \mathbb{R}^h \to \mathbb{R}^h$ is *homogeneous*, meaning that (1) $\sigma$ is applied pointwise in the standard basis and (2) there exists $\alpha > 0$ such that $\sigma(cz) = c^\alpha \sigma(z)$ for all $c \in \mathbb{R}_{>0}$ and $z \in \mathbb{R}^h$. Such an activation is equivariant under the *positive scaling group* $G \subset \mathrm{GL}_h$ consisting of diagonal matrices with positive diagonal entries. Explicitly, the group $G$ consists of diagonal matrices $g = \mathrm{diag}(\mathbf{c})$ with $\mathbf{c} = (c_1, \ldots, c_h) \in \mathbb{R}_{>0}^h$. For $z = (z_1, \ldots, z_h) \in \mathbb{R}^h$ and $g \in G$, we have $\sigma(gz) = \sum_j \sigma(c_j z_j) = \sum_j c_j^\alpha \sigma(z_j) = g^\alpha \sigma(z)$. Hence, the equivariance equation is satisfied with $\pi(g) = g^\alpha$.

**Example 3.4. LeakyReLU** This is a special case of homogeneous activation, defined as $\sigma(z) = \max(z, 0) + s \min(z, 0)$, with $s \in \mathbb{R}_{\geq 0}$. We have $\alpha = 1$, and $\pi(g) = g$.

**Example 3.5. Radial rescaling activations** A less trivial example of continuous symmetries is the case of a radial rescaling activation (Ganev et al., 2022) where for $z \in \mathbb{R}^h$, we have $\sigma(z) = f(\|z\|)z$ for some function $f : \mathbb{R} \to \mathbb{R}$. Radial rescaling activations are equivariant under rotations of the input: for any orthogonal transformation $g \in O(h)$ (that is, $g^T g = I$) we have $\sigma(gz) = g\sigma(z)$ for all $z \in \mathbb{R}^h$. Indeed, $\sigma(gz) = f(\|gz\|)(gz) = g(f(\|z\|)z) = g\sigma(z)$, where we use the fact that $\|gz\| = z^T g^T g z = z^T z = \|z\|$ for $g \in O(h)$. Hence, (4) is satisfied with $\pi(g) = g$.

We arrive at our first novel result, whose proof appears in Appendix C.6.

**Theorem 3.3.** *The dimension of a generic orbit in* **Param** *under the appropriate symmetry group is given as follows. The cases are divided based on whether $h \leq \max(n, m)$ or not.*

| Activation | Symmetry Group | Orbit Dimension | |
|:---:|:---:|:---:|:---:|
| | | $h \leq \max(n, m)$ | $h \geq \max(n, m)$ |
| Identity | $\mathrm{GL}_h(\mathbb{R})$ | $h^2$ | $h(n+m) - nm$ |
| Homogeneous | Positive rescaling | $h$ | $\max(n, m)$ |
| Radial rescaling | $O(h)$ | $\binom{h}{2}$ | $\binom{h}{2} - \binom{h-\max(m,n)}{2}$ |

As an aside, we note that a familiar example where (4) is satisfied involves the permutation of neurons. More precisely, suppose $\sigma$ is pointwise and let $G$ be the finite group of $h \times h$ permutation matrices. Then (4) holds with $\pi(g) = g$. However, the permutation group is finite (0-dimensional), and so does not imply the presence of flat minima.

### 3.4 INFINITESIMAL SYMMETRIES

Deriving conserved quantities from symmetries requires the infinitesimal versions of parameter space symmetries. Recall that any smooth action of a matrix Lie group $G \subseteq \mathrm{GL}_h$ induces an action of the infinitesimal generators of the group, i.e., elements of its Lie algebra. Concretely, let $\mathfrak{g} = \mathrm{Lie}(G) = T_I G$ be the Lie algebra, which can be identified with a certain subspace of matrices in $\mathfrak{gl}_h = \mathbb{R}^{h \times h}$. For every $M \in \mathfrak{g}$, we have an exponential map $\exp_M : \mathbb{R} \to G$ defined as $\exp_M(t) = \sum_{k=0}^{\infty} \frac{(tM)^k}{k!}$. If $\rho : G \to \mathrm{GL}_h$ is a (linear) representation, then the *infinitesimal action* is given by $d\rho : \mathfrak{g} \to \mathfrak{gl}_h$ by $d\rho(M) = \frac{d}{dt}\big|_0 \rho(\exp_M(t))$. In the case of the action appearing in (3), the corresponding infinitesimal action of the Lie algebra $\mathfrak{g}$ induced by (3) is given by:

$$M \cdot U = -U \, d\pi(M), \qquad M \cdot V = MV \tag{5}$$

More generally, suppose $G$ acts linearly on parameter space (see Appendix C for non-linear versions). Set $d$ to be the dimension of the parameter space[3], and make the identification $\mathbf{Param} \simeq \mathbb{R}^d$ by flattening matrices into column vectors. The general linear group $\mathrm{GL}(\mathbf{Param}) \simeq \mathrm{GL}_d(\mathbb{R})$ consists of all invertible linear transformations of $\mathbf{Param}$. Suppose $G$ is a subgroup of $\mathrm{GL}(\mathbf{Param})$, so its Lie algebra $\mathfrak{g}$ is a Lie subalgebra of $\mathfrak{gl}_d = \mathbb{R}^{d \times d}$. For $M \in \mathfrak{g}$ and $\boldsymbol{\theta} \in \mathbf{Param}$, the *infinitesimal action* is given simply by matrix multiplication: $M \cdot \boldsymbol{\theta}$.

In the case of a parameter space symmetry, the invariance of $\mathcal{L}$ translates into the following orthogonality condition, where the inner product $\langle , \rangle : \mathbf{Param} \times \mathbf{Param} \to \mathbb{R}$ is calculated by contracting all indices, e.g. $\langle A, B \rangle = \sum_{ijk...} A_{ijk...} B_{ijk...}$.

**Proposition 3.4.** *Let $G$ be a matrix Lie group and a symmetry of $\mathcal{L}$. Then the gradient vector field is point-wise orthogonal to the action of any $M \in \mathfrak{g}$:*

$$\langle \nabla_{\boldsymbol{\theta}} \mathcal{L} , M \cdot \boldsymbol{\theta} \rangle = 0, \qquad \forall \boldsymbol{\theta} \in \mathbf{Param} \tag{6}$$

## 4 NONLINEAR DATA-DEPENDENT SYMMETRIES

For common activation functions, the equivariance $\sigma(gz) = \pi(g)\sigma(z)$ of (4) holds only for $g$ belonging to a relatively small subgroup of $\mathrm{GL}_h$. For ReLU, $g$ must be in the positive scaling group, while for the usual sigmoid activation, the equation only holds for trivial $g = \mathrm{id}$. However, under certain conditions, it is possible to define a nonlinear action of the *full* $\mathrm{GL}_h$ which applies to many different activations. The subtlety of such an action is that it is data-dependent, which means that, for any $g \in \mathrm{GL}_h$, the transformation of the parameter space depends on the input data[4] $x$.

**The nonlinear action.** For any nonzero vector $z \in \mathbb{R}^h$, let $(r, \alpha_1, \ldots, \alpha_{h-1})$ be the spherical coordinates[5] of $z$, and define the following $h$ by $h$ matrix:

$$(R_z)_{ij} = \begin{cases} z_i \cos(\alpha_{j-1}) \left( \prod_{k=1}^{j-1} \sin(\alpha_k) \right)^{-1} & \text{if } j \leq i \text{ and } \prod_{k=1}^{i-1} \sin(\alpha_k) \neq 0 \\ -r \sin(\alpha_i) & \text{if } j = i + 1 \\ 0 & \text{otherwise} \end{cases}$$

where $\alpha_0 = 0$ by convention. We observe that $R_z$ is the product of a rotation matrix and rescaling by $|z|$. Moreover, since $z \neq 0$, the first column of $R_z$ is the unit vector $z/|z|$ and $R_z$ has inverse given by $R_z^{-1} = \frac{1}{|z|^2} R_z^T$. Using these facts, one arrives at the following result, stated in the case of a two-layer neural network with notation from Section 3.3, and proven in Appendix D:

---

[3]In terms of the widths, we have $d = \sum_{i=1}^{L} n_i n_{i-1}$.

[4]That is, rather than being a map $\mathrm{GL}_h \times \mathbf{Param} \to \mathbf{Param}$ satisfying the group action axioms, a data-dependent action is a map $\mathrm{GL}_h \times (\mathbf{Param} \times \mathbb{R}^n) \to \mathbf{Param} \times \mathbb{R}^n$ satisfying the same axioms.

[5]Hence, $r = |z|$ is the norm, and the $i$-th coordinate of $z$ is $z_i = r \cos(\alpha_i) \prod_{k=1}^{i-1} \sin(\alpha_k)$, where $\alpha_h = 0$.

**Theorem 4.1.** *Suppose $\sigma(z)$ is nonzero for any $z \in \mathbb{R}^h$. Then there is an action $\mathrm{GL}_h \times (\mathbf{Param} \times \mathbb{R}^n) \to \mathbf{Param} \times \mathbb{R}^n$ given by*

$$g \cdot (U, V, x) = (U R_{\sigma(Vx)} R_{\sigma(gVx)}^{-1} \,,\, gV \,,\, x). \tag{7}$$

*The evaluation of the feedforward function at $x$ unchanged: $F_{(U,V)}(x) = F_{(U R_{\sigma(Vx)} R_{\sigma(gVx)}^{-1}, gV)}(x)$.*

We emphasize that a necessary and sufficient condition for the particular action of Theorem 4.1 to be well-defined is that $\sigma(z)$ be nonzero for any $z \in \mathbb{R}^h$; this is the case for usual sigmoid. Moreover, in Appendix D.2, we provide a generalization to the case where $\sigma(z)$ is only required to be nonzero for any *nonzero* $z \in \mathbb{R}^h$, a condition satisfied by hyperbolic tangent, leaky ReLU, and many other activations. The cost of such a generalization is a restriction to a 'non-degenerate locus' of $\mathbf{Param} \times \mathbb{R}^n$ where $Vx \neq 0$. Theorem 4.1 also generalizes to mutli-layer networks, as explained in Appendix D.3. We have the following explicit algorithm to compute the action of Theorem 4.1:

0. Input: weight matrices $(U, V)$, input vector $x \in \mathbb{R}^n$, matrix $g \in \mathrm{GL}_h$.
1. Determine the spherical coordinates of $\sigma(Vx)$ and $\sigma(gVx)$, and construct the matrices $R_{\sigma(Vx)}$ and $R_{\sigma(gVx)}$.
2. Compute the inverse $R_{\sigma(gVx)}^{-1} = \frac{1}{|\sigma(gVx)|^2} R_{\sigma(gVx)}^T$.
3. Set $U' = U R_{\sigma(Vx)} R_{\sigma(gVx)}^{-1}$ and $V' = gV$.
4. Output: the transformed weights $(U', V')$. The data $x \in \mathbb{R}^n$ remains unchanged.

**Lipschitz bounds.** Unlike the exact symmetries of Section 3, a data-dependent action may alter the loss in the *function space*. This is evident from (7): while the transformed and original feedforward functions have the same value at $x$, they will differ at other points. That is, if $\tilde{x} \in \mathbb{R}^h$ is an input value different from $x$, then $F_{(U,V)}(\tilde{x}) \neq F_{(U R_{\sigma(Vx)} R_{\sigma(gVx)}^{-1}, gV)}(\tilde{x})$ in general.

However, the transformed feedforward function will differ from the original one in a controlled way. More precisely, when $\sigma$ is Lipschitz continuous, we show that there is a bound on how much the Lipschitz bound of the feedforward changes after the nonlinear action. The relevance of such a bound originates in the fact that we expect the distance between data points to encode important information about shared features. To be more specific, fix weight matrices $(U, V)$, which provide the feedforward function $F(\tilde{x}) = U \sigma(V \tilde{x})$. For any input vector $x \in \mathbb{R}^n$ and matrix $g \in \mathrm{GL}_h$, the transformed weight matrices $(U R_{\sigma(Vx)} R_{\sigma(gVx)}^{-1}, gV)$ provide a new feedforward function given by:

$$F_{(U,V)}^{(g,x)} : \mathbb{R}^n \to \mathbb{R}^m \qquad F_{(U,V)}^{(g,x)}(\tilde{x}) = U R_{\sigma(Vx)} R_{\sigma(gVx)}^{-1} \sigma(gV\tilde{x}) \tag{8}$$

**Proposition 4.2** (Lipschitz bounds from equivariance)**.** *Let $\sigma$ be Lipschitz continuous with Lipschitz constant $\eta$. Then $F_{(U,V)}^{(g,x)}$ is Lipschitz continuous with bound $\eta \|U\| \|V\| \frac{|\sigma(Vx)| \|g\|}{|\sigma(gVx)|}$.*

In particular, the Lipschitz bound of the original feedforward function is $\eta \|U\| \|V\|$. Thus, if it happens that $|\sigma(Vx)| \|g\| < |\sigma(gVx)|$, then the Lipschitz bound decreases when transforming the parameters. Additionally, we observe that the nonlinear action does not disrupt latent distribution of data significantly. See Appendix D.5 for proof of Proposition 4.2, which relies on iterative applications of the Cauchy-Schwarz inequality, as well as the fact that $\|R_z^{\pm 1}\| = |z|^{\pm 1}$.

**General equivariance.** The action described is an instance of a more general framework of equivariance. Specifically, a map $c : \mathrm{GL}_h \times \mathbb{R}^h \to \mathrm{GL}_h$ is said to be an *equivariance* if it satisfies (1) $c(\mathrm{id}_h, z) = \mathrm{id}_h$ for all $z$, and (2) $c(g_1, g_2 z) c(g_2, z) = c(g_1 g_2, z)$ for all $g_1, g_2 \in \mathrm{GL}_h$ and $z$. These two conditions on $c$ translate directly into the unit and multiplication axioms of a group[6], generalizing $\pi(g_1 g_2) = \pi(g_1)\pi(g_2)$, and $\pi(\mathrm{id}_h) = \mathrm{id}_h$. Every equivariance gives rise to a nonlinear action of $\mathrm{GL}_h$ on $\mathbf{Param} \times \mathbb{R}^h$ given by $g \cdot (U, V, x) = (U c(g, Vx)^{-1}, gV, x)$. This action is a symmetry preserving the loss if and only if the following generalization of (4) holds:

$$\text{General Equivariance:} \qquad \sigma(gz) = c(g, z)\sigma(z) \qquad \forall g \in \mathrm{GL}_h \; \forall z \in \mathbb{R}^h \tag{9}$$

An explicit example of such an equivariance is $c(g, z) = R_{\sigma(gz)} R_{\sigma(z)}^{-1}$, and Proposition 4.2 generalizes to any general equivariance by replacing $\frac{|\sigma(Vx)| \|g\|}{|\sigma(gVx)|}$ with $\|c(g, Vx)^{-1}\|$.

---

[6] In fact, $c$ defines a $\mathrm{GL}_h$-equivariant structure on the tangent bundle of $\mathbb{R}^h$.

## 5 CONSERVED QUANTITIES OF GRADIENT FLOW

We have shown that continuous symmetries lead us along extended flat minima in the loss landscape. In this section, we identify quantities that (partially) parameterize these minima. We first show that certain real-valued functions on the parameter space remain constant during gradient flow. We refer to such functions as *conserved quantities*. Applying symmetries changes the value of the conserved quantity. Therefore, conserved quantities can be used to parameterize flat minima.

**Gradient flow (GF).** Recall that GD proceeds in discrete steps with the update rule $\boldsymbol{\theta}_{t+1} = \boldsymbol{\theta}_t - \varepsilon \nabla \mathcal{L}(\boldsymbol{\theta}_t)$ where $\varepsilon$ is the learning rate (which in general can be a symmetric matrix), and $t = 0, 1, 2, \ldots$ are the time steps. In gradient flow, we can define a smooth curve in the parameter space from a choice of initial values to the limiting local minimum without discretizing over time. The curve is a function of a continuous time variable $t \in \mathbb{R}$, and velocity of this curve at any point is equal to the gradient of the loss function, scaled by the negative of the learning rate. In other words, the dynamics of the parameters under GF are given by:

$$\dot{\boldsymbol{\theta}}(t) = d\boldsymbol{\theta}(t)/dt = -\varepsilon \nabla_{\boldsymbol{\theta}(t)} \mathcal{L}. \tag{10}$$

From an initialization $\boldsymbol{\theta}(0)$ at $t = 0$, GF defines a trajectory $\boldsymbol{\theta}(t) \in \mathbf{Param}$ for $t \in \mathbb{R}_{>0}$, which limits to a critical point. In this way, GF is a continuous version of GD.

**Conserved quantities.** A *conserved quantity* of GF is a function $Q : \mathbf{Param} \to \mathbb{R}$ such that the value of $Q$ at any two time points $s, t \in \mathbb{R}_{>0}$ along a GF trajectory is the same: $Q(\boldsymbol{\theta}(s)) = Q(\boldsymbol{\theta}(t))$. In other words, we have $dQ(\boldsymbol{\theta}(t))/dt = 0$. Note that, if $f : \mathbb{R} \to \mathbb{R}$ is any function, and $Q$ is a conserved quantity, then the composition $f \circ Q$ is also a conserved quantity. Several conserved quantities of GF have appeared in the literature, most notably layer imbalance $Q_{\text{imb}} \equiv \|W_i\|^2 - \|W_{i-1}\|^2$ (Du et al., 2018) for each pair of successive feedforward linear layers ($\sigma(x) = x$), and its full matrix version $Q_i = W_i^T W_i - W_{i-1} W_{i-1}^T$.

We now propose a generalization of the layer imbalance by associating a conserved quantity to any infinitesimal symmetry. As in Section 3.2, suppose a matrix Lie group $G$ acts linearly on the parameter space. Then, from (6), we have the identity $\langle \nabla_\theta \mathcal{L}, M \cdot \boldsymbol{\theta} \rangle = 0$ for any element $M$ in the Lie algebra $\mathfrak{g}$. Using the gradient flow dynamics (10), this identity becomes:

$$\left\langle \varepsilon^{-1} \dot{\boldsymbol{\theta}} , M \cdot \boldsymbol{\theta} \right\rangle = 0 \tag{11}$$

In other words, the velocity at any point of a gradient flow curve is orthogonal to the infinitesimal action. For simplicity, we set the learning rate to the identity: $\varepsilon = I$ (all results generalize to symmetric $\varepsilon$.) The following proposition (whose proof is elementary and well-known) provides a way of 'integrating' equation (11), in the appropriate sense, in order to obtain conserved quantities:

**Proposition 5.1.** *Suppose the action of $G$ on $\mathbf{Param}$ is linear[7] and leaves $\mathcal{L}$ invariant. For any $M \in \mathfrak{g}$, there is a conserved quantity $Q_M : \mathbf{Param} \to \mathbb{R}$ given by $Q_M(\boldsymbol{\theta}) = \langle \boldsymbol{\theta} , M \cdot \boldsymbol{\theta} \rangle$.*

While Proposition 5.1 directly links the infinitesimal action to conserved quantities, it has the limitation that the conserved quantity corresponding to an anti-symmetric matrix $M = -M^T$ in $\mathfrak{g}$ is constantly zero, and we do not obtain meaningful conserved quantities. Instead, we can only conclude that flow curves satisfy the differential equation (11). Fixing a basis $(\theta^1, \cdots \theta^d)$ for $\mathbf{Param} \simeq \mathbb{R}^d$, this equation becomes $\sum_{i<j} M_{ij} r_{ij}^2 \dot{\phi}_{ij} \equiv 0$ where $(r_{ij}, \phi_{ij})$ are the polar coordinates for the point $(\boldsymbol{\theta}_i, \boldsymbol{\theta}_j) \in \mathbb{R}^2$ (see Appendix C.9.5). In summary, we find:

| $M \in \mathfrak{g}$ | symmetric $M$ | anti-symmetric $M$ |
|---|---|---|
| differential equation | conserved quantity | differential equation |
| $\dot{\theta}^T M \theta = 0$ | $Q_M(\theta) = \theta^T M \theta$ | $\sum_{i<j} m_{ij} r_{ij}^2 \dot{\phi}_{ij} \equiv 0$ |

**Conserved quantities parametrize symmetry flat directions.** We observe that applying a symmetry changes the values of the conserved quantities $Q_M$ (Figure 1). Indeed, for $M \in \mathfrak{g}$ and $g \in G$,

---

[7]For simplicity, we also assume that $G$ is closed under taking transposes, and acts faithfully on the parameter space. These assumptions generally hold in practice; see Appendix C for a version with fewer assumptions.

we have $Q_M(g \cdot \boldsymbol{\theta}) = Q_{g^T M g}(\boldsymbol{\theta})$ for all $\boldsymbol{\theta} \in \mathbf{Param}$, so applying the group action[8] transforms the conserved quantity $Q_M$ to $Q_{g^T M g}$. As discussed in Section 3, applying $g$ to a minimum $\boldsymbol{\theta}^*$ of $\mathcal{L}$ yields another minimum $g \cdot \boldsymbol{\theta}^*$; hence applying symmetries leads to a partial parameterization of flat minima. Note that, in general, we may lack sufficient number of $Q_M$ to fully parameterize a flat minimum. For example, in the linear network $UVx$, $G = \mathrm{GL}_h$ and flat minima generically have $h^2$ dimensions, whereas the number of independent nonzero $Q_M$ is $h(h+1)/2$, which is the dimension of the space of symmetric matrices $M = M^T$ in $\mathfrak{gl}_h$.

In gradient descent, the values of these conserved quantities may change due to the time discretization. However, the change in $Q$ is expected to be small. For example, in two-layer linear networks, the change of $Q$ is bounded by the square of learning rate. Appendix E contains derivations and empirical observations of the magnitude of change in $Q$.

**Relation to Noether's theorem.** In physics, Noether's theorem (Noether, 1918) states that continuous symmetries give rise to conserved quantities. Recently, Tanaka & Kunin (2021) showed that Noether's theorem can also be applied to GD by approximating it as a *second order* GF. We show that in the limit where the second order GF reduces to first order GF (10), results from Noether's theorem reduce to our conservation law $\langle \overline{M}_{\boldsymbol{\theta}}, \nabla \mathcal{L} \rangle = 0$ (6). In short, using Noether's theorem, the conserved Noether current is $J_M = e^{t/\tau} J_{0M}$ with $J_{0M} = \langle \overline{M}_{\boldsymbol{\theta}}, \varepsilon^{-1} \dot{\boldsymbol{\theta}} \rangle$. In the limit $\tau \to 0$, using (10), $J_{0M} = \langle \overline{M}_{\boldsymbol{\theta}}, \nabla \mathcal{L} \rangle = 0$ and the conservation $dJ_M/dt = 0$ implies $J_{0M} = 0$, meaning we recover (6). Details appear in Appendix B.

**Examples.** We present examples of conserved quantities for two-layer neural networks. all of which directly generalize to the multi-layer case. See Appendix C.9 for full derivations (which heavily rely on properties of the trace). We adopt the notation of Section 3.3.

**Example 5.1. General equivariant activation** Suppose $\sigma$ is equivariant under a linear action of a subgroup $G \subseteq \mathrm{GL}_h(\mathbb{R})$, so that $\pi(g)\sigma(z) = \sigma(gz)$. Then the two-layer network $F(z) = U\sigma(Vz)$ is invariant under $G$, as is the loss function. For symmetric $M \in \mathfrak{g}$, Proposition 5.1 yields the following conserved quantity:

$$Q_M : \mathbf{Param} \to \mathbb{R}, \qquad Q_M(U, V) = \mathrm{Tr}\left[V^T M V\right] - \mathrm{Tr}\left[U^T U d\pi(M)\right] \tag{12}$$

Indeed, this follows from the fact that $M \cdot (U, V) = (-U d\pi(M), MV)$, as in (5).

**Example 5.2. Imbalance in linear layers** Suppose the network is linear. Then $\sigma(z) = z$ and the loss is invariant under $\mathrm{GL}_h(\mathbb{R})$. For symmetric $M$ we have the conserved quantity $Q_M(U, V) = \mathrm{Tr}\left[(VV^T - U^T U)M\right]$. Moreover, each component of the matrix $VV^T - U^T U$ is conserved.

**Example 5.3. Homogeneous activation under scaling** Suppose $\sigma$ is a homogeneous activation of degree $\alpha$. Let $G = (\mathbb{R}_{>0})^h$ be the positive rescaling group, so that $\sigma(gz) = g^\alpha \sigma(z)$ for any $g \in G$ and $z \in \mathbb{R}^h$. Note that the Lie algebra of $G$ consists of all diagonal matrices in $\mathfrak{gl}_h$, so that, in particular, each $M \in \mathfrak{g}$ is symmetric. Since $d\pi(M) = \alpha M$ for any $M \in \mathfrak{g}$, we obtain the conserved quantity $Q_M(U, V) = \mathrm{Tr}\left[(VV^T - \alpha U^T U)M\right]$. Using the basis $M = E_{kk}$, we see that $\mathbf{Q} = \mathrm{diag}\left[VV^T - \alpha U^T U\right]$ is conserved (here, $\mathrm{diag}[A]$ is the leading diagonal). Special cases of this are LeakyReLU and ReLU with $\alpha = 1$.

**Example 5.4. Radial rescaling activations** Let $\sigma$ be such a radial rescaling activation. As in Section 3.3, the orthogonal group $G = O(h)$ is a symmetry of $\mathcal{L}$. The Lie algebra $\mathfrak{g} = \mathfrak{so}_h$ comprises anti-symmetric matrices, and so Proposition 5.1 yields no non-trivial conserved quantities. However, using the canonical basis of $\mathfrak{g} = \mathfrak{so}_h$ given by $E_{[kl]} = E_{kl} - E_{lk}$ (so $[kl]$ indicates anti-symmetrized indices), one uses equation (11) to deduce the following novel result (see Appendix C.9):

**Theorem 5.2.** *When $\sigma$ is a radial rescaling activation, we have:*

$$V\dot{V}^T - \dot{V}V^T + U^T\dot{U} - \dot{U}^T U = 0 \tag{13}$$

*for any $(U, V) \in \mathbf{Param}$, where the dots indicate derivatives with respect to gradient flow.*

Expanding the $(k, l)$ entry of the matrix on the left-hand-side of (13), we obtain: $\sum_{s=1}^n r_{U,s;kl}^2 \dot{\phi}_{U,s;kl} + \sum_{s=1}^m r_{V^T,s;kl}^2 \dot{\phi}_{V^T,s;kl} = 0$, where $(r_{U,s;kl}, \phi_{U,s;kl})$ and $(r_{V^T,s;kl}, \phi_{V^T,s;kl})$

---

[8]Note that this procedure only works if $g^T M g$ belongs to $\mathfrak{g}$, which is the case the examples we consider.

Figure 2: Overview of empirical observations with more details in Appendix G, H, and I. (a) In a two-layer neural network, the convergence rate depends on the conserved quantity $Q$. (b) The distribution of the eigenvalues of the Hessian at the minimum is related to the value of $Q$. (c) The ensemble created by group actions has similar loss values when $\varepsilon$ is small. (d) The ensemble model improves robustness against fast gradient sign method attacks.

are the 2D polar coordinates of the points $(U_{sk}, U_{sl})$ and $(V_{sk}^T, V_{sl}^T)$. This is analogous to the "angular momentum" in 2D, that is: $x \wedge \dot{x} = r^2 \dot{\phi}$. Intuitively, Theorem 5.2 implies that in every 2D plane $(k, l)$, the angular momenta of the rows of $U$ and the columns of $V$ sum to zero. These results also apply to linear networks $F(x) = UVx$, since rotational symmetries are linear.

## 6 APPLICATIONS

We present a set of experiments aimed at assessing the utility of the nonlinear group action and conserved quantities[9]. A summary of the results are shown in Figure 2. We show that the value of conserved quantities can impact convergence rate and generalizability. We also find the nonlinear action to be viable for ensemble building to improve robustness under certain adversarial attacks.

**Exploration of the minimum.** While $Q$ is often unbounded, common initialization methods such as (Glorot & Bengio, 2010) limit the values of $Q$ to a small range (Appendix F). As a result, only a small part of the minimum is reachable by the models. Symmetries allow us to explore portions of flat minima that gradient descent rarely reaches.

**Convergence rate and generalizability.** Conserved quantities are by definition unchanged during gradient flows. By relating the values of conserved quantities to convergence rate and model generalizability, we have access to properties of the trajectory and the final model before the gradient flow starts. This knowledge allows us to choose good conserved quantity values at initialization. In Appendix G, we derive the relation between $Q$ and convergence rate for two example optimization problems, and provide numerical evidence that initializing parameters with certain conserved quantity values accelerates convergence. In Appendix H, we derive the relation between conserved quantities and sharpness of minima in a simple two-layer network, and show empirically that $Q$ values affect the eigenvalues of the Hessian (and possibly generalizability) in larger networks.

**Ensemble models.** Applying the nonlinear group action allows us to obtain an ensemble without any retraining or searching. We show that even with stochasticity in the data, the loss is approximately unchanged under the group action. The ensemble has the potential to improve robustness under adversarial attacks (Appendix I).

## 7 DISCUSSION

In this paper, we present a general framework of equivariance and introduce a new class of nonlinear, data-dependent symmetries of neural network parameter spaces. These symmetries give rise to conserved quantities in gradient flows, with important implications in improving optimization and robustness of neural networks. While our work sheds new light onto the link between symmetries and group, it contains several limitations, which merit further investigation. First, we have not been able to determine conserved quantities in the radial rescaling case, only a differential equation that gradient flow curves must satisfy. Second, one major contribution of this paper is the non-linear group action of Section 4. However, our formulation only gurantees full $\mathrm{GL}_h$ equivariance for batch size $k = 1$. In future work, we plan to explore more consequences and variations of this non-linear group action, with the hope of generalizing to greater batch size. Finally, in many cases, parameter space symmetries lead to model compression: i.e., finding a lower-dimension space of parameters with the same expressivity of the original space.

---

[9]Our code is available at https://github.com/Rose-STL-Lab/Gradient-Flow-Symmetry.

## ACKNOWLEDGMENTS

This work was supported in part by U.S. Department Of Energy, Office of Science grant DE-SC0022255, U. S. Army Research Office grant W911NF-20-1-0334, and NSF grants #2134274 and #2146343. I. Ganev was supported by the NWO under the CORTEX project (NWA.1160.18.316). R. Walters was supported by the Roux Institute and the Harold Alfond Foundation and NSF grants #2107256 and #2134178.

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

CONTENTS

## A  ADDITIONAL RELATED WORKS

**Mode connectivity / flat regions / ensembles**  In neural networks, the optima of the loss functions are connected by curves or volumes, on which the loss is almost constant (Freeman & Bruna, 2017; Garipov et al., 2018; Draxler et al., 2018; Benton et al., 2021; Izmailov et al., 2018). In these works, various algorithms have been proposed to find these low-cost curves, which provides a low-cost way to create an ensemble of models from a single trained model. Other related works include linear mode connectivity (Frankle et al., 2020), using mode connectivity for loss landscape analysis (Gotmare et al., 2018), and studying flat region by removing symmetry (Pittorino et al., 2022).

**Sharpness of minima and generalization**  Recent theory and empirical studies suggest that sharp minimum do not generalize well (Hochreiter & Schmidhuber, 1997; Keskar et al., 2017; Petzka et al., 2021). Explicitly searching for flat minimum has been shown to improve generalization bounds and model performance (Chaudhari et al., 2017; Foret et al., 2020; Kim et al., 2022). The sharpness of minimum can be defined using the largest loss value in the neighborhood of a minima (Keskar et al., 2017; Foret et al., 2020; Kim et al., 2022), visualization of the change in loss under perturbation with various magnitudes on weights (Izmailov et al., 2018), singularity of Hessian (Sagun et al., 2017), or the volume of the basin that contains the minimum (approximated by Radon measure (Zhou et al., 2020) or product of the eigenvalues of the Hessian (Wu et al., 2017)). Under most of these metrics, however, equivalent models can be built to have minimum with different sharpness but same generalization ability (Dinh et al., 2017). Applications include explaining the good generalization of SGD by examining asymmetric minimum (He et al., 2019), and new pruning algorithms that search for minimizers close to flat regions (Chao et al., 2020).

**Parameter space symmetry and optimization**  While sets of parameter values related by the symmetries produce the same output, the gradients at these points are different, resulting in different learning dynamics (Kunin et al., 2021; Van Laarhoven, 2017). This insight leads to a number of new advances in optimization. Neyshabur et al. (2015) and Meng et al. (2019) propose optimization algorithms that are invariant to symmetry transformations on parameters. Armenta et al. (2023) and Zhao et al. (2022) apply loss-invariant transformations on parameters to improve the magnitude of gradients, and consequently the convergence speed.

The structures encoded in known symmetries have also led to new optimization methods and insights of the loss landscape. Bamler & Mandt (2018) improves the convergence speed when optimizing in the direction of weakly broken symmetry. Zhang et al. (2020) discusses how symmetry helps in obtaining global minimizers for a class of nonconvex problems. The potential relevance of continuous symmetries in optimization problems was also discussed in Leake & Vishnoi (2021), which also provides an overview of Lie groups.

## B  RELATION TO NOETHER'S THEOREM

We will now show how the approach in Tanaka & Kunin (2021) relates to our conservation law $dQ/dt = \left\langle \dot{\boldsymbol{\theta}}, M\boldsymbol{\theta} \right\rangle = 0$. Assuming a small time-step $\tau \ll 1$, we can write GD as $\boldsymbol{\theta}(t+\tau) - \boldsymbol{\theta}(t) = -\tilde{\varepsilon}\nabla\mathcal{L}(\boldsymbol{\theta}(t))$. Expanding the l.h.s to second order in $\tau$ and discarding $O(\tau^3)$ terms defines the 2nd order GF equation

$$\text{2nd order GF:} \qquad \frac{d\boldsymbol{\theta}}{dt} + \frac{\tau}{2}\frac{d^2\boldsymbol{\theta}}{dt^2} = -\varepsilon\nabla\mathcal{L}. \tag{14}$$

Here $\varepsilon = \tilde{\varepsilon}/\tau$. To use Noether's theorem, the dynamics (i.e. GF) must be a variational (Euler-Lagrange (EL)) equation derived from an "action" $S(\boldsymbol{\theta})$ (objective function), which for (14) is the time integral of Bregman Lagrangian (Wibisono & Wilson, 2015) $L$

$$S(\boldsymbol{\theta}) = \int dt L(\boldsymbol{\theta}(t), \dot{\boldsymbol{\theta}}(t); t) = \int_\gamma dt e^{t/\tau} \left[ \frac{\tau}{2} \left\langle \dot{\boldsymbol{\theta}}, \varepsilon^{-1}\dot{\boldsymbol{\theta}} \right\rangle - \mathcal{L}(\boldsymbol{\theta}) \right] \tag{15}$$

where $\boldsymbol{\theta} : \mathbb{R} \to \mathbf{Param}$ is a trajectory (flow path) in $\mathbf{Param}$, parametrized by $t$. The variational EL equations find the paths $\gamma^*$ which minimize the action, meaning $\partial S_\gamma/\partial\gamma|_{\gamma^*} = 0$.

**Noether's theorem**  states that if $M \in \mathfrak{g}$ is a symmetry of the *action* $S(\boldsymbol{\theta})$ (15) (not just the loss $\mathcal{L}(\boldsymbol{\theta})$), then the Noether current $J_M$ is conserved

$$\text{Noether current:} \qquad J_M = \left\langle \overline{M}\boldsymbol{\theta}, \frac{\partial L}{\partial\dot{\boldsymbol{\theta}}} \right\rangle = e^{t/\tau}\left\langle \overline{M}\boldsymbol{\theta}, \varepsilon^{-1}\dot{\boldsymbol{\theta}} \right\rangle = e^{t/\tau}J_{0M},$$

$$\text{Conservation:} \qquad \frac{dJ_M}{dt} = e^{t/\tau}\left[\frac{1}{\tau}J_{0M} + \frac{dJ_{0M}}{dt}\right] = 0, \quad \Rightarrow \quad J_{0M}(t) = J_{0M}e^{-t/\tau} \tag{16}$$

Tanaka & Kunin (2021) also derived the Noether current (16), but concludes that because $L(\boldsymbol{\theta}, \dot{\boldsymbol{\theta}}) \neq \mathcal{L}(\boldsymbol{\theta})$, the symmetries are "broken" and therefore doesn't derive conserved charges for the types of symmetries we discussed above. However, while Tanaka & Kunin (2021) focuses on 2nd order GF, we note that our conserved $Q$ were derived for *first order GF*, which is found from the $\tau \to 0$ limit of 2nd oder GF. In this limit $L \to e^{t/\tau}\mathcal{L}$ and thus symmetries of $L$ also becomes symmetries of $\mathcal{L}$. When $\tau \to 0$, 2nd order GF reduces to $\varepsilon^{-1}\dot{\boldsymbol{\theta}} = -\nabla\mathcal{L}$ the conserved charge goes to

$$\lim_{\tau\to 0} J_{0M} = \left\langle \overline{M}\boldsymbol{\theta}, \nabla\mathcal{L} \right\rangle = J_{0M}(0)\lim_{\tau\to 0} e^{-t/\tau} = 0, \tag{17}$$

which means that we recover the invariance under infinitesimal action (6). In fact, for linear symmetries and symmetric $M \in \mathfrak{g}$, $J_{0M} = dQ_M/dt = 0$.

## C  NEURAL NETWORKS: LINEAR GROUP ACTIONS

In this appendix, we provide an extended discussion of the topics of Section 3, including full proofs of all results. Specifically, after some technical background material on Jacobians and differentials, we specify our conventions for neural network parameter space symmetries. In contrast to the

discussion of the main text, we (1) assume that neural networks have biases, and (2) focus on the multi-layer case rather than just the two-layer case. We then turn our attention to group actions of the parameter space that leave the loss invariant, and the resulting infinitesimal symmetries. The groups we consider are all subgroups of a large group of change-of-basis transformations of the hidden feature spaces; we call this group the 'hidden symmetry group'. We also compute the dimensions of generic extended flat minima in various relevant examples. Finally, we explore consequences of invariant group actions for conserved quantities.

## C.1 JACOBIANS AND DIFFERENTIALS

In this section, we summarize background material on Jacobians and differentials. We adopt notation and conventions from differential geometry. Let $U \subset \mathbb{R}^n$ be an open subset of Euclidean space $\mathbb{R}^n$, and let $F : U \to \mathbb{R}^m$ be a differentiable function. Let $F_1, \ldots, F_m : U \to \mathbb{R}$ be the components of $F$, so that $F(u) = (F_1(u), \ldots, F_m(u))$. The Jacobian of $F$, also know as differential of $F$, at $u \in U$ is the following matrix of partial derivatives evaluated at $u$:

$$dF_u = \begin{bmatrix} \frac{\partial F_1}{\partial x_1}\big|_u & \frac{\partial F_1}{\partial x_2}\big|_u & \cdots & \frac{\partial F_1}{\partial x_n}\big|_u \\ \frac{\partial F_2}{\partial x_1}\big|_u & \frac{\partial F_2}{\partial x_2}\big|_u & \cdots & \frac{\partial F_2}{\partial x_n}\big|_u \\ \vdots & \vdots & \ddots & \vdots \\ \frac{\partial F_m}{\partial x_1}\big|_u & \frac{\partial F_m}{\partial x_2}\big|_u & \cdots & \frac{\partial F_m}{\partial x_n}\big|_u \end{bmatrix}$$

The differential $dF_u$ defines a linear map from $\mathbb{R}^n$ to $\mathbb{R}^m$, that is, an element of $\mathbb{R}^{m \times n}$. Observe that if $F$ itself is linear, then, as matrices, $dF_u = F$ for all points $u \in U$. If $G : \mathbb{R}^m \to \mathbb{R}^p$ is another differentiable map, then the chain rule implies that, for all $u \in \mathbb{R}^n$, we have:

$$d(G \circ F)_u = dG_{F(u)} \circ dF_u.$$

In the special case the $m = 1$, the differential is a $1 \times n$ row vector, and the *gradient* $\nabla_u F$ of $F$ at $u \in \mathbb{R}^n$ is defined as the transpose of the Jacobian $dF_u$:

$$\nabla_v F = (dF_u)^T = \begin{bmatrix} \frac{\partial F}{\partial x_1}\big|_u & \cdots & \frac{\partial F}{\partial x_n}\big|_u \end{bmatrix}^T = \left( \frac{\partial F}{\partial x_i}\Big|_u \right)_{i=1}^n \in \mathbb{R}^n$$

## C.2 NEURAL NETWORK PARAMETER SPACES

Consider a neural network with $L$ layers, input dimension $n_0 = n$, output dimension $n_L = m$, and hidden dimensions given by $n_1, \ldots, n_{L-1}$. For convenience, we group the dimensions into a tuple $\mathbf{n} = (n_0, n_1, \ldots, n_L)$. The parameter space is given by:

$$\mathbf{Param}(\mathbf{n}) = \mathbb{R}^{n_L \times n_{L-1}} \times \mathbb{R}^{n_{L-1} \times n_{L-2}} \times \cdots \times \mathbb{R}^{n_2 \times n_1} \times \mathbb{R}^{n_1 \times n_0} \times \mathbb{R}^{n_L} \times \mathbb{R}^{n_{L-1}} \times \cdots \times \mathbb{R}^{n_1}.$$

We write an element therein as a pair $\boldsymbol{\theta} = (\mathbf{W}, \mathbf{b})$ of tuples $\mathbf{W} = (W_L, \ldots, W_1)$ and $\mathbf{b} = (b_L, \ldots, b_1)$, so that $W_i$ is a $n_i \times n_{i-1}$ matrix and $b_i$ is a vector in $\mathbb{R}^{n_i}$ for $i = 1, \ldots, L$. When $\mathbf{n}$ is clear from context, we write simply $\mathbf{Param}$ for the parameter space. Fix a piecewise differentiable function $\sigma_i : \mathbb{R}^{n_i} \to \mathbb{R}^{n_i}$ for each $i = 1, \ldots, L$. The activations can be pointwise (as is conventionally the case), but are not necessarily so. The feedforward function

$$F = F_{\boldsymbol{\theta}} : \mathbb{R}^n \to \mathbb{R}^m$$

corresponding to parameters $\boldsymbol{\theta} = (\mathbf{W}, \mathbf{b}) \in \mathbf{Param}$ with activations $\sigma_i$ is defined in the usual recursive way. To be explicit, we define the partial feedforward function $F_{\boldsymbol{\theta}, i} = F_i : \mathbb{R}^n \to \mathbb{R}^{n_i}$ to be the map taking $x \in \mathbb{R}^n$ to $\sigma_i(W_i F_{i-1}(x) + b_i)$, for $i = 1, \ldots, L$, with $F_0 = \mathrm{id}_{\mathbb{R}^{n_0}}$. Then the feedforward function is $F = F_L$. The "loss function" $\mathcal{L}$ of our model is defined as:

$$\mathcal{L} : \mathbf{Param} \times \mathbf{Data} \to \mathbb{R}, \qquad \mathcal{L}(\boldsymbol{\theta}, (x, y)) = \mathrm{Cost}(y, F_{\boldsymbol{\theta}}(x)). \qquad (18)$$

where $\mathbf{Data} = \mathbb{R}^{n_0} \times \mathbb{R}^{n_L}$ is the space of data (i.e., possible training data pairs), and $\mathrm{Cost} : \mathbb{R}^{n_L} \times \mathbb{R}^{n_L} \to \mathbb{R}$ is a differentiable cost function, such as mean square error or cross-entropy. Many, if not most, of our results involve properties of the loss function $\mathcal{L}$ that hold for any training data. Hence, unless specified otherwise, we take a fixed batch of training data $\{(x_i, y_i)\}_{i=1}^k \subseteq \mathbf{Data}$, and consider the loss to be a function of the parameters only.

The above constructions generalize to multiple samples. Specifically, instead of $x \in \mathbb{R}^{n_0}$ and $y \in \mathbb{R}^{n_L}$, one has matrices $X \in \mathbb{R}^{n_o \times k}$ and $Y \in \mathbb{R}^{n_L \times k}$ whose columns are the $k$ samples. Additionally, one uses the Frobenius norm of $n_L \times k$ matrices to compute the loss function. The $i$-th partial feedforward function is $F_{\boldsymbol{\theta},i} : \mathbb{R}^{n_0 \times k} \to \mathbb{R}^{n_i \times k}$, and we have the feedforward function $F_{\boldsymbol{\theta}} : \mathbb{R}^{n_0 \times k} \to \mathbb{R}^{n_L \times k}$. (We use the same notation as in the case $k = 1$; the number of samples will be understood from context.)

**Example C.1.** Consider the case $L = 2$ with no biases and no output activation. The dimension vector is $(n_0, n_1, n_2) = (n, h, m)$, so the parameter space is $\mathbf{Param}(n, h, m) = \mathbb{R}^{h \times n} \times \mathbb{R}^{m \times h}$. Taking the mean square error cost function, the loss function for data $(X, Y) \in \mathbb{R}^{n \times k} \times \mathbb{R}^{m \times k}$ takes the form $\mathcal{L}(\boldsymbol{\theta}, (X, Y)) = \frac{1}{k} \|Y - U\sigma(VX)\|^2$, where $\boldsymbol{\theta} = (W_2, W_1) = (U, V) \in \mathbf{Param}$.

## C.3  ACTION OF CONTINUOUS GROUPS AND INFINITESIMAL SYMMETRIES

Let $G$ be a group. An action of $G$ on the parameter space $\mathbf{Param}$ is a function $\cdot : G \times \mathbf{Param} \to \mathbf{Param}$, written as $g \cdot \boldsymbol{\theta}$, that satisfies the unit and multiplication axioms of the group, meaning $I \cdot \boldsymbol{\theta} = \boldsymbol{\theta}$ where $I$ is the identity of $G$, and $g_1 \cdot (g_2 \cdot \boldsymbol{\theta}) = (g_1 g_2) \cdot \boldsymbol{\theta}$ for all $g_1, g_2 \in G$. Recall that we say an action $G \times \mathbf{Param} \to \mathbf{Param}$ is a *symmetry* of $\mathbf{Param}$ with respect to $\mathcal{L}$ if it leaves the loss function invariant, that is:

$$\mathcal{L}(g \cdot \boldsymbol{\theta}) = \mathcal{L}(\boldsymbol{\theta}), \qquad\qquad \forall \boldsymbol{\theta} \in \mathbf{Param}, \quad g \in G \qquad (19)$$

The groups within the scope of this paper are all matrix Lie groups, which are topologically closed subgroups $G \subseteq \mathrm{GL}_n(\mathbb{R})$ of the general linear group of invertible $n \times n$ real matrices. Any smooth action of such a group induces an action of the infinitesimal generators of the group, i.e., elements of its Lie algebra. Concretely, let $\mathfrak{g} = \mathrm{Lie}(G) = T_I G$ be the Lie algebra, which can be thought of as a certain subspace of matrices in $\mathfrak{gl}_n = \mathbb{R}^{n \times n}$, or (equivalently) as the tangent space[10]. at the identity $I$ of $G$. For every matrix $M \in \mathfrak{g}$, we have an exponential map $\exp_M : \mathbb{R} \to G$ defined as $\exp_M(t) = \sum_{k=0}^{\infty} \frac{(tM)^k}{k!}$. Given an action of $G$ on $\mathbf{Param}$, the *infinitesimal action* of $M \in \mathfrak{g}$ is a vector field $\overline{M}$:

$$\text{Infinitesimal action of } M \text{ vector field:} \quad \overline{M}_{\boldsymbol{\theta}} := \frac{d}{dt}\Big|_{t \to 0} \left(\exp_M(t) \cdot \boldsymbol{\theta}\right), \qquad \forall \boldsymbol{\theta} \in \mathbf{Param}. \quad (20)$$

Hence, the value of the vector field $\overline{M}$ at the parameter value $\boldsymbol{\theta}$ is given by the derivative at zero of the function $t \mapsto \left(\exp_M(t) \cdot \boldsymbol{\theta}\right)$. In the case of a parameter space symmetry, the invariance of $\mathcal{L}$ translates into the orthogonality condition in Proposition 3.4, where the inner product $\langle, \rangle :$ $\mathbf{Param} \times \mathbf{Param} \to \mathbb{R}$ is calculated by contracting all indices, e.g. $\langle A, B \rangle = \sum_{ijk\ldots} A_{ijk\ldots} B_{ijk\ldots}$.

*Proof of Proposition 3.4.* The gradient is the transpose of the Jacobian (see Section C.1), so the left-hand-side becomes $d\mathcal{L}_{\boldsymbol{\theta}} \left(\frac{d}{dt}\big|_0 \left(\exp_M(t) \cdot \boldsymbol{\theta}\right)\right)$. We compute:

$$d\mathcal{L}_{\boldsymbol{\theta}} \left(\frac{d}{dt}\Big|_0 \left(\exp_M(t) \cdot \boldsymbol{\theta}\right)\right) = \frac{d}{dt}\Big|_0 \mathcal{L}(\exp_M(t) \cdot \boldsymbol{\theta}) = \frac{d}{dt}\Big|_0 \mathcal{L}(\boldsymbol{\theta}) = 0$$

where the first equality follows by the chain rule, the second equality uses the invariance of $\mathcal{L}$, and the third equality follows since $\mathcal{L}(\boldsymbol{\theta})$ does not depend on $t$. $\qquad\square$

Next, we comment on the case of a linear action. Observe that the parameter space is a vector space of dimension $d = \dim(\mathbf{Param}) = \sum_{i=1}^{L} n_i(1 + n_{i-1})$. Hence, there is an isomorphism $\mathbf{Param} \simeq \mathbb{R}^d$ which flattens any tuple $\boldsymbol{\theta} = (\mathbf{W}, \mathbf{b})$ into a vector in $\mathbb{R}^d$. We can identify the group $\mathrm{GL}(\mathbf{Param})$ of all invertible linear transformations of the parameter space with the group $\mathrm{GL}_d(\mathbb{R})$ of invertible $d \times d$ matrices, and the Lie algebra of $\mathrm{GL}(\mathbf{Param})$ with $\mathfrak{gl}_d = \mathbb{R}^{d \times d}$. Suppose $G$ acts linearly on the parameter space. Then we can identify[11] $G$ with a subgroup of $\mathrm{GL}(\mathbf{Param})$, acting on $\mathbf{Param}$ with matrix multiplication. Similarly, we can identify the Lie algebra $\mathfrak{g}$ of $G$ with a Lie subalgebra of $\mathfrak{gl}_d = \mathbb{R}^{d \times d}$. In this case, the infinitesimal action is given by matrix multiplication: $\overline{M}_{\boldsymbol{\theta}} = M \cdot \boldsymbol{\theta}$. We recover Equation (6).

---

[10]Hence, elements of the Lie algebra are 'velocities' at the identity of $G$. More precisely, for every Lie algebra element $\xi$, there is a path $\gamma_\xi : (-\epsilon, \epsilon) \to G$ whose value at 0 is the identity of $G$ and whose derivative (i.e., velocity) at zero is $\xi$.

[11]Modding out by the kernel, if necessary.

## C.4 THE HIDDEN SYMMETRY GROUP

Consider a neural network with $L$ layers and dimensions $\mathbf{n}$. The *hidden symmetry group* corresponding to dimensions $\mathbf{n}$ is defined as the following product of general linear groups:

$$\mathrm{GL}_{\mathbf{n}^{\text{hidden}}} = \mathrm{GL}_{n_1} \times \cdots \times \mathrm{GL}_{n_{L-1}}$$

An element is a tuple of invertible matrices $\mathbf{g} = (g_1, \ldots, g_{L-1})$, where $g_i \in \mathrm{GL}_{n_i}$. Consider the action of the hidden symmetry group on the parameter space given by

$$\mathrm{GL}_{\mathbf{n}^{\text{hidden}}} \circlearrowright \mathbf{Param}(\mathbf{n}) \qquad \mathbf{g} \cdot (\mathbf{W}, \mathbf{b}) = \left( g_i W_i g_{i-1}^{-1} , \; g_i b_i \right)_{i=1}^{L} . \tag{21}$$

where $g_0 = \mathrm{id}_{n_0}$ and $g_L = \mathrm{id}_{n_L}$. This action amounts to changing the basis at each hidden feature space. The Lie algebra of $\mathrm{GL}_{\mathbf{n}^{\text{hidden}}}$ is

$$\mathfrak{gl}_{\mathbf{n}^{\text{hidden}}} = \mathfrak{gl}_{n_1} \times \cdots \times \mathfrak{gl}_{n_{L_1}},$$

and the infinitesimal action of the tuple $M \in \mathfrak{gl}_{\mathbf{n}^{\text{hidden}}}$ is given by:

$$\mathfrak{gl}_{\mathbf{n}^{\text{hidden}}} \circlearrowright \mathbf{Param}(\mathbf{n}) \qquad M \cdot (\mathbf{W}, \mathbf{b}) \mapsto (M_i W_i - W_i M_{i-1} , \; M_i b_i)_{i=1}^{L}$$

where we set $M_0$ and $M_L$ to be the zero matrices.

**Example C.2.** In the case $L = 2$, with dimension vector $\mathbf{n} = (n, h, m)$ and no biases. The hidden symmetry group is $\mathrm{GL}_h$ with Lie algebra $\mathfrak{gl}_h$. The action of the group and the infinitesimal action of the Lie algebra are given by:

$$\mathrm{GL}_h \circlearrowright \mathbf{Param}(n, h, m) = \mathbb{R}^{h \times n} \times \mathbb{R}^{m \times h} \qquad g \cdot (V, U) = \left( gV, Ug^{-1} \right)$$

$$\mathfrak{gl}_h \circlearrowright \mathbf{Param}(n, h, m) = \mathbb{R}^{h \times n} \times \mathbb{R}^{m \times h} \qquad M \cdot (V, U) = (MV, -UM)$$

## C.5 LINEAR SYMMETRIES

Consider a feedforward fully-connected neural network with widths $\mathbf{n} = (n_0, \ldots, n_L)$, so that the parameters space consists of tuples of weights and biases $\boldsymbol{\theta} = (W_i \in \mathbb{R}^{n_i \times n_{i-1}}, b_i \in \mathbb{R}^{n_i})_{i=1}^{L}$. For each hidden layer $0 < i < L$, let $G_i$ be a subgroup of $\mathrm{GL}_{n_i}$, and let $\pi_i : G_i \to \mathrm{GL}_{n_i}(\mathbb{R})$ be a representation (in many cases, we take $\pi_i(g) = g$). Hence the product $G = G_1 \times G_{L-1}$ is a subgroup of the hidden symmetry group $\mathrm{GL}_{\mathbf{n}^{\text{hidden}}}$. Define an action of $G$ on $\mathbf{Param}$ via

$$\forall g = (g_1, \ldots, g_L) \in G, \qquad g \cdot W_i = g_i W_i \pi_{i-1}(g_{i-1}^{-1}) \qquad g \cdot b_i = g_i b_i \tag{22}$$

where $g_0$ and $g_L$ are the identity matrices $\mathrm{id}_{n_0}$ and $\mathrm{id}_{n_L}$, respectively. This is a version of the action defined in (21), with the addition of the twists resulting from the representations $\pi_i$.

We now consider the resulting infinitesimal action. For each $i$, the representation $\pi_i$ induces a Lie algebra representation $d(\pi_i) : \mathfrak{g}_i \to \mathfrak{gl}_{n_i}$. The infinitesimal action of the Lie algebra $\mathfrak{g} = \mathfrak{g}_1 \times \cdots \times \mathfrak{g}_L$ induced by 22 is given by:

$$\forall M = (M_1, \ldots, M_L) \in \mathfrak{g}, \quad M \cdot W_i = M_i W_i - W_i d(\pi_{i-1})(M_{i-1}) \quad M \cdot b_i = M_i b_i \tag{23}$$

The proof of the first part of the following Proposition proceeds by induction, where the key computation is that of from (4). The second part relies on (6).

**Proposition C.1.** *Suppose that, for each $i = 1, \ldots, L$, the activation $\sigma_i$ intertwines the two actions of $G_i$, that is, $\sigma_i(g_i z_i) = \pi_i(g_i)\sigma(z_i)$ for all $g_i \in G_i$, $z_i \in \mathbb{R}^{n_i}$. Then:*

1. *(Combined equivariance of activations) The action of $G = G_1 \times \cdots \times G_L$ defined in 22 is a symmetry of the parameter space.*

2. *(Infinitesimal equivariant action) The action of $\mathfrak{g} = \mathfrak{g}_1 \times \cdots \times \mathfrak{g}_L$ defined in 23 satisfies $\langle \nabla_{\boldsymbol{\theta}} \mathcal{L}, M \cdot \boldsymbol{\theta} \rangle$ for all $\boldsymbol{\theta} \in \mathbf{Param}$ and all $M \in \mathfrak{g}$.*

*Proof.* Let $\mathbf{g} = (g_i)_{i=1}^{L-1} \in G$, so that $g_i \in G_i \subseteq \mathrm{GL}_{n_i}$. As usual, set $g_0 = \mathrm{id}_{n_0}$ and $g_L = \mathrm{id}_{n_L}$. Also set $\pi_0$ and $\pi_L$ to be the identity maps on $\mathrm{GL}_{n_0}$ and $\mathrm{GL}_{n_L}$, respectively. Fix parameters $\boldsymbol{\theta}$

and an input value $x \in \mathbb{R}^n$. We show by induction that the following relation between the partial feedforward functions holds:

$$F_{\mathbf{g} \cdot \boldsymbol{\theta}, i}(x) = \pi_i(g_i)(F_{\boldsymbol{\theta}, i}(x))$$

for $i = 0, \ldots, L$. The base step is trivial. For the induction step, we use the recursive definition of the partial feedforward functions:

$$
\begin{aligned}
F_{\mathbf{g} \cdot \boldsymbol{\theta}, i}(x) &= \sigma_i(g_i W_i \pi_{i-1}(g_{i-1}^{-1}) F_{\mathbf{g} \cdot \boldsymbol{\theta}, i-1}(x) + g_i b_i) \\
&= \sigma_i(g_i (W_i \pi_{i-1}(g_{i-1}^{-1}) \pi_{i-1}(g_{i-1}) F_{\boldsymbol{\theta}, i-1}(x) + b_i) \\
&= \pi_i(g_i) \sigma_i(W_i F_{\boldsymbol{\theta}, i-1}(x) + b_i) = \pi_i(g_i) F_{\boldsymbol{\theta}, i}(x)
\end{aligned}
$$

Hence $\boldsymbol{\theta}$ and $g \cdot \boldsymbol{\theta}$ define the same feedforward function. Since the loss function depends on the parameters only through the feedforward function, the first claim follows. The second claim is a consequence of Proposition 3.4. $\square$

Note that two-layer case of the above result amounts to (4) (the argument can be simplified in that case). While Proposition C.1 is stated for feedforward networks, it can easily be adopted to more general settings, such as networks with skip connections and quiver neural networks. Denoting the Jacobian of $\sigma_i : \mathbb{R}^{n_i} \to \mathbb{R}^{n_i}$ at $z \in \mathbb{R}^{n_i}$ by $d(\sigma_i)_z \in \mathbb{R}^{n_i \times n_i}$, the infinitesimal form of version of $\sigma_i(gz) = \pi_i(g)\sigma_i(z)$ is

$$\langle d(\sigma_i)_z, Mz \rangle = d\pi_i(M)\sigma_i(z) \qquad \forall M \in \mathfrak{gl}_{n_i} \tag{24}$$

When the activation is pointwise, we have $Mz \odot \sigma_i'(z) = d(\pi)_i(M)\sigma_i(z)$, where $\odot$ denotes elementwise multiplication. We now illustrate Proposition C.1 through examples in the two-layer case with input dimension $n$, hidden dimension $h$, hidden activation $\sigma$, and output dimension $m$.

**Example C.3. Linear networks** For linear networks, we have $\sigma(x) = x$. One can take $\pi(g) = g$ and $G = \mathrm{GL}_h(\mathbb{R})$.

**Example C.4. Homogeneous activations** Suppose the activation $\sigma : \mathbb{R}^h \to \mathbb{R}^h$ is *homogeneous*, so that (1) $\sigma$ is applied pointwise in the standard basis, and (2) t there exists $\alpha > 0$ such that $\sigma(cz) = c^\alpha \sigma(z)$ for all $c \in \mathbb{R}_{>0}$ and $z \in \mathbb{R}^h$. These $\sigma$ are equivariant under the *positive scaling group* $G \subset \mathrm{GL}_h$ consisting of diagonal matrices with positive diagonal entries. For $g \in G$, we have $g = \mathrm{diag}(\mathbf{c})$ is a diagonal matrix with $\mathbf{c} = (c_1, \ldots, c_h) \in \mathbb{R}_{>0}^h$. For $z = (z_1, \ldots, z_h) \in \mathbb{R}^h$, we have $\sigma(gz) = \sum_j \sigma(c_j z_j) = \sum_j c_j^\alpha \sigma(z_j) = g^\alpha \sigma(z)$. Hence, the equivariance condition holds with $\pi(g) = g^\alpha$. Since $d\pi(M) = \alpha M$ for any element $M$ of the Lie algebra $\mathfrak{g}$ of $G$, the infinitesimal version of rescaling invariance of homogeneous $\sigma$ becomes $Mz \odot \sigma'(z) = \alpha M\sigma(z)$.

**Example C.5. LeakyReLU** This is a special case of homoegeneous activation, defined as $\sigma(z) = \max(z, 0) - s \min(z, 0)$, with $s \in \mathbb{R}_{>0}$. We have $\alpha = 1$, and $\pi(g) = g$. Since $\sigma(z) = z\sigma'(z)$, infinitesimal equivariance becomes $Mz \odot \sigma'(z) = M\sigma(z)$.

**Example C.6. Radial rescaling activations** A less trivial example of continuous symmetries is the case of a radial rescaling activation (Ganev et al., 2022) where for $z \in \mathbb{R}^h \setminus \{0\}$, $\sigma(z) = f(\|z\|)z$ for some function $f : \mathbb{R} \to \mathbb{R}$. Radial rescaling activations are equivariant under rotations of the input: for any orthogonal transformation $g \in O(h)$ (that is, $g^T g = I$) we have $\sigma(gz) = g\sigma(z)$ for all $z \in \mathbb{R}^h$. Indeed, $\sigma(gz) = f(\|gz\|)(gz) = g(f(\|z\|)z) = g\sigma(z)$, where we use the fact that $\|gz\| = z^T g^T gz = z^T z = \|z\|$ for $g \in O(h)$. Hence, (4) is satisfied with $\pi(g) = g$.

## C.6    LINEAR SYMMETRIES LEAD TO EXTENDED, FLAT MINIMA

In this section, we show that, in the case of a linear group action, applying the action of any element of the group to a local minimum yields another local minimum. This fact is a corollary of a more general result; in order to describe it and remove ambiguity, we include the following clarifications. Let $G$ be a matrix Lie group acting as a linear symmetry. Fix a basis $(\theta_1, \ldots, \theta_d)$ of the parameter space. The gradient $\nabla_{\boldsymbol{\theta}} \mathcal{L}$ of the loss $\mathcal{L}$ at a point $\boldsymbol{\theta} \in \mathbf{Param}$ in the parameter space as another vector in $\mathbf{Param} \simeq \mathbb{R}^d$, whose $i$-th coordinate is the partial derivative $\frac{\partial \mathcal{L}}{\partial \theta_i}\big|_{\boldsymbol{\theta}}$. Hence, it makes sense to apply the group action to the gradient: $g \cdot \nabla_{\boldsymbol{\theta}} \mathcal{L}$. We regard vectors in $\mathbf{Param} \simeq \mathbb{R}^d$ as column vectors with $d$ rows. Thus, the transpose of any vector is a row vector with $d$ columns. In the case of the gradient, its transpose at $\boldsymbol{\theta}$ matches the *Jacobian* $d\mathcal{L}_{\boldsymbol{\theta}} \in \mathbb{R}^{1 \times d}$ of $\mathcal{L}$ (see Appendix C.1), that is: $d\mathcal{L}_{\boldsymbol{\theta}} = (\nabla_{\boldsymbol{\theta}} \mathcal{L})^T$. Alternative notation for the Jacobian is $d\mathcal{L}_{\boldsymbol{\theta}_0} = \frac{\partial \mathcal{L}}{\partial \boldsymbol{\theta}}\big|_{\boldsymbol{\theta}_0}$, where we now use $\boldsymbol{\theta}$

as a dummy variable and $\boldsymbol{\theta}_0 \in \mathbf{Param}$ as a specific value. As noted above, we are interested in matrix Lie groups $G \subseteq \mathrm{GL}_d(\mathbb{R}) = \mathrm{GL}(\mathbf{Param})$, and assume that the matrix transpose $g^T$ belongs to $G$ for any $g \in G$. These assumptions hold in all examples of interest. We have the following reformulation of Proposition 3.2:

**Proposition C.2.** *Suppose the action of $G$ on the parameter space is linear and leaves the loss invariant. Then the gradients of $\mathcal{L}$ at any $\boldsymbol{\theta}_0$ and $g \cdot \boldsymbol{\theta}_0$ are related as follows:*

$$g^T \cdot \nabla_{g \cdot \boldsymbol{\theta}_0} \mathcal{L} = \nabla_{\boldsymbol{\theta}_0} \mathcal{L} \qquad \forall g \in G, \ \forall \boldsymbol{\theta}_0 \in \mathbf{Param} \tag{25}$$

*If $\boldsymbol{\theta}^*$ is a critical point (resp. local minimum) of $\mathcal{L}$, then so is $g \cdot \boldsymbol{\theta}^*$.*

*Sketch of proof.* Let $T_g : \mathbf{Param} \to \mathbf{Param}$ be the transformation corresponding to $g \in G$. The the Jacobian $d\mathcal{L}_{\boldsymbol{\theta}_0}$ is given by:

$$d\mathcal{L}_{\boldsymbol{\theta}_0} = \left.\frac{\partial \mathcal{L}}{\partial \boldsymbol{\theta}}\right|_{\boldsymbol{\theta}_0} = \left.\frac{\partial (\mathcal{L} \circ T_g)}{\partial \boldsymbol{\theta}}\right|_{\boldsymbol{\theta}_0} = \left.\frac{\partial \mathcal{L}}{\partial \boldsymbol{\theta}}\right|_{g \cdot \boldsymbol{\theta}_0} \left.\frac{\partial T_g}{\partial \boldsymbol{\theta}}\right|_{\boldsymbol{\theta}_0} = d\mathcal{L}_{g \cdot \boldsymbol{\theta}_0} \circ T_g$$

where we use the definition of the Jacobian, the invariance of the loss ($\mathcal{L} \circ T_g = \mathcal{L}$), the chain rule, and the linearity of the action. The result follows from applying $T_{g^{-1}}$ on the right to both sides, and taking transposes (see Appendix C.1). The last statement follows from the invariance of $\mathcal{L}$ under the action of $G$, and the fact that $\nabla_{\boldsymbol{\theta}^*} \mathcal{L} = 0$ at a critical point $\boldsymbol{\theta}^*$ of $\mathcal{L}$. $\square$

We conclude that, if $\boldsymbol{\theta}^*$ is a critical point, then the set $\{g \cdot \boldsymbol{\theta} \mid g \in G\}$ belongs to the critical locus. This set is known as the *orbit* of $\boldsymbol{\theta}$ under the action of $G$, and is isomorphic to the quotient $G/\mathrm{Stab}_G(\boldsymbol{\theta})$, where $\mathrm{Stab}_G(\boldsymbol{\theta}) = \{g \in G \mid g \cdot \boldsymbol{\theta} = \boldsymbol{\theta}\}$ is the *stabilizer* subgroup of $\boldsymbol{\theta}$ in $G$. In the case of a linear action, the orbit is a smooth manifold. While the results above imply that the critical locus is a union of $G$-orbits, they do not imply, in general, that the critical locus is a single $G$-orbit. They also do not rule out the case that the stabilizer is a somewhat 'large' subgroup of $G$, in which case the orbit would have low dimension. However, in many cases, there is a topologically dense subset of parameter values $\boldsymbol{\theta} \in \mathbf{Param}$ whose orbits all have the same dimension. We call such an orbit a 'generic' orbit. We now turn our attention to examples of two-layer networks where such a generic orbit exists.

### C.6.1 FLAT DIRECTIONS IN THE TWO-LAYER CASE

Recall that the parameter space of a two-layer network is $\mathbf{Param} = \mathbb{R}^{m \times h} \times \mathbb{R}^{h \times n}$, where the dimension vector is $(m, h, n)$, and we write elements as $(U, V)$. The action of $\mathrm{GL}_h$ is $g \cdot (U, V) = (Ug^{-1}, gV)$. Let $\mathbf{Param}^\circ \subseteq \mathbf{Param}$ be the subset of pairs $(U, V)$ where each of $U$ and $V$ have full rank. This is an open dense subset of $\mathbf{Param}$, and is preserved by the $\mathrm{GL}_h$-action.

**Proposition C.3.** *The $\mathrm{GL}_h$-orbit of each element of $\mathbf{Param}^\circ$ has dimension*

$$\dim(\mathrm{Orbit}) = h^2 - \max(0, h - n) \max(0, h - m).$$

*Proof.* Fix $(U, V) \in \mathbf{Param}^\circ$. Suppose $h \leq n$, so that $h^2 - \max(0, h - n) \max(0, h - m) = h^2$. Then $V \in \mathbb{R}^{h \times n}$ defines a surjective linear map, so has a right inverse $V^\dagger \in \mathbb{R}^{n \times h}$. If $g \in \mathrm{GL}_h$ belongs to the stabilizer of $(U, V)$, then we have $gV = V$. Applying $V^\dagger$ on the right to both sides, we obtain $g = \mathrm{id}_h$. Thus, the stabilizer of $(U, V)$ is trivial, and the orbit has dimension equal to the dimension of the group, namely $h^2$. The case $h \leq m$ is similar.

Now suppose $h > \max(n, m)$. In this case, $h^2 - \max(0, h - n) \max(0, h - m) = h(n + m) - nm$. Set $U_0 = [\mathrm{id}_m \quad 0] \in \mathbb{R}^{m \times h}$ and $V_0 = \begin{bmatrix} \mathrm{id}_n \\ 0 \end{bmatrix} \in \mathbb{R}^{h \times n}$, so that the last $h - m$ rows of $U_0$ are zero, and the last $h - n$ columns of $V_0$ are zero. Then, by the rank assumption, there exists $g_1 \in \mathrm{GL}_h$ such that $Ug_1 = U_0$, and there exists $g_2 \in \mathrm{GL}_h$ such that $g_2^{-1}V = V_0$. Without loss of generality, we can take $g_1$ and $g_2$ such that $\det(g_1) > 0$ and $\det(g_2) > 0$. Thus, both $g_1$ and $g_2$ belong to the component of the identity in $\mathrm{GL}_h$.

Next, consider the action of $G = \mathrm{GL}_h$ on full rank matrices in $\mathbb{R}^{m \times h}$ and $\mathbb{R}^{h \times n}$ individually. We have that $\mathrm{Stab}_G(U) = g_1 \mathrm{Stab}_G(U_0) g_1^{-1}$ and $\mathrm{Stab}_G(V) = g_2 \mathrm{Stab}_G(V_0) g_2^{-1}$. The stabilizer in

$\mathrm{GL}_h$ of the pair $(U, V) \in \mathbf{Param}^\circ$ can be written as:

$$\mathrm{Stab}_G(U, V) = \{g \in G \mid Ug^{-1} = U \text{ and } gV = V\} \tag{26}$$

$$= \mathrm{Stab}_G(U) \cap \mathrm{Stab}_G(V) \tag{27}$$

$$= (g_1 \mathrm{Stab}_G(U_0) g_1^{-1}) \cap (g_2 \mathrm{Stab}_G(V_0) g_2^{-1}) \tag{28}$$

Since $g_1$ and $g_2$ belong to the connected component of the identity, the dimension of $(g_1 \mathrm{Stab}_G(U_0) g_1^{-1}) \cap (g_2 \mathrm{Stab}_G(V_0) g_2^{-1})$ is equal to the dimension[12] of $\mathrm{Stab}_G(U_0) \cap \mathrm{Stab}_G(V_0)$. Hence we reduce the problem to computing the dimension of $\mathrm{Stab}_G(U_0) \cap \mathrm{Stab}_G(V_0)$.

To this end, observe that a matrix $g$ belongs to $\mathrm{Stab}_G(U_0)$ (resp. $\mathrm{Stab}_G(V_0)$) if and only if is of the form:

$$g = \begin{bmatrix} \mathrm{id}_m & 0 \\ * & * \end{bmatrix} \qquad (\text{resp.} \quad g = \begin{bmatrix} \mathrm{id}_n & * \\ 0 & * \end{bmatrix})$$

where the lower left $* \in \mathbb{R}^{(h-m) \times m}$ and the lower right $* \in \mathrm{GL}_{h-m}$ (resp. upper right $* \in \mathbb{R}^{n \times (h-n)}$ and lower right $* \in \mathrm{GL}_{h-n}$) are arbitrary. If $m \geq n$, taking the intersection amounts to considering matrices of the form:

$$g = \begin{bmatrix} \mathrm{id}_n & 0 & 0 \\ 0 & \mathrm{id}_{m-n} & 0 \\ 0 & * & * \end{bmatrix}$$

where the rows and columns are divided according to the partition $h = n + (m - n) + (h - m)$. If $m \geq n$, taking the intersection amounts to considering matrices of the form:

$$g = \begin{bmatrix} \mathrm{id}_m & 0 & 0 \\ 0 & \mathrm{id}_{n-m} & * \\ 0 & 0 & * \end{bmatrix}$$

where the rows and columns are divided according to $h = m + (n - m) + (h - n)$. In both cases, the dimension of the intersection is $(h - n)(h - m) = h^2 - hn - hm + nm$. We obtain the dimension of the orbit as: $h^2 - (h - n)(h - m) = h(n + m) - nm$. $\qquad \square$

Recall that the symmetry group for homogenous activations is the coordinate-wise positive rescaling subgroup of $\mathrm{GL}_h$, consisting of diagonal matrices with positive entries along the diagonal. We denote this subgroup as $T_+(h)$. Similarly, the symmetry group for radial rescaling activation is the orthogonal group $O(h)$. For linear networks, the activation is the identity function, so the symmetry group is all of $\mathrm{GL}_h$.

**Corollary C.4.** *The orbit of a point in* $\mathbf{Param}^\circ$ *under the appropriate symmetry group is given by:*

| Type of activation | Symmetry group | Dimension of generic orbit |
|---|---|---|
| Linear | $\mathrm{GL}_h$ | $h^2 - \max(0, h - n) \max(0, h - m)$ |
| Homogeneous | $T_+(h)$ | $\min(h, \max(n, m))$ |
| Radial rescaling | $O(h)$ | $\begin{cases} \binom{h}{2} & \text{if } h \leq \max(n, m) \\ \binom{h}{2} - \binom{h - \max(m, n)}{2} & \text{otherwise} \end{cases}$ |

*Proof.* Adopt the notation of the proof of the above Proposition. The stabilizer in $T_+(h)$ of $(U_0, V_0)$ is the intersection of the stabilizer in $\mathrm{GL}_h$ of $(U_0, V_0)$ and $T_+(h)$. This intersection is easily seen to have dimension $\max(0, h - \max(n, m))$. Subtracting this from $\dim(T_+(h)) = h$, we obtain the result for the homogeneous case. For the orthogonal case, the stabilizer in $O(h)$ of $(U_0, V_0)$ is the intersection of the stabilizer in $\mathrm{GL}_h$ of $(U_0, V_0)$ and $O(h)$. This intersection has dimension $0$ if $h \leq \max(n, m)$ and $\binom{h - \max n, m}{2}$ otherwise. Subtracting from $\dim(O(h)) = \binom{h}{2}$, we obtain the result for the radial rescaling case. $\qquad \square$

---

[12]Explicitly, fix a continuous path $\gamma_i : [0, 1]$ such that $\gamma_i(0)$ is the identity in $G$ and $\gamma_i(1) = g_i$, for $i = 1, 2$. The dimension of $(\gamma_1(t) \mathrm{Stab}_G(U_0) \gamma_1(t)^{-1}) \cap (\gamma_2(t) \mathrm{Stab}_G(V_0) \gamma_2(t)^{-1})$ is constant along this path.

### C.7 CONSERVED QUANTITIES

We now turn our attention to gradient flow and conserved quantities. In this section, we give a formal definition of a conserved quantity. Let $\mathcal{V} = \mathbb{R}^d$ be the standard vector space of dimension $d$. Suppose $L : \mathcal{V} \to \mathbb{R}$ is a differentiable function. Let $\mathrm{Flow}_t : \mathcal{V} \to \mathcal{V}$ be the flow for time $t$ along the reverse gradient vector field, so that:

$$\frac{d}{dt}\bigg|_0 [t \mapsto \mathrm{Flow}_t(v)] = -\nabla_v \mathcal{L}$$

Note that $\mathrm{Flow}_0$ is the identity on $\mathcal{V}$, and, for any $s, t$, the composition of $\mathrm{Flow}_s$ and $\mathrm{Flow}_t$ is $\mathrm{Flow}_{s+t}$. We will write $v(t)$ for $\mathrm{Flow}_t(v)$, so that $\dot{v}(s) = -\nabla_{v(s)}\mathcal{L}$. A *conserved quantity* is a function $Q : \mathcal{V} \to \mathbb{R}$ that satisfies either of the following equivalent conditions:

1. For any $t$, we have $Q(v(t)) = Q(v)$.

2. Let $\dot{Q}(v) = \frac{d}{dt}\big|_0[t \mapsto Q(v(t))]$ be the derivative of $Q$ along the flow. Then $\dot{Q} \equiv 0$.

3. The gradients of $Q$ and $\mathcal{L}$ are point-wise orthogonal, that is, $\langle \nabla_v Q, \nabla_v \mathcal{L} \rangle = 0$ for all $v \in \mathcal{V}$.

The equivalence of (1) and (2) are immediate. To show the equivalence of the third and second statements, let $v \in \mathcal{V}$ and compute:

$$\langle \nabla_v Q, \nabla_v \mathcal{L} \rangle = dQ_{v(0)} \circ \nabla_v \mathcal{L} = dQ_{v(0)} \circ \frac{d}{dt}\bigg|_0 v(t) = \frac{d}{dt}\left(Q \circ v(t)\right),$$

where we use the definition of the flow in the second equality, and the chain rule in the third.

We note that, if $f : \mathbb{R} \to \mathbb{R}$ is any function, and $Q$ is a conserved quantity, the $f \circ Q$ is also a conserved quantity. Additionally, any linear combination of conserved quantities is again a conserved quantity. Let $\mathrm{Conserv}(\mathcal{V}, L)$ denote the vector space of conserved quantities for the gradient flow of $L : \mathcal{V} \to \mathbb{R}$. For any $v \in \mathcal{V}$, there a map:

$$\nabla_v : \mathrm{Conserv}(\mathcal{V}, L) \to T_v \mathcal{V} = \mathbb{R}^d, \qquad Q \mapsto \nabla_v Q$$

taking a conserved quantity to the value of its gradient at $v$. By the above discussion, the map $\nabla_v$ is valued in the kernel of the differential $dL_v$.

### C.8 CONSERVED QUANTITIES FROM A GROUP ACTION

Let $G$ be a subgroup of the general linear group $\mathrm{GL}_d(\mathbb{R})$. Thus, there is a linear action of $G$ on $\mathcal{V} = \mathbb{R}^d$. Suppose the function $L$ is invariant for the action of $G$, that is,

$$L(g \cdot v) = L(v) \qquad \forall v \in \mathcal{V} \quad \forall g \in G.$$

Let $\mathfrak{g} = \mathrm{Lie}(G)$ be the Lie algebra of $G$, which is a Lie subalgebra of $\mathfrak{gl}_d = \mathbb{R}^{d \times d}$. The *infinitesimal action* of $\mathfrak{g}$ on $\mathcal{V}$ is given by $\mathfrak{g} \times V \to V$, taking $(M, v)$ to $Mv$.

**Proposition C.5.** *Let $L : \mathcal{V} \to \mathbb{R}$ be a $G$-invariant function, and let $M \in \mathfrak{g}$.*

1. *For any $v \in \mathcal{V}$, the gradient of $L$ and the infinitesimal action of $M$ are orthogonal:*

$$\langle \nabla_v L, Mv \rangle = 0.$$

2. *Suppose $\gamma : (a, b) \to \mathcal{V}$ is a gradient flow curve for $L$. Then:*

$$(\dot{\gamma}(t))^T M \gamma(t) = 0 \qquad \forall t \in (a, b)$$

3. *Suppose $M^T$ belongs to $\mathfrak{g}$. Then the function*

$$Q_M : \mathcal{V} \to \mathbb{R}, \qquad v \mapsto v^T M v$$

*is a conserved quantity for the gradient flow of $L$.*

*Proof.* For the first claim, observe that the invariance of $L$ implies that the left diagram commutes:

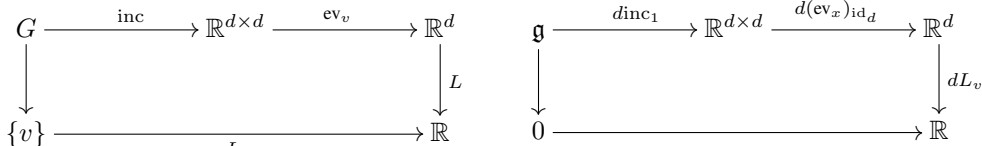

where $\mathrm{inc} : G \to \mathbb{R}^{d \times d}$ is the inclusion (which passes through the inclusion of $G$ in to $\mathrm{GL}_d(\mathbb{R})$), $\mathrm{ev}_v : \mathbb{R}^{d \times d} \to \mathbb{R}^d$ be the evaluation map at $v$, and the left vertical map is the constant map at $\{v\}$. Indeed, the clockwise composition is $g \mapsto L(gv)$, which is equal to the constant map at $g \mapsto L(v)$. The chain rule implies that taking Jacobians at the identity $1$ of $G$ results in the commutative diagram on the right. where $\mathfrak{g}$ is the Lie algebra of $G$, identified with the tangent space of $G$ at the identity; the tangent space of the vector space $\mathbb{R}^d$ at $v$ is canonically identified with $\mathbb{R}^d$; and the the zero appears because the tangent space of a single point is zero. The derivative of the inclusion map is the inclusion $\mathfrak{g} \hookrightarrow \mathfrak{gl}_d = \mathbb{R}^{d \times d}$, while the derivative the evaluation map is itself as it is a linear map. Hence, for $M \in \mathfrak{g}$, we have:

$$0 = dL_v \circ d(\mathrm{ev}_v)_{\mathrm{id}_d} \circ \mathrm{dinc}_1(M) = dL_v \circ \mathrm{ev}_v(M) = dL_v(Mv) = \langle \nabla_v L, Mv \rangle.$$

The first claim follows. The second claim is consequence of the first claim, together with the definition of a gradient flow curve. For the third claim, we take the derivative of the composition of $Q_M$ with a gradient flow curve $\gamma$:

$$\begin{aligned}
\frac{d}{dt}(Q_M \circ \gamma) &= (\dot{\gamma}(t))^T (M + M^T) \gamma(t) \\
&= \langle \dot{\gamma}(t), (M + M^T)\gamma(t) \rangle \\
&= -\langle \nabla_{\gamma(t)} L, (M + M^T)\gamma(t) \rangle \\
&= -\langle \nabla_{\gamma(t)} L, M\gamma(t) \rangle - \langle \nabla_{\gamma(t)} L, M^T \gamma(t) \rangle
\end{aligned}$$

Both terms in the last expression are constantly zero by the second claim. Hence $Q_M$ is constant on any gradient flow curve, and so it is a conserved quantity. $\qquad\square$

We summarize some of the results and constructions of this section diagrammatically. Let $\mathfrak{g}^{\mathrm{sym}}$ denote the vector space of symmetric matrices in $\mathfrak{g}$ (this is not a Lie subalgebra in general). Observe that $\mathfrak{g} \cap \mathfrak{g}^T$ is the set of all $M \in \mathfrak{g}$ such that $M^T \in \mathfrak{g}$. Let $\mathrm{Infin}_G(\mathcal{V}, L)$ denote the vector space of infinitesimal-action conserved quantities for the gradient flow of the $G$-invariant function $L : \mathcal{V} \to \mathbb{R}$. We have:

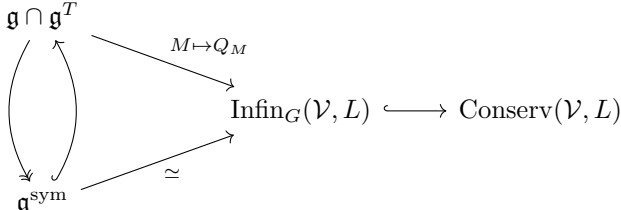

where the map $\mathfrak{g} \cap \mathfrak{g}^T \to \mathfrak{g}^{\mathrm{sym}}$ takes $M$ to its symmetric part $\frac{1}{2}(M + M^T)$, while the map $\mathfrak{g}^{\mathrm{sym}} \hookrightarrow \mathfrak{g} \cap \mathfrak{g}^T$ is the natural inclusion. We note that $\mathfrak{g} \cap \mathfrak{g}^T$ is the Lie algebra of the group $G \cap G^T$, while $\mathfrak{g}^{\mathrm{sym}}$ is in general not a Lie algebra. By definition, the vector space $\mathrm{Infin}_G(\mathcal{V}, L)$ is the image of the map $M \to Q_M$ defined on $\mathfrak{g} \cap \mathfrak{g}^T$. It is straightforward to verify the following result:

**Corollary C.6.** *The map $M \to Q_M$ establishes an isomorphism of vector spaces:* $\mathfrak{g}^{\mathrm{sym}} \xrightarrow{\sim} \mathrm{Infin}_G(\mathcal{V}, L)$.

As discussed in Section 3.2, applying a symmetry $g$ to a minimum $\boldsymbol{\theta}^*$ of $\mathcal{L}$ yields another minimum $g \cdot \boldsymbol{\theta}^*$. Using flattened $\boldsymbol{\theta} \in \mathbb{R}^d$ it is easy to show that acting with $g$ changes some $Q_M(\boldsymbol{\theta}) = \boldsymbol{\theta}^T M \boldsymbol{\theta}$. Let $g = \exp_{M'}(t) \approx I + tM'$, with $M' \in \mathfrak{g}$ and $0 < \eta \ll 1$ be a $g \in G$ close to identity. We have $Q_M(g \cdot \boldsymbol{\theta}) = Q_M + t\boldsymbol{\theta}^T \left( M'^T M + M M' \right) \boldsymbol{\theta} + O(\eta^2)$. Thus, whenever $M'^T M + M M' \neq 0$, applying $g$ changes the value of $Q_M$. Therefore, $Q_M$ can be used to parameterize the flat minima. However, for anti-symmetric $M$, we could not find nonzero $Q$ explicitly.

### C.8.1 ANTI-SYMMETRIC CASE

Suppose $M \in \mathfrak{g}$ is anti-symmetric, so $M = -M^T$. Let $\gamma : (a, b) \to \mathbb{R}^d$ be a gradient flow curve. Write $\gamma$ in coordinates as $\gamma = (\gamma_1, \ldots, \gamma_d)$. Proposition C.5 implies that $(\dot{\gamma}(t))^T M\gamma = 0$. Hence we have:

$$0 = \sum_{i<j} m_{ij} \left( \dot{\gamma}_i \gamma_j - \gamma_i \dot{\gamma}_j \right) = \sum_{i<j} m_{ij} \gamma_i^2 \left( \frac{\gamma_j}{\gamma_i} \right)' = \sum_{i<j} m_{ij} r_{ij}^2 \dot{\theta}_{ij}$$

where $r_{ij} = r_{ij}(t)$ is equal to $\sqrt{\gamma_i(t)^2 - \gamma_j(t)^2}$ and $\theta_{ij} = \theta_{ij}(t)$ is the angle between the $i$-th coordinate axis and the ray from the origin to the projection of $\gamma(t)$ to the $(i, j)$-plane. One verifies the last equality using the definition of $\theta_{ij}$ in terms of the arctangent of the quotient $\gamma_j/\gamma_i$. We see that $(r_{ij}(t), \theta_{ij}(t))$ are the polar coordinates for the point $(\gamma_i(t), \gamma_j(t)) \in \mathbb{R}^2$.

**Case $d = 2$.** Then $M = \begin{bmatrix} 0 & a \\ -a & 0 \end{bmatrix}$ for some nonzero $a \in \mathbb{R}$, and so:

$$0 = \dot{\gamma}^T M\gamma = a\dot{\gamma}_1(t)\gamma_2(t) - a\gamma_1(t)\dot{\gamma}_2(t) = ar(t)^2 \dot{\theta}(t)$$

where $r(t)$ and $\theta(t)$ are polar coordinates. Setting the final expression equal to zero, we obtain that $\theta(t)$ is constant along any flow line $\gamma(t)$ that begins away from the origin.

**Case $d = 3$.** Then $M = \begin{bmatrix} 0 & a & b \\ -a & 0 & c \\ -b & -c & 0 \end{bmatrix}$ for some $a, b, c \in \mathbb{R}$, and so:

$$0 = \dot{\gamma}^T M\gamma = ar_{12}^2 \dot{\theta}_{12} + br_{13}^2 \dot{\theta}_{13} + cr_{23}^2 \dot{\theta}_{23}$$

### C.9 EXAMPLES OF CONSERVED QUANTITIES FOR NEURAL NETWORKS

We now compute conserved quantities for gradient flow on neural network parameter spaces in the case of linear, homogeneous, and radial networks. In each case, we state results first in the for a general multi-layer network, and then for the running example of a two-layer network. Throughout, $\odot$ denotes the Hadamard product of matrices, defined by entrywise multiplication. We also set $\tau(M)$ to be the sum of all entries in a matrix $M$. We note that, for square matrices $M$ and $N$ of the same size, $\tau(M \odot N) = \text{Tr}(M^T N)$, which is the same as the inner product of the flattened versions of $M$ and $N$.

The notation for the running example of a two-layer network is as follows. We set the input and output dimensions both equal to one, hidden dimension equal to two, and no bias vectors. The hidden layer activation is $\sigma : \mathbb{R}^2 \to \mathbb{R}^2$. The parameter space is $\mathbf{Param} = \mathbb{R}^{2 \times 1} \times \mathbb{R}^{1 \times 2}$, with elements written as a pair of matrices: $(U, V) = \left( \begin{bmatrix} u_1 & u_2 \end{bmatrix}, \begin{bmatrix} v_1 \\ v_2 \end{bmatrix} \right)$. The hidden symmetry group is $\text{GL}_2$, with action given by:

$$\text{GL}_2(\mathbb{R}) \times \mathbf{Param} \to \mathbf{Param}, \qquad (g, U, V) \mapsto g \cdot (U, V) = \left( Ug^{-1}, gV \right).$$

The Lie algebra $\mathfrak{gl}_2$ of $\text{GL}_2(\mathbb{R})$ consists of all two-by-two matrices.

### C.9.1 CONSERVED QUANTITIES FOR LINEAR NETWORKS

Suppose a neural network with $L$ layers has the identity activation $\sigma_i = \text{id}_{n_i}$ in each layer, so that the resulting network is linear. Then it is straightforward to verify that the networks with parameters $\mathbf{g} \cdot (\mathbf{W}, \mathbf{b})$ and $(\mathbf{W}, \mathbf{b})$ have the same feedforward function. Consequently, the loss is invariant for the group action: its value the original and transformed parameters is the same for any choice of training data. (As we will see below, for more sophisticated activations, one needs to restrict to a subgroup of the hidden symmetry group to achieve such invariance.)

Suppose $\mathbf{M} \in \mathfrak{gl}_{\mathbf{n}^{\text{hidden}}}$ is such that $M_i \in \mathfrak{gl}_{n_i}$ is symmetric for each $i$. The conserved quantity implied by Proposition C.5 is:

$$Q_{\mathbf{M}}(\mathbf{W}, \mathbf{b}) = \sum_{i=1}^{L-1} \left( \tau(W_i \odot M_i W_i) + \tau(b_i \odot M_i b_i) - \tau(W_{i+1} \odot W_{i+1} M_i) \right)$$

$$= \sum_{i=1}^{L-1} \text{Tr} \left( \left( W_i W_i^T + b_i b_i^T - W_{i+1}^T W_{i+1} \right) M_i \right)$$

We examine these conserved quantities in the following convenient basis for the space of symmetric matrices in $\mathfrak{gl}_{\mathbf{n}^{\text{hidden}}}$. For $j = 1, \ldots, L$ and $\{k, \ell\} \subseteq \{1, \ldots, n_j\}$, set:

$$E_{\{k,\ell\}}^{(j)} := \begin{cases} E_{kk}^{(n_j)} & \text{if } k = \ell \\ \frac{1}{2}\left( E_{k\ell}^{(n_j)} + E_{\ell k}^{(n_j)} \right) & \text{if } k \neq \ell \end{cases}$$

where $E_{k\ell}^{(n_j)}$ is the elementary $n_j \times n_j$ matrix with the entry in the $k$-th row and $\ell$-th column equal to one, and all other entries equal to zero. Then one computes:

$$Q_{E_{\{k,\ell\}}^{(j)}}(\mathbf{W}, \mathbf{b}) = b_k^{(j)} b_\ell^{(j)} + \sum_{t=1}^{n_{j-1}} w_{kt}^{(j)} w_{\ell t}^{(j)} - \sum_{r=1}^{n_{j+1}} w_{rk}^{(j+1)} w_{r\ell}^{(j+1)}$$

In other words, we take the sum of the following three terms: the product of the $k$-th and $\ell$-th entries of the bias vector $b_j$, the dot product of the $k$-th and $\ell$-th rows of $W_j$, and the dot product of the $k$-th and $\ell$-th columns of $W_{j+1}$. In particular, we see that every entry of the matrix

$$\mu_i(\mathbf{W}, \mathbf{b}) := W_i W_i^T + b_i b_i^T - W_{i+1}^T W_{i+1} \in \mathfrak{gl}_{n_i}$$

is a conserved quantity valued in $\mathfrak{gl}_{n_i}$ rather than in $\mathbb{R}$. Additionally, we have a *moment map*:

$$\mathbf{Q} : \mathbf{Param} \to \mathfrak{gl}_{\mathbf{n}^{\text{hidden}}}^*, \qquad (\mathbf{W}, \mathbf{b}) \mapsto \left[ \mathbf{M} \mapsto \sum_{i=1}^{L-1} \langle \mu_i(\mathbf{W}, \mathbf{b}), M_i \rangle \right]$$

### C.9.2 CONSERVED QUANTITIES FOR LINEAR NETWORKS: TWO-LAYER CASE

In the two layer case of a linear network, we have that that the single hidden activation is the identity: $\sigma = \text{id}_2 : \mathbb{R}^2 \to \mathbb{R}^2$. The hidden symmetry group is $\text{GL}_2$ with Lie algebra all of $\mathfrak{gl}_2$. The space of symmetric matrices in $\mathfrak{gl}_2$ is spanned by the matrices:

$$E_{11} = \begin{bmatrix} 1 & 0 \\ 0 & 0 \end{bmatrix}, \qquad E_{22} \begin{bmatrix} 0 & 0 \\ 0 & 1 \end{bmatrix}, \qquad \text{and} \qquad E_{(1,2)} = \begin{bmatrix} 0 & 1 \\ 1 & 0 \end{bmatrix}.$$

The corresponding conserved quantities are:

$$Q_{E_{11}}(U, V) = v_1^2 - u_1^2 \qquad Q_{E_{22}}(U, V) = v_2^2 - u_2^2 \qquad Q_{E_{(1,2)}}(U, V) = v_1 v_2 - u_1 u_2$$

Thus, we obtain a three-dimensional space of conserved quantities. (Since $\text{GL}_2$ also contains the orthogonal group $O(2)$, Equation 29 below holds along any gradient flow curve.)

### C.9.3 CONSERVED QUANTITIES FOR ReLU NETWORKS

The pointwise ReLU activation commutes with positive rescaling, so we consider the subgroup of the hidden symmetry group consisting of tuples of diagonal matrices with positive diagonal entries, that is:

$$G = \{ \mathbf{g} \in \text{GL}_{\mathbf{n}^{\text{hidden}}} \mid g_i = \text{Diag}(s_1, \ldots, s_{n_i}), \ s_j > 0 \}$$

This subgroup, also known as the positive coordinate-wise rescaling subgroup, is isomorphic to the product $(\mathbb{R}_{>0})^{\sum_{i=1}^{L-1} n_i}$. Its Lie algebra is spanned by the elements $E_{kk}^{(j)}$ defined above, for $j = 1, \ldots, L-1$ and $k = 1, \ldots, n_j$. The conserved quantity implied by Proposition C.5 is:

$$Q_{E_{kk}^{(j)}}(\mathbf{W}, \mathbf{b}) = \left( b_k^{(j)} \right)^2 + \sum_{t=1}^{n_{j-1}} \left( w_{kt}^{(j)} \right)^2 - \sum_{r=1}^{n_{j+1}} \left( w_{rk}^{(j+1)} \right)^2$$

In other words, we take the sum of the following three terms: the square of the $k$-th entry of the bias vector $b_j$, the norm of the $k$-th row of $W_j$, and the norm of the $k$-th column of $W_{j+1}$.

### C.9.4 CONSERVED QUANTITIES FOR RELU NETWORKS: TWO-LAYER CASE

In the two-layer case, the positive rescaling group is:

$$G = \left\{ \begin{bmatrix} g_1 & 0 \\ 0 & g_2 \end{bmatrix} \in \mathrm{GL}_2(\mathbb{R}) \mid g_1 \text{ and } g_2 \text{ are positive.} \right\}$$

The Lie algebra of $G$ is the two-dimensional space of diagonal matrices in $\mathfrak{gl}_2$ (with not necessarily positive diagonal entries). In other words, $\mathfrak{g}$ is spanned by the matrices $E_{11} = \begin{bmatrix} 1 & 0 \\ 0 & 0 \end{bmatrix}$ and $E_{22} = \begin{bmatrix} 0 & 0 \\ 0 & 1 \end{bmatrix}$. One computes the conserved quantities corresponding to these elements as:

$$Q_{E_{11}}(U, V) = v_1^2 - u_1^2 \qquad Q_{E_{22}}(U, V) = v_2^2 - u_2^2$$

Hence there is a two-dimensional space of conserved quantities coming from the infinitesimal action.

### C.9.5 CONSERVED ANGULAR MOMENTUM FOR RADIAL RESCALING NETWORKS

Suppose each $\sigma_i$ is a radial rescaling activation $\sigma_i(z) = \lambda_i(|z|)z$, where $\lambda_i : \mathbb{R} \to \mathbb{R}$ is the rescaling factor. Each such activation commutes with orthogonal transformations, so we consider the subgroup of the hidden symmetry group consisting of tuples of orthogonal matrices:

$$G = \{\mathbf{g} \in \mathrm{GL}_{\mathbf{n}^{\text{hidden}}} \mid g_i g_i^T = \mathrm{id}_{n_i} \text{ for all } i\}$$

The Lie algebra of this subgroup consists only of anti-symmetric matrices, and so there are no infinitesimal-action conserved quantities. However, given an anti-symmetric matrix $M_i \in \mathfrak{gl}_{n_i}$ for each $i$, any gradient flow curve satisfies the following differential equation (encoding conservation of angular momentum):

$$\sum_{i=1}^{L-1} \left( \tau(\dot{W}_i \odot M_i W_i) + \tau(\dot{b}_i \odot M_i b_i) - \tau(\dot{W}_{i+1} \odot W_{i+1} M_i) \right) = 0$$

(cf. Section C.8.1). An equivalent way to write this equation is:

$$\sum_{i=1}^{L-1} \mathrm{Tr} \left( \left( W_i \dot{W}_i^T + b_i \dot{b}_i^T + W_{i+1}^T \dot{W}_{i+1} \right) M_i \right) = 0$$

Indeed, one uses the facts that $\tau(A \odot B) = \mathrm{Tr}(A^T B)$, $\mathrm{Tr}(A^T) = \mathrm{Tr}(A)$, and $\mathrm{Tr}(AB) = \mathrm{Tr}(BA)$, for any two matrices $A, B$ of the appropriate size in each case. Using a basis of anti-symmetric matrices, one can show that the matrix

$$\nu_i(\mathbf{W}, \mathbf{b}) := W_i \dot{W}_i^T - \dot{W}_i W_i^T + b_i \dot{b}_i^T - \dot{b}_i b_i^T + W_{i+1}^T \dot{W}_{i+1} - \dot{W}_{i+1}^T W_{i+1} \in \mathfrak{gl}_{n_i}$$

is equal to zero: $\nu_i(\mathbf{W}, \mathbf{b}) = 0$. Note that $\nu_i$ depends on taking derivatives with respect to the flow. In fact, $\nu_i$ is more properly formulated as a function on the tangent bundle $T(\mathbf{Param})$ of $\mathbf{Param}$, which is then evaluated on the gradient flow vector field. Similarly, we have a moment map $T(\mathbf{Param}) \to \mathfrak{gl}_{\mathbf{n}^{\text{hidden}}}^*$, and the gradient flow vector field is contained in the preimage of zero. We omit the details.

A basis for the space of anti-symmetric matrices in $\mathfrak{gl}_{\mathbf{n}^{\text{hidden}}}$ is given by:

$$E_{k<\ell}^{(j)} := E_{k\ell}^{(n_j)} - E_{\ell k}^{(n_j)}$$

where $j = 1, \ldots, L-1$, and $k, \ell \in \{1, \ldots, n_j\}$ satisfy $k < \ell$. The differential equation corresponding to $E_{k \leq \ell}^{(j)}$ is given by:

$$r_{b_j;k,\ell}^2 \dot{\theta}_{b_j;k,\ell} + \sum_{t=1}^{n_{j-1}} r_{W_j;ks,\ell s}^2 \dot{\theta}_{W_j;ks,\ell s} + \sum_{r=1}^{n_{j+1}} r_{W_{j+1};rk,r\ell}^2 \dot{\theta}_{W_{j+1};rk,r\ell} = 0$$

where $(r_{b_j;k\ell}, \theta_{b_j;k,\ell})$ are the polar coordinates of the image of $b_j$ under the projection $\mathbb{R}^{n_j} \to \mathbb{R}^2$ which selects only the $k$-th and $\ell$-th coordinates. Similarly, for any pair matrix entries we have a projection $\mathbb{R}^{n_j \times n_{j-1}} \to \mathbb{R}^2$ and can take the polar coordinates of the image of $W_j$ under this projection.

### C.9.6 Conserved angular momentum for radial rescaling networks: two-layer case

In the two-layer radial rescaling case, suppose the dimension vector is $(n, h, m)$, and that there are no bias vectors. For $U \in \mathbb{R}^{m \times h}$, $V \in \mathbb{R}^{h \times n}$ and $M \in \mathfrak{so}(\mathfrak{h})$, we have:

$$
\begin{aligned}
\left\langle \dot{\boldsymbol{\theta}}, M \cdot \boldsymbol{\theta} \right\rangle &= \left\langle (\dot{U}, \dot{V}), (-UM, MV) \right\rangle = -\mathrm{Tr}(\dot{U}^T U M) + \mathrm{Tr}(\dot{V}^T M V) \\
&= \mathrm{Tr}(V \dot{V}^T M) - \mathrm{Tr}(M^T U^T \dot{U}) = \mathrm{Tr}(V \dot{V}^T M) + \mathrm{Tr}(M U^T \dot{U}) \\
&= \mathrm{Tr}(V \dot{V}^T M) + \mathrm{Tr}(U^T \dot{U} M) = \mathrm{Tr}\left[ \left( V \dot{V}^T + U^T \dot{U} \right) M \right]
\end{aligned}
$$

Hence we obtain the differential equation:

$$
\mathrm{Tr}\left[ \left( V \dot{V}^T + U^T \dot{U} \right) M \right] = 0
$$

In the case where $(n, h, m) = (1, 2, 1)$, we have the two by two orthogonal group:

$$
G = O(2) = \left\{ g \in \mathrm{GL}_2(\mathbb{R}) \mid g^T g = \mathrm{id} \right\}
$$

The Lie algebra of $G$ consists of anti-symmetric matrices in $\mathfrak{gl}_2$, and contains no non-zero symmetric matrices. Hence, we do not obtain any conserved quantities from the infinitesimal action in this case. However, using the element $\begin{bmatrix} 0 & 1 \\ -1 & 0 \end{bmatrix} \in \mathfrak{g}$, we obtain that the following differential equation holds along any gradient flow curve:

$$
r_U^2 \dot{\theta}_U + r_V^2 \dot{\theta}_V = 0 \tag{29}
$$

where $(r_U, \theta_U)$ are the polar coordinates of $(u_1, u_2) \in \mathbb{R}^2$, and similarly for $(r_V, \theta_V)$. Note that the left-hand side of Equation 29 is a function of $t$; so if $\gamma : (a, b) \to \mathcal{V}$ is a gradient flow curve, then a more precise version of the equation is $\left( r_U^2 \circ \gamma \right)(t) \cdot (\theta_U \circ \gamma)'(t) + \left( r_V^2 \circ \gamma \right)(t) \cdot (\theta_V \circ \gamma)'(t) = 0$ for all $t$.

### C.10 Jacobians: special cases

We conclude this appendix with a side remark on special cases of the Jacobian formalism.

**Manifolds.** Suppose $M$ and $N$ are smooth manifolds, and suppose $F : M \to N$ is a smooth map. The differential of $F$ at $m \in M$ is a linear map between the tangent spaces:

$$
dF_m : T_m M \to T_{F(m)} N
$$

The map $dF_m$ is computed in local coordinate charts as the Jacobian of partial derivatives. If $G : N \to L$ is another smooth map, then the chain rule becomes $d(G \circ F)_m = dG_{F(m)} \circ dF_m$, for any $m \in M$.

**Matrix case.** Suppose $L : \mathbb{R}^{m \times n} \to \mathbb{R}$ is a differentiable function. In this case, we regard the Jacobian at $W \in \mathbb{R}^{m \times n}$ as an $n \times m$ matrix:

$$
dL_W = \begin{bmatrix}
\left. \frac{\partial L}{\partial w_{11}} \right|_W & \left. \frac{\partial L}{\partial w_{21}} \right|_W & \cdots & \left. \frac{\partial L}{\partial w_{m1}} \right|_W \\
\left. \frac{\partial L}{\partial w_{12}} \right|_W & \left. \frac{\partial L}{\partial w_{22}} \right|_W & \cdots & \left. \frac{\partial L}{\partial w_{m2}} \right|_W \\
\vdots & \vdots & \ddots & \vdots \\
\left. \frac{\partial L}{\partial w_{1n}} \right|_W & \left. \frac{\partial L}{\partial w_{2n}} \right|_W & \cdots & \left. \frac{\partial L}{\partial w_{mn}} \right|_W
\end{bmatrix} \in \mathbb{R}^{m \times n}
$$

where $w_{ij}$ are the matrix coordinates. If $F : \mathbb{R} \to \mathbb{R}^{m \times n}$ is a differentiable function, we regard its Jacobian at $s \in \mathbb{R}$ as a $m \times n$ matrix:

$$
dF_t = \begin{bmatrix}
\frac{dF_{11}}{dt}\Big|_s & \frac{dF_{12}}{dt}\Big|_s & \cdots & \frac{dF_{1n}}{dt}\Big|_s \\
\frac{dF_{21}}{dt}\Big|_s & \frac{dF_{22}}{dt}\Big|_s & \cdots & \frac{dF_{2n}}{dt}\Big|_s \\
\vdots & \vdots & \ddots & \vdots \\
\frac{dF_{m1}}{dt}\Big|_s & \frac{dF_{m2}}{dt}\Big|_s & \cdots & \frac{dF_{mn}}{dt}\Big|_s
\end{bmatrix} \in \mathbb{R}^{n \times m}
$$

where $F_{ij} : \mathbb{R} \to \mathbb{R}$ are the coordinates of $F$. Then the chain rule becomes:

$$
\frac{d}{dt}\Big|_s (L \circ F) = \sum_{i=1}^{m} \sum_{j=1}^{n} \left( \frac{\partial L}{\partial w_{ij}}\Big|_{F(s)} \frac{dF_{ij}}{dt}\Big|_s \right) = \mathrm{Tr}(dL_{F(s)} \cdot dF_s).
$$

In other words, the derivative of the composition $L \circ F$ at $s \in \mathbb{R}$ is the trace of the product of the matrices $dL_{F(s)} \in \mathbb{R}^{n \times m}$ and $dF_s \in \mathbb{R}^{m \times n}$.

## D    NEURAL NETWORKS: NON-LINEAR ACTIONS GROUP ACTIONS

In this section, we consider a non-linear action of the hidden symmetry group on the parameter space. This action has the advantage that exists for a wider variety of activation functions (such as the usual sigmoid, which has no linear equivariance properties), and that it is defined for the full general linear group. However, in constrast to the linear action, the non-linear action is data-dependent: the transformation of the weights and biases depends on the input data.

### D.1    ROTATIONS

We first define certain orthogonal matrices.

**Definition D.1.** *For any tuple of real numbers $\boldsymbol{\beta} = (\beta_1, \ldots, \beta_n)$, define an $(n+1) \times (n+1)$ matrix $R(\boldsymbol{\beta})$ as follows:*

$$
(R(\boldsymbol{\beta}))_{ij} = \begin{cases}
\cos(\beta_{j-1}) \left( \prod_{k=j}^{i-1} \sin(\beta_k) \right) \cos(\beta_i) & \text{if } j \leq i \\
-\sin(\beta_i) & \text{if } j = i+1 \\
0 & \text{if } j > i+1
\end{cases}
$$

*where, by convention, we set $\beta_0 = \beta_{n+1} = 0$.*

For example, when $n = 1, 2$, we have:

$$
R(\beta) = \begin{bmatrix} \cos(\beta) & -\sin(\beta) \\ \sin(\beta) & \cos(\beta) \end{bmatrix} \qquad R(\beta_1, \beta_2) = \begin{bmatrix} \cos(\beta_1) & -\sin(\beta_1) & 0 \\ \sin(\beta_1)\cos(\beta_2) & \cos(\beta_1)\cos(\beta_2) & -\sin(\beta_2) \\ \sin(\beta_1)\sin(\beta_2) & \cos(\beta_1)\sin(\beta_2) & \cos(\beta_2) \end{bmatrix}
$$

**Lemma D.2.** *For any tuple of real numbers $\boldsymbol{\beta} = (\beta_1, \ldots, \beta_n)$, we have:*

1. $\sum_{i=1}^{n} \cos^2(\beta_i) \prod_{k=1}^{i-1} \sin^2(\beta_k) + \prod_{k=1}^{n} \sin^2(\beta_k) = 1$

2. *The matrix $R(\boldsymbol{\beta})$ is orthogonal.*

*Sketch of proof.* The first identity follows from a straightforward induction argument, while the proof of the second claim amounts to a computation that invokes the identity of the first claim.  □

**Proposition D.3.** *There is a continuous map $R : \mathbb{R}^h \setminus \{0\} \to \mathrm{GL}_h$, written $z \mapsto R_z$, such that:*

1. *For any $z \in \mathbb{R}^h \setminus \{0\}$, the first column of $R_z$ is $z$. Hence $R_z e_1 = z$, where $e_1 = (1, 0, \ldots, 0)$ is the first basis vector.*

2. *The operator norm of $R_z$ is $\|R_z\| = |z|$.*

3. *If $|z| = 1$, then $R_z$ is an orthogonal matrix.*

*Proof.* Let $z \in \mathbb{R}^h \setminus \{0\}$, and let $(r, \alpha_1, \ldots, \alpha_{n-1})$ be the (reverse) $h$-spherical coordinates of $z$. Hence, $r = |z|$ is the norm of $z$ and the $i$-th coordinate of $z$ is $z_i = r \left( \prod_{k=1}^{i-1} \sin(\alpha_k) \right) \cos(\alpha_i)$, where $\alpha_h = 0$ by convention. Now set $R_z = |z| R(\alpha_1, \ldots, \alpha_{h-1})$. Using Lemma D.2, one concludes that $R_z$ is invertible with inverse $\frac{1}{|z|^2} R_z^T$, so that $R_z$ has operator norm is $|z|$ and $R_z$ is orthogonal if $|z| = 1$. It is also clear that the first column of $R_z$ is equal to $z$. $\square$

We note the the matrix in D.1 has a form similar, but not identical, to the Jacobian matrix for the transformation to $n$-spherical coordinates. Euler angles provide another way to construct a map $\mathbb{R}^h \setminus \{0\} \to \mathrm{GL}_h$ the same properties as in Proposition D.3.

## D.2 NON-LINEAR ACTION: TWO-LAYER CASE

Consider a two-layer network with dimension vector $(m, h, n)$, no bias vectors, and no output activation. The parameter space is $\mathbf{Param} = \mathbb{R}^{m \times h} \times \mathbb{R}^{h \times n}$. Define the non-degenerate locus as:

$$(\mathbf{Param} \times \mathbb{R}^n)^\circ = \{(U, V, x) \in \mathbb{R}^{m \times h} \times \mathbb{R}^{h \times n} \times \mathbb{R}^n \mid Vx \neq 0\}$$

Let $\tilde{F} : \mathbf{Param} \times \mathbb{R}^n \to \mathbb{R}^m$ be the extended feedforward function, taking $(U, V, x)$ to $F_{(U,V)}(x) = U\sigma(Vx)$. We now state and prove a more general version of Theorem 4.1.

**Theorem D.4.** *Suppose $\sigma(z) \neq 0$ for all nonzero $z \in \mathbb{R}^h \setminus \{0\}$.*

1. *There is an action:*

$$\mathrm{GL}_h \times (\mathbf{Param} \times \mathbb{R}^n)^\circ \to (\mathbf{Param} \times \mathbb{R}^n)^\circ$$
$$g \cdot (U, V, x) = (U R_{\sigma(Vx)} R_{\sigma(gVx)}^{-1} \, , \, gV \, , \, x)$$

2. *Suppose, in addition, that $\sigma(0) \neq 0$, so that $\sigma$ is nonzero on all of $\mathbb{R}^h$. There is an action:*

$$\mathrm{GL}_h \times (\mathbf{Param} \times \mathbb{R}^n) \to (\mathbf{Param} \times \mathbb{R}^n)$$
$$g \cdot (U, V, x) = (U R_{\sigma(Vx)} R_{\sigma(gVx)}^{-1} \, , \, gV \, , \, x)$$

*In both cases, the extended feedforward function is invariant for this action, that is: $\tilde{F}(g \cdot (U, V, x)) = \tilde{F}(U, V, x)$.*

*Proof.* We first verify that the action is well-defined. In the second case, $\sigma(gVx) \neq 0$ for all $(U, V, x)$ and hence $R_{\sigma(gVx)}$ is defined and invertible for any $g \in \mathrm{GL}_h$. For the first case, let $(U, V, x)$ be in the non-degenerate locus. The non-degeneracy condition $Vx \neq 0$ guarantees that $gVx \neq 0$ for all $g \in \mathrm{GL}_h$. The hypothesis on $\sigma$ in turn implies that $R_{\sigma(gVx)}$ is defined and invertible for any $g \in \mathrm{GL}_h$. Hence the action is well-defined in both cases.

To check the unit axiom, observe that, when $g = \mathrm{id}_h$ is the identity of $\mathrm{GL}_h$, we have $R_{\sigma(Vx)} R_{\sigma(gVx)}^{-1} = R_{\sigma(Vx)} R_{\sigma(Vx)}^{-1} = \mathrm{id}_h$ and $gV = V$. It follows that $\mathrm{id} \cdot (U, V, x) = (U, V, x)$. To check the multiplication axiom, let $g_1, g_2 \in \mathrm{GL}_h$. Then:

$$\left( R_{\sigma(g_1 g_2 v)} R_{\sigma(g_2 v)}^{-1} \right) \left( R_{\sigma(g_2 v)} R_{\sigma(v)}^{-1} \right) = R_{\sigma(g_1 g_2 v)} R_{\sigma(v)}^{-1}.$$

It follows that $g_1 \cdot (g_2 \cdot (U, V, x)) = (g_1 g_2) \cdot (U, V, x)$. For the last claim, we compute:

$$\tilde{F}(g \cdot (U, V, x)) = \tilde{F}(U R_{\sigma(Vx)} R_{\sigma(gVx)}^{-1}, gV, x) = U R_{\sigma(Vx)} R_{\sigma(gVx)}^{-1} \sigma(gVx)$$
$$= U R_{\sigma(Vx)} e_1 = U\sigma(Vx)$$

where the first equality follows from the definition of the action; the second from the extended feedforward function $\tilde{F}$; and the third and fourth follow from Proposition D.3. $\square$

From the proof, we see that a key property of the matrices $R_z$ is that:

$$R_{\sigma(gz)}R_{\sigma(z)}^{-1}\sigma(z) = \sigma(gz) \tag{30}$$

This can be interpreted as a data-dependent generalization of the equivariance condition appearing in Equation (4). We emphasize that a sufficient condition for the existence of such an action is that $\sigma(z)$ is nonzero for any nonzero $z \in \mathbb{R}^h$; this is the case for usual sigmoid, hyperbolic tangent, leaky ReLU, and many other activations.

Finally, we remark on a differential-geometric interpretation of the construction of this section. One can regard $\sigma$ as a section of the trivial bundle on $\mathbb{R}^h \setminus \{0\}$ with fiber $\mathbb{R}^h$. The map $z \mapsto R_{\sigma(gz)}R_{\sigma(z)}^{-1}$ defines a $\mathrm{GL}_h$-equivariant structure on this bundle such that $\sigma$ is an equivariant section. Indeed, the action of $\mathrm{GL}_h$ on the total space $(\mathbb{R}^h \setminus \{0\}) \times \mathbb{R}^h$ is given by $g \cdot (z, a) = (gz, R_{\sigma(gz)}R_{\sigma(z)}^{-1}a)$, and the equivariance of $\sigma$ is precisely the condition $R_{\sigma(gz)}R_{\sigma(z)}^{-1}\sigma(z) = \sigma(gz)$.

### D.3 NON-LINEAR ACTION: MULTI-LAYER CASE

We adopt the notation of Section C.2. In particular, consider a neural network with $L$ layers and widths $\mathbf{n} = (n_0, n_1, \ldots, n_L)$. The parameter space is given by:

$$\mathbf{Param}(\mathbf{n}) = \mathbb{R}^{n_L \times n_{L-1}} \times \mathbb{R}^{n_{L-1} \times n_{L-2}} \times \cdots \times \mathbb{R}^{n_2 \times n_1} \times \mathbb{R}^{n_1 \times n_0} \times \mathbb{R}^{n_L} \times \mathbb{R}^{n_{L-1}} \times \cdots \times \mathbb{R}^{n_1}.$$

So for each layer $i$, we have a matrix $W_i \in \mathbb{R}^{n_i \times n_{i-1}}$ and vector $b_i \in \mathbb{R}^{n_i}$. We write $\boldsymbol{\theta} = (W_i, b_i)_{i=1}^L$ for a choice of parameters. Fix activations $\sigma_i : \mathbb{R}^{n_i} \to \mathbb{R}^{n_i}$ for each $i = 1, \ldots, L$. Let

$$F = F_{\boldsymbol{\theta}} : \mathbb{R}^{n_0} \to \mathbb{R}^{n_L}$$

be the feedforward function corresponding to parameters $\boldsymbol{\theta} = (\mathbf{W}, \mathbf{b}) \in \mathbf{Param}$ with activations $\sigma_i$. Taking the parameters into account, we form the extended feedforward function:

$$\widetilde{F} : \mathbf{Param} \times \mathbb{R}^{n_0} \to \mathbb{R}^{n_L}, \qquad \widetilde{F}(\boldsymbol{\theta}, x) = F_{\boldsymbol{\theta}}(x)$$

One can also define the extension of the partial feedforward function $\widetilde{F}_i : \mathbf{Param} \times \mathbb{R}^n \to \mathbb{R}^{n_i}$ as $(\boldsymbol{\theta}, x) \mapsto F_{\boldsymbol{\theta},i}(x)$. Furthermore, let $Z_i : \mathbf{Param} \times \mathbb{R}^n \to \mathbb{R}^{n_i}$ be the function defined recursively as:

$$Z_i(\boldsymbol{\theta}, x) = \begin{cases} x & \text{if } i = 0 \\ W_1 x + b_1 & \text{if } i = 1 \\ W_i \sigma_{i-1}(Z_{i-1}(\boldsymbol{\theta}, x)) + b_i & \text{for } i = 2, \ldots, L \end{cases}$$

We have $\tilde{F}_i = \sigma_i \circ Z_i$ for $i = 1, \ldots, L$, and the extended feedforward function is $\tilde{F} = \sigma_L \circ Z_L$. Define the non-degenerate locus as:

$$(\mathbf{Param} \times \mathbb{R}^n)^\circ = \{(\boldsymbol{\theta}, x) \mid Z_i(\boldsymbol{\theta}, x) \neq 0 \text{ for } i = 1, \ldots, L - 1\}.$$

**Proposition D.5.** *Suppose that, for $i = 1, \ldots, L - 1$, the activation $\sigma_i : \mathbb{R}^{n_i} \to \mathbb{R}^{n_i}$ satisfies $\sigma_i^{-1}(0) \subseteq \{0\}$. Then there is an action of the hidden symmetry group $\mathrm{GL}_{\mathbf{n}^{\mathrm{hidden}}}$ on the non-degenerate locus given by:*

$$\mathrm{GL}_{\mathbf{n}^{\mathrm{hidden}}} \times (\mathbf{Param} \times \mathbb{R}^n)^\circ \to (\mathbf{Param} \times \mathbb{R}^n)^\circ$$

$$g \cdot (\boldsymbol{\theta}, x) = \left( \left( g_i W_i R_{\tilde{F}_{i-1}(\boldsymbol{\theta}, x)} R_{\sigma_i(g_{i-1} Z_{i-1}(\boldsymbol{\theta}, x))}^{-1}, g_i b_i \right)_{i=1}^{L-1}, x \right)$$

*Moreover, this action preserves the extended feedforward function.*

*Proof.* The fact that the action is well-defined follows from the assumption on each $\sigma_i$ and the non-degeneracy condition. The unit and multiplication axioms are shown in the same way as in the proof of Theorem D.4. For the last claim, one first verifies by induction that $Z_i(g \cdot (\boldsymbol{\theta}, x)) = g_i Z_i(\boldsymbol{\theta}, x)$ for $i = 0, 1, \ldots, L$. Hence,

$$\tilde{F}(g \cdot \boldsymbol{\theta}, x) = \sigma_L(Z_L(g \cdot (\boldsymbol{\theta}, x))) = \sigma_L(g_L Z_L(\boldsymbol{\theta}, x)) = \sigma_L(Z_L(\boldsymbol{\theta}, x)) = \tilde{F}(\boldsymbol{\theta}, x)$$

using the fact that $g_L$ is the identity. So the extended feedforward function is preserved under this action. $\square$

### D.4 Discussion: increasing the batch size

In this section, we discuss difficulties in adopting the construction of the previous sections to cases where the batch size is greater than one. Fix a batch size $k$, so that the feature space of the hidden layer is $\mathbb{R}^{h \times k}$. By abuse notation, we write $\sigma : \mathbb{R}^{h \times k} \to \mathbb{R}^{h \times k}$ for the map applying $\sigma$ column-wise. We say that $\sigma$ *preserves full rank matrices* if $\sigma(Z)$ is full rank for any full-rank matrix in $Z \in \mathbb{R}^{h \times k}$. As a final piece of notation, let $\left(\mathbb{R}^{h \times k}\right)^{\circ} \subseteq \mathbb{R}^{h \times k}$ be the subset of full rank matrices.

**Lemma D.6.** *Suppose that $k \leq h$, and that $\sigma$ preserves full-rank matrices. Then there exists a map* $c : \mathrm{GL}_h \times \left(\mathbb{R}^{h \times k} \setminus \{0\}\right) \to \mathrm{GL}_h$ *satisfying the following identities for any nonzero $Z \in \mathbb{R}^{h \times k}$ and* $g, g_1, g_2 \in \mathrm{GL}_h$:

$$c(\mathrm{id}_h, Z) = \mathrm{id}_h \tag{31}$$
$$c(g_1, g_2 Z)c(g_2, Z) = c(g_1 g_2, Z) \tag{32}$$
$$c(g, Z)\sigma(Z) = \sigma(gZ) \tag{33}$$

We omit a proof of this lemma. A key tool is the fact that, for $k \leq h$, any two matrices in $\left(\mathbb{R}^{h \times k}\right)^{\circ}$ are related by an element of $\mathrm{GL}_h$. This lemma implies that, for a multi-layer network, if $\sigma_i$ preserves full rank matrices in $\mathbb{R}^{n_i \times k}$ for each $i$, then there is a non-linear group action as in Proposition D.5, where the appropriate version of the non-degenerate locus is:

$$\left(\mathbf{Param} \times \mathbb{R}^{n_0 \times k}\right)^{\circ} = \{(\boldsymbol{\theta}, X) \mid Z_i(\boldsymbol{\theta}, X) \in \mathbb{R}^{n_i \times k} \text{ is of full rank for } i = 1, \dots, L - 1\}.$$

However, as the following examples show, the condition that $\sigma$ preserves full rank matrices is not satisfied in the case of common activation functions.

**Example D.1.**

1. For $k > 1$, the column-wise application of the usual sigmoid activation does not preserve full rank matrices. For example, for $k = 2$, take:

$$Z = \begin{bmatrix} \sigma^{-1}\left(\frac{1}{5}\right) & \sigma^{-1}\left(\frac{2}{5}\right) \\ \sigma^{-1}\left(\frac{2}{5}\right) & \sigma^{-1}\left(\frac{4}{5}\right) \end{bmatrix} \simeq \begin{bmatrix} -1.3863 & -0.4055 \\ -0.4055 & 1.3863 \end{bmatrix}$$

Then $\det(\sigma(Z)) = 0$ while $\det(Z) = -2.0862$.

2. For $k > 1$, the column-wise application of hyperbolic tangent does not preserve full rank matrices. To see this, set $k = 1$ and consider:

$$Z = \begin{bmatrix} \tanh^{-1}\left(\frac{1}{5}\right) & \tanh^{-1}\left(\frac{2}{5}\right) \\ \tanh^{-1}\left(\frac{2}{5}\right) & \tanh^{-1}\left(\frac{4}{5}\right) \end{bmatrix} \simeq \begin{bmatrix} 0.2027 & 0.4236 \\ 0.4236 & 1.0986 \end{bmatrix}$$

Then $\det(\tanh(Z)) = 0$ while $\det(Z) = 0.0432$.

3. Let s be a real number with $0 < s < 1$. The corresponding leaky ReLU activation function is given by $\sigma(z) = sz \min(0, z) + z \max(0, z)$. For $k > 1$, the column-wise application of leaky ReLU tangent does not preserve full rank matrices. Indeed, for $k = 2$, set:

$$Z = \begin{bmatrix} s & -1 \\ -1 & s \end{bmatrix}$$

Then $\det(\sigma(Z)) = \det\left(\begin{bmatrix} s & -s \\ -s & s \end{bmatrix}\right) = 0$ while $\det(Z) = s^2 - 1 \neq 0$.

Finally, in the case $k > h$, the action of $\mathrm{GL}_h$ on full rank $h \times k$ matrices is not transitive. Hence, there will generally be no matrix in $\mathrm{GL}_h$ taking $\sigma(Z)$ to $\sigma(gZ)$, even if both are full rank.

### D.5    LIPSCHTIZ BOUNDS

*Proof of Proposition 4.2.* Let $(U, V, x)$ be in the non-degenerate locus, let $g \in \mathrm{GL}_h$, and let $x_1, x_2 \in \mathbb{R}^n$. Using the Lipschitz constant of $\sigma$ and the definition of operator norms, we compute:

$$
\begin{aligned}
|F_{(U,V)}^{(g,x)}(x_1 - x_2)| &\leq |U R_{\sigma(Vx)} R_{\sigma(gVx)}^{-1} \sigma(gV(x_1 - x_2))| \\
&\leq \eta \|U\| \|R_{\sigma(Vx)}\| \|R_{\sigma(gVx)}^{-1}\| \|g\| \|V\| |x_1 - x_2| \\
&= \eta \|U\| \|\sigma(Vx)\| \|\frac{1}{|\sigma(gVx)|^2} R_{\sigma(gVx)}^T\| \|g\| \|V\| |x_1 - x_2| \\
&= \eta \|U\| \frac{|\sigma(Vx)| \|g\|}{|\sigma(gVx)|} \|V\| |x_1 - x_2|
\end{aligned}
$$

The result follows. $\qquad\square$

## E    GRADIENT DESCENT AND DRIFTING CONSERVED QUANTITIES

While gradient flows are well approximated by gradient descent (Elkabetz & Cohen, 2021), the conserved quantities of gradient descent are no longer conserved in gradient flow due to non-infinitesimal time steps. However, with small learning rate, we expect the change in the conserved quantities to be small. In this section, we first prove that the change of $Q$ is bounded by the square of learning rate for two layer linear networks, then show empirically that the change $Q$ is small for nonlinear networks.

### E.1    CHANGE IN $Q$ IN GRADIENT DESCENT (LINEAR LAYERS)

**Proposition E.1.** *Consider the two layer linear network, where $U \in \mathbb{R}^{m \times h}, V \in \mathbb{R}^{h \times n}$ are the only parameters, and the loss function $\mathcal{L}$ is a function of $UV$. In gradient descent with learning rate $\eta$, the change in the conserved quantity $Q = \mathrm{Tr}\left[U^T U - V V^T\right]$ at step $t$ is bounded by*

$$
|Q_{t+1} - Q_t| \leq \eta^2 \left| \frac{d\mathcal{L}(t)}{dt} \right|. \tag{34}
$$

*Proof.* Let $U_t$ and $V_t$ be the value of $U$ and $V$ at time $t$ in a gradient descent. The update rule is

$$
U_{t+1} = U_t - \eta \frac{\partial \mathcal{L}}{\partial U}, \qquad\qquad V_{t+1} = V_t - \eta \frac{\partial \mathcal{L}}{\partial V} \tag{35}
$$

Consider the two layer linear reparametrization $W = UV$.

$$
\begin{aligned}
Q_t &= \mathrm{Tr}\left[U_t^T U_t - V_t V_t^T\right] \\
Q_{t+1} &= \mathrm{Tr}\left[U_{t+1}^T U_{t+1} - V_{t+1} V_{t+1}^T\right]
\end{aligned} \tag{36}
$$

Substituting in $U_{t+1}$ and $V_{t+1}$, expanding $Q_{t+1}$, and subtracting by $Q_t$, we have

$$
\begin{aligned}
Q_{t+1} - Q_t = \mathrm{Tr}\Bigg[ &\eta^2 \left(\frac{\partial \mathcal{L}}{\partial U_t}\right)^T \frac{\partial \mathcal{L}}{\partial U_t} - \eta \left(\frac{\partial \mathcal{L}}{\partial U_t}\right)^T U_t - \eta U_t^T \frac{\partial \mathcal{L}}{\partial U_t} \\
&- \eta^2 \frac{\partial \mathcal{L}}{\partial V_t} \left(\frac{\partial \mathcal{L}}{\partial V_t}\right)^T + \eta \frac{\partial \mathcal{L}}{\partial V_t} V_t^T + \eta V_t \left(\frac{\partial \mathcal{L}}{\partial V_t}\right)^T \Bigg].
\end{aligned} \tag{37}
$$

Note that

$$
\left(\frac{\partial \mathcal{L}}{\partial U_t}\right)^T U_t = (\nabla \mathcal{L} V_t^T)^T U_t = V_t \nabla \mathcal{L}^T U_t = V_t \left(\frac{\partial \mathcal{L}}{\partial V_t}\right)^T, \tag{38}
$$

and similarly

$$
U_t^T \frac{\partial \mathcal{L}}{\partial U_t} = \frac{\partial \mathcal{L}}{\partial V_t} V_t^T. \tag{39}
$$

Therefore, (37) simplifies to

$$Q_{t+1} - Q_t = \eta^2 \text{Tr}\left[\left(\frac{\partial \mathcal{L}}{\partial U_t}\right)^T \frac{\partial \mathcal{L}}{\partial U_t} - \frac{\partial \mathcal{L}}{\partial V_t}\left(\frac{\partial \mathcal{L}}{\partial V_t}\right)^T\right]$$

$$= \eta^2 \left(\text{Tr}\left[\left(\frac{\partial \mathcal{L}}{\partial U_t}\right)^T \frac{\partial \mathcal{L}}{\partial U_t}\right] - \text{Tr}\left[\left(\frac{\partial \mathcal{L}}{\partial V_t}\right)^T \frac{\partial \mathcal{L}}{\partial V_t}\right]\right), \tag{40}$$

and the variation of Q in each step is bounded by the convergence rate:

$$|Q_{t+1} - Q_t| = \eta^2 \left|\text{Tr}\left[\left(\frac{\partial \mathcal{L}}{\partial U_t}\right)^T \frac{\partial \mathcal{L}}{\partial U_t}\right] - \text{Tr}\left[\left(\frac{\partial \mathcal{L}}{\partial V_t}\right)^T \frac{\partial \mathcal{L}}{\partial V_t}\right]\right|$$

$$\leq \eta^2 \left|\text{Tr}\left[\left(\frac{\partial \mathcal{L}}{\partial U_t}\right)^T \frac{\partial \mathcal{L}}{\partial U_t}\right] + \text{Tr}\left[\left(\frac{\partial \mathcal{L}}{\partial V_t}\right)^T \frac{\partial \mathcal{L}}{\partial V_t}\right]\right|$$

$$= \eta^2 \left|\frac{d\mathcal{L}}{dt}\right| \tag{41}$$

$\square$

## E.2 EMPIRICAL OBSERVATIONS

In gradient flow, the conserved quantity $Q$ is constant by definition. In gradient descent, $Q$ varies with time. In order to see how applicable our theoretical results are in gradient descent, we investigate the amount of variation in $Q$ in gradient descent using two-layer neural networks.

Since $Q$ is the difference between the two terms $f_1(U) = \frac{1}{2}\text{Tr}[U^T U]$ and $f_2(V) = \sum_{a,j} \int_{x_0}^{V_{aj}} dx \frac{\sigma(x)}{\sigma'(x)}$, we normalize $Q$ by the initial value of $f_1(U)$ and $f_2(V)$, i.e.,

$$\tilde{Q} = \frac{\left|\frac{1}{2}\text{Tr}[U^T U] - \sum_{a,j} \int_{x_0}^{V_{aj}} dx \frac{\sigma(x)}{\sigma'(x)}\right|}{\left|\frac{1}{2}\text{Tr}[U_0^T U_0]\right| + \left|\sum_{a,j} \int_{x_0}^{V_{0aj}} dx \frac{\sigma(x)}{\sigma'(x)}\right|}$$

and denote the amount of change in $\tilde{Q}$ as

$$\Delta\tilde{Q}(t) = \tilde{Q}(t) - \tilde{Q}(0) \tag{42}$$

We run gradient descent on two-layer networks with whitened input with the following objective

$$\text{argmin}_{U,V}\{\mathcal{L}(U,V) = \|Y - U\sigma(V^T)\|_F^2\} \tag{43}$$

where $\sigma$ is the identity function, ReLU, sigmoid, or tanh. $Y \in \mathbb{R}^{5 \times 10}$, $U \in \mathbb{R}^{5 \times 50}$ and $V \in \mathbb{R}^{10 \times 50}$ have random Gaussian initialization with zero mean. We repeat the gradient descent with learning rate 0.1, 0.01, and 0.001.

The variation $\Delta\tilde{Q}(t)$ and loss is shown in Fig.3. The amount of change in $Q$ is small relative to the magnitude of $f_1(U)$ and $f_2(V)$, indicating that conserved quantities in gradient flow are approximately conserved in gradient descent. The error in $Q$ grows with step size, as $\Delta\tilde{Q}(t)$ is larger with the largest learning rate we used, although it has the same magnitude as those of smaller learning rates. We also observe that $Q$ stays constant after loss converges.

## F   DISTRIBUTION OF $Q$ UNDER XAVIER INITIALIZATION

We first consider a linear two-layer neural network $UVX$, where $U \in \mathbb{R}^{m \times h}, V \in \mathbb{R}^{h \times n}$, and $X \in \mathbb{R}^{n \times k}$. We choose the following form of the conserved quantity:

$$Q = \frac{1}{2}\text{Tr}[U^T U - VV^T]. \tag{44}$$

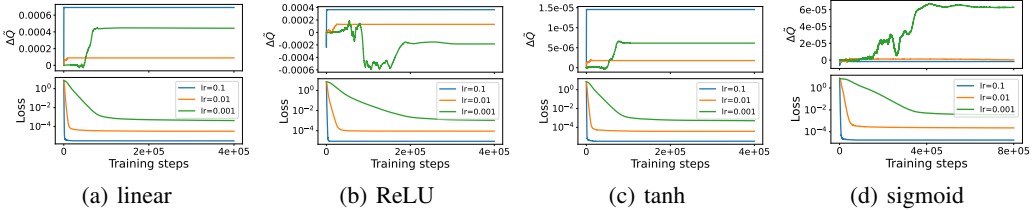

(a) linear          (b) ReLU          (c) tanh          (d) sigmoid

Figure 3: Dynamics of conserved quantities in GD. The amount of change in $Q$ is small relative to its magnitude, and $Q$ converges when loss converges.

Xavier initialization keeps the variance of each layer's output the same as the variance of the input. Under Xavier initialization (Glorot & Bengio, 2010), each element in a given layer is initialized independently, with mean 0 and variance equal to the inverse of the layer's input dimension:

$$U_{ij} = \mathcal{N}\left(0, \frac{1}{h}\right) \qquad\qquad V_{ij} = \mathcal{N}\left(0, \frac{1}{n}\right) \tag{45}$$

The expected value of $Q$ is

$$\mathbb{E}[Q] = Var(U_{ij}) \times m \times h + Var(V_{ij}) \times h \times n = m - h. \tag{46}$$

Figure 4 shows the distribution of $Q$ for 2-layer linear NN with different layer dimensions. For each dimension tuples $(m, h, n)$, we constructed 1000 sets of parameters using Xavier initialization. The centers of the distributions of $Q$ match Eq. (46).

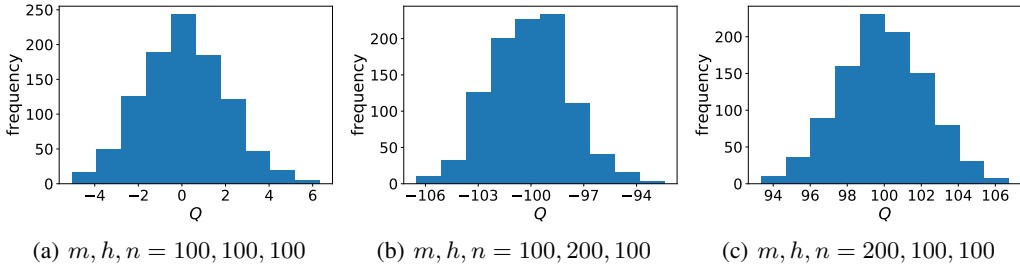

(a) $m, h, n = 100, 100, 100$          (b) $m, h, n = 100, 200, 100$          (c) $m, h, n = 200, 100, 100$

Figure 4: Distribution of $Q$ for 2-layer linear NN with different layer dimensions.

Next, we consider the nonlinear two-layer neural network $U\sigma(VX)$, where $\sigma : \mathbb{R} \to \mathbb{R}$ is an element-wise activation function. For simplicity, we assume whitened input ($X = I$). We choose the following form of the conserved quantity:

$$Q = \frac{1}{2}\text{Tr}[U^T U] - \sum_{a,j} \int_0^{V_{aj}} dx \frac{\sigma(x)}{\sigma'(x)} \tag{47}$$

Figure 5 shows the distribution of $Q$ for 2-layer linear NN with different nonlinearities, each with 1000 sets of parameters created under Xavier initialization. The shapes of the distributions are similar to that of linear networks. The value of $Q$ is usually concentrated around a small range of values. Since the range of $Q$ is unbounded, the Xavier initialization limits the model to a small part of the global minimum.

## G    CONSERVED QUANTITY AND CONVERGENCE RATE

The values of conserved quantities are unchanged throughout the gradient flow. Since the conserved quantities parameterize trajectories, initializing parameters with certain conserved quantity values accelerates convergence. For two-layer linear reparametrization, Tarmoun et al. (2021) derived the explicit relation between layer imbalance and convergence rate. We derive the relation between conserved quantities and convergence rate for two example optimization problems and provide numerical evidence that initializing parameters with optimal conserved quantity values accelerates convergence.

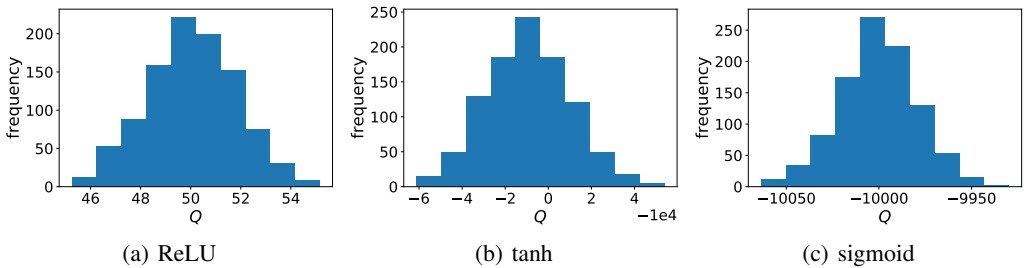

Figure 5: Distribution of $Q$ for 2-layer linear NN with different nonlinearities, with parameter dimensions $m = h = n = 100$.

## G.1 EXAMPLE 1: ELLIPSE

We first show that the convergence rate is related to the conserved quantity in a toy optimization problem. Consider the following loss function with $a \in \mathbb{R}$:

$$\mathcal{L}(w_1, w_2) = w_1^2 + aw_2^2$$
$$\nabla \mathcal{L} = (2w_1, 2aw_2) \tag{48}$$

Assuming gradient flow,

$$\frac{dw_1}{dt} = -\nabla_{w_1}\mathcal{L} = -2w_1 \qquad\qquad \frac{dw_2}{dt} = -\nabla_{w_2}\mathcal{L} = -2aw_2 \tag{49}$$

Then $w_1, w_2$ are governed by the following differential equations:

$$w_1(t) = w_{1_0}e^{-2t} \qquad\qquad w_2(t) = w_{2_0}e^{-2at} \tag{50}$$

where $w_{1_0}, w_{2_0}$ are initial values of $w_1$ and $w_2$. We can find conserved quantities by using an ansatz $Q = f(w_1^i w_2^k)$ and solving $\nabla Q \cdot \nabla L = 0$ for $i, k$. Below we use the following form of conserved quantity:

$$Q = \frac{w_1^{2a}}{w_2^2} = \frac{w_{1_0}^{2a}}{w_{2_0}^2} \tag{51}$$

To show the effect of $Q$ on the convergence rate, we fix $L(0)$ and derive how $Q$ affects $L(t)$. Let $L(0) = w_{1_0}^2 + aw_{2_0}^2 = L_0$. Let $w_{2_0}$ continue to be an independent variable. Then $w_{1_0}^2 = L_0 - aw_{2_0}^2$. Substitute in $w_{1_0}^2$, the loss at time $t$ is

$$L(t) = w_1(t)^2 + aw_2(t)^2 = (L_0 - aw_{2_0}^2)e^{-4t} + aw_{2_0}^2 e^{-4at} \tag{52}$$

and $Q$ becomes

$$Q = \frac{w_{1_0}^{2a}}{w_{2_0}^2} = \frac{(L_0 - aw_{2_0}^2)^a}{w_{2_0}^2} \tag{53}$$

The derivative of $L$ in the direction of $Q$ is

$$\partial_Q L(t) = \frac{dL(t)}{dw_{2_0}}\frac{dw_{2_0}}{dQ} = \frac{dL(t)}{dw_{2_0}}\left(\frac{dQ}{dw_{2_0}}\right)^{-1}$$

$$= \left(-2aw_{2_0}e^{-4t} + 2aw_{2_0}e^{-4at}\right)\left(\frac{a(L_0 - aw_{2_0}^2)^{a-1}(-2aw_{2_0})w_{2_0}^2 - 2w_{2_0}(L_0 - aw_{2_0}^2)^a}{w_{2_0}^4}\right)^{-1}$$

$$= \frac{\left(-2aw_{2_0}e^{-4t} + 2aw_{2_0}e^{-4at}\right)w_{2_0}^4}{a(L_0 - aw_{2_0}^2)^{a-1}(-2aw_{2_0})w_{2_0}^2 - 2w_{2_0}(L_0 - aw_{2_0}^2)^a}$$

$$= \frac{2aw_{2_0}^5\left(e^{-4at} - e^{-4t}\right)}{2w_{2_0}(L_0 - aw_{2_0}^2)^{a-1}\left(-a^2w_{2_0}^2 - (L_0 - aw_{2_0}^2)\right)} \tag{54}$$

In general, $\partial_Q L(t) \neq 0$, meaning that the loss at time $t$ depends on $Q$. Since we have fixed the initial loss, the convergence rate $L(t) - L(0)$ also depends on $Q$. Special cases where $\partial_Q L(t) = 0$ include $a = 1$ (circle), $a = 0$ (collapsed dimension), and certain initializations such as $w_{2_0} = 0$ (local maximum of gradient magnitude).

### G.2 EXAMPLE 2: RADIAL ACTIVATION FUNCTIONS

In this example, we find the conserved quantities and their relation with convergence rate for two-layer reparametrization with radial activation functions under spectral initialization.

Define radial function $g : \mathbb{R}^{m \times n} \to \mathbb{R}^{m \times n}$ as

$$g(W)_{ij} = h\left(|W_i|\right) W_{ij}, \tag{55}$$

where $|W_i| = \left(\sum_k W_{ik}^2\right)^{\frac{1}{2}}$ is the norm of the $i^{th}$ row of $W$, and $h : \mathbb{R} \to \mathbb{R}$ outputs a scalar.

Consider the following objective:

$$\text{argmin}_{U,V}\{\mathcal{L}(U,V) = \frac{1}{2}\|Y - Ug(V^T)\|_F^2\} \tag{56}$$

with spectral initializations

$$U_0 = \Phi\overline{U}_0, V_0 = \Psi\overline{V}_0,$$

where $\Phi, \Psi$ come from the singular value decomposition $Y = \Phi\Sigma_Y\Psi^T$, and $\overline{U}_0, \overline{V}_0$ are random diagonal matrices.

**Proposition G.1.** *Under the gradient flow $U = -\nabla_U\mathcal{L}$ and $V = -\nabla_V\mathcal{L}$, the following quantity is an invariant:*

$$Q = \frac{1}{2}\text{Tr}[U^T U] - \sum_i \int_{x_0}^{\overline{V}_{ii}} dx \frac{g(x)}{g'(x)} \tag{57}$$

*Proof.* Since $g$ is a radial function on rows and $\Psi^T$ is an orthogonal matrix, $g(\overline{V}^T\Psi^T) = g(\overline{V}^T)\Psi^T$. With spectral initialization, the loss function can be reduced to only involving diagonal matrices:

$$\begin{aligned} \mathcal{L} &= \frac{1}{2}\|Y - Ug(V^T)\|_F^2 \\ &= \frac{1}{2}\|\Phi\Sigma\Psi^T - \Phi\overline{U}g[(\Psi\overline{V})^T]\|_F^2 \\ &= \frac{1}{2}\|\Phi\Sigma\Psi^T - \Phi\overline{U}g(\overline{V}^T)\Psi^T\|_F^2 \\ &= \frac{1}{2}\|\Phi\left(\Sigma - \overline{U}g(\overline{V}^T)\right)\Psi^T\|_F^2 \\ &= \frac{1}{2}\|\Sigma - \overline{U}g(\overline{V}^T)\|_F^2 \end{aligned} \tag{58}$$

Since $\overline{V}$ is a diagonal matrix, $g$ is now an element wise function on $\overline{V}$. Let $\overline{W} = \overline{U}g(\overline{V}^T)$. The gradients for $\overline{U}$ and $\overline{V}$ are

$$\begin{aligned} \frac{\partial\mathcal{L}}{\partial\overline{U}} &= \nabla_{\overline{W}}\mathcal{L}g(\overline{V})^T \\ \frac{\partial\mathcal{L}}{\partial\overline{V}} &= \nabla_{\overline{W}}\mathcal{L}^T\overline{U} \odot g'(\overline{V}) \end{aligned} \tag{59}$$

where $g'(x) = dg(x)/dx$ is the derivative of the nonlinearity. Additionally, since $\mathcal{L}$ does not depend on $\Phi$ and $\Psi$,

$$\frac{\partial\mathcal{L}}{\partial\Phi} = \frac{\partial\mathcal{L}}{\partial\Psi} = 0 \tag{60}$$

Since the rows of $\Phi, \Psi$ are orthogonal,

$$\begin{aligned} \frac{\partial\mathcal{L}}{\partial U} &= \frac{\partial\mathcal{L}}{\partial\overline{U}}\Phi^T = \nabla_{\overline{W}}\mathcal{L}g(\overline{V})^T\Phi^T \\ \frac{\partial\mathcal{L}}{\partial V} &= \frac{\partial\mathcal{L}}{\partial\overline{V}}\Psi^T = \left(\nabla_{\overline{W}}\mathcal{L}^T\overline{U} \odot g'(\overline{V})\right)\Psi^T \end{aligned} \tag{61}$$

$\Phi$ and $\Psi$ are not changed in gradient flow, so $\frac{\partial Q}{\partial U} = \frac{\partial Q}{\partial \overline{U}}\Phi^T$ and $\frac{\partial Q}{\partial V} = \frac{\partial Q}{\partial \overline{V}}\Psi^T$. Define inner product on matrices as $\langle X, Y \rangle = \text{Tr}[X^T Y]$. For $Q$ to be a conserved quantity, we need $\langle \nabla \mathcal{L}, \nabla Q \rangle = 0$:

$$
\begin{aligned}
\langle \nabla \mathcal{L}, \nabla Q \rangle &= \langle \frac{\partial \mathcal{L}}{\partial U}, \frac{\partial Q}{\partial U} \rangle + \langle \frac{\partial \mathcal{L}}{\partial V}, \frac{\partial Q}{\partial V} \rangle \\
&= \langle \nabla_{\overline{W}}\mathcal{L}g(\overline{V})^T \Phi^T, \frac{\partial Q}{\partial \overline{U}}\Phi^T \rangle + \langle \left( \nabla_{\overline{W}}\mathcal{L}^T \overline{U} \odot g'(\overline{V}) \right) \Psi^T, \frac{\partial Q}{\partial \overline{V}}\Psi^T \rangle \\
&= \langle \nabla_{\overline{W}}\mathcal{L}g(\overline{V})^T, \frac{\partial Q}{\partial \overline{U}} \rangle + \langle \left( \nabla_{\overline{W}}\mathcal{L}^T \overline{U} \odot g'(\overline{V}) \right), \frac{\partial Q}{\partial \overline{V}} \rangle \\
&= \text{Tr} \left[ \partial_{\overline{U}^T} Q \nabla_{\overline{W}}\mathcal{L}g(\overline{V})^T + \overline{U}^T \nabla_{\overline{W}}\mathcal{L}(\partial_V Q \odot g'(\overline{V})) \right] = 0
\end{aligned}
\tag{62}
$$

Following the same procedure as for elementwise functions, to have a $Q$ which satisfies (62) it is sufficient to have

$$
\frac{\partial Q}{\partial \overline{U}_{ia}} = f(\overline{U}, \overline{V})\overline{U}_{ia} \qquad \frac{\partial Q}{\partial \overline{V}_{aj}}g'(\overline{V})_{aj} = -f(\overline{U}, \overline{V})g(\overline{V})_{aj} \qquad f(\overline{U}, \overline{V}) \in \mathbb{R}
\tag{63}
$$

For simplicity, let $f(\overline{U}, \overline{V}) = 1$. Then, (63) is satisfied by

$$
Q = \frac{1}{2}\text{Tr}[\bar{U}^T \bar{U}] - \sum_i \int_{x_0}^{\bar{V}_{ii}} dx \frac{g(x)}{g'(x)}
\tag{64}
$$

$\square$

Tarmoun et al. (2021) shows that the conserved quantity $Q$ appears as a term in the convergence rate of the matrix factorization gradient flow. We observe a similar relationship between $Q$ and convergence rate when the loss function is augmented with a radial activation function, as shown in the following proposition.

**Proposition G.2.** *Consider the objective function and spectral initialization defined in Proposition G.1. Let $h(|W_i|) = |W_i|^{-2}$, and $X = Ug(V^T) = \Phi\Sigma_X\Psi^T$. Then, the eigencomponent of $X$ approaches the corresponding eigencomponent of $Y$ at a rate of*

$$
\dot{\sigma}_i^X = \frac{1}{\lambda_i}(\sigma_i^Y - \sigma_i^X)(\sigma_i^{X2} + 1)^2,
\tag{65}
$$

*where $\sigma_i^X = diag(\Sigma_X)_i$, $\sigma_i^Y = diag(\Sigma_Y)_i$, and $\lambda_i = \bar{U}_{ii}^2 + \bar{V}_{ii}^2$ are conserved quantities.*

*Proof.* Similar to Tarmoun et al. (2021), components can be decoupled, and we have a set of differential equations on scalars:

$$
\begin{aligned}
\dot{\overline{u}}_i &= [\sigma_i^Y - \overline{u}_i g(\overline{v}_i)]g(\overline{v}_i) \\
\dot{\overline{v}}_i &= [\sigma_i^Y - \overline{u}_i g(\overline{v}_i)]\overline{u}_i \frac{dg(\overline{v}_i)}{d\overline{v}_i}
\end{aligned}
\tag{66}
$$

We also have

$$
\dot{g}(\overline{v}_i) = \frac{dg}{d\overline{v}_i}\frac{d\overline{v}_i}{dt} = [\sigma_i^Y - \overline{u}_i g(\overline{v}_i)]\overline{u}_i \left( \frac{dg(\overline{v}_i)}{d\overline{v}_i} \right)^2.
\tag{67}
$$

Let $\sigma_i^X = \overline{u}_i g(\overline{v}_i)$. Then

$$
\begin{aligned}
\dot{\sigma}_i^X &= \dot{\overline{u}}_i g(\overline{v}_i) + \overline{u}_i \dot{g}(\overline{v}_i) \\
&= [\sigma_i^Y - \overline{u}_i g(\overline{v}_i)] \left[ g(\overline{v}_i)^2 + \overline{u}_i^2 \left( \frac{dg(\overline{v}_i)}{d\overline{v}_i} \right)^2 \right].
\end{aligned}
\tag{68}
$$

Since $\overline{V}$ is a diagonal matrix, $g$ is now an element wise function on $\overline{V}$. Specifically, $g(\overline{v}_i) = \frac{1}{\overline{v}_i}$. According to Proposition G.1, the following quantity is invariant:

$$
\frac{1}{2}\overline{u}_i^2 - \int dx \frac{g(x)}{g'(x)} = \frac{1}{2}\overline{u}_i^2 - \int dx \frac{\overline{v}_i^{-1}}{-\overline{v}_i^{-2}} = \frac{1}{2}\overline{u}_i^2 + \frac{1}{2}\overline{v}_i^2
\tag{69}
$$

Since any function of the invariant is also invariant, we will use the following form:

$$Q = \overline{U}^T \overline{U} + \overline{V}^T \overline{V}, \tag{70}$$

and define

$$\lambda_i = Q_{ii} = \overline{u}_i^2 + \overline{v}_i^2 \tag{71}$$

Using the $g$ that we defined,

$$\sigma_i^X = \overline{u}_i g(\overline{v}_i) = \overline{u}_i \overline{v}_i^{-1}. \tag{72}$$

In order to relate $\sigma^X$ and $Q$, we first write $\overline{u}_i$ and $\overline{v}_i$ as functions of $\sigma_i^X$ ad $Q$ using (71) and (72):

$$\overline{u}_i^2 = \frac{\lambda_i \sigma_i^{X\,2}}{\sigma_i^{X\,2} + 1}, \qquad\qquad \overline{v}_i^2 = \frac{\lambda_i}{\sigma_i^{X\,2} + 1}. \tag{73}$$

Then, substitute $\overline{u}_i$, $\overline{v}_i$, $g(\overline{v}_i)$, and $\frac{dg(\overline{v}_i)}{d\overline{v}_i}$ into (68), and we have

$$
\begin{aligned}
\dot{\sigma}_i^X &= [\sigma_i - \overline{u}_i g(\overline{v}_i)] \left[ g(\overline{v}_i)^2 + \overline{u}_i^2 \left( \frac{dg(\overline{v}_i)}{d\overline{v}_i} \right)^2 \right] \\
&= [\sigma_i^Y - \overline{u}_i g(\overline{v}_i)] \left[ \left( \frac{1}{\overline{v}_i} \right)^2 + \overline{u}_i^2 \left( -\overline{v}_i^{-2} \right)^2 \right] \\
&= [\sigma_i^Y - \overline{u}_i g(\overline{v}_i)] \left[ (\overline{v}_i^2)^{-1} + \overline{u}_i^2 (\overline{v}_i^2)^{-2} \right] \\
&= [\sigma_i^Y - \sigma_i^X] \left[ \left( \frac{\lambda_i}{\sigma_i^{X\,2} + 1} \right)^{-1} + \frac{\lambda_i \sigma_i^{X\,2}}{\sigma_i^{X\,2} + 1} \left( \frac{\lambda_i}{\sigma_i^{X\,2} + 1} \right)^{-2} \right] \\
&= [\sigma_i^Y - \sigma_i^X] \left[ \frac{\sigma_i^{X\,2} + 1}{\lambda_i} + \frac{\sigma_i^{X\,2}(\sigma_i^{X\,2} + 1)}{\lambda_i} \right] \\
&= [\sigma_i^Y - \sigma_i^X] \left[ \frac{\sigma_i^{X\,4} + 2\sigma_i^{X\,2} + 1}{\lambda_i} \right] \\
&= \frac{1}{\lambda_i} (\sigma_i^Y - \sigma_i^X)(\sigma_i^{X\,2} + 1)^2
\end{aligned}
\tag{74}
$$

$\square$

Proposition G.2 relates the rate of change in parameters $\dot{\sigma}_i^X$ and the conserved quantity $\lambda_i$. To get a more explicit expression of how $\lambda_i$ affects convergence rate, we will derive a bound for $|\sigma_i^Y - \sigma_i^X|$, which describes the distance between trainable parameters to their desired value.

**Proposition G.3.** *The difference between the singular values of $Ug(V^T)$ and $Y$ is bounded by*

$$|\sigma_i^X - \sigma_i^Y| \le |\sigma_i^X(0) - \sigma_i^Y| e^{-\frac{t}{\lambda_i}}. \tag{75}$$

*Proof.* Note that

$$\dot{\sigma}_i^X = \frac{1}{\lambda}(\sigma_i^Y - \sigma_i^X)(\sigma_i^{X\,2} + 1)^2 \ge \frac{1}{\lambda_i}(\sigma_i^Y - \sigma_i^X) \tag{76}$$

Consider the following two differential equations, with same initialization $a(0) = b(0)$:

$$\dot{a} = \frac{1}{\lambda}(\sigma - a)(a^2 + 1)^2$$

$$\dot{b} = \frac{1}{\lambda}(\sigma - b) \tag{77}$$

In these equations, both $a$ and $b$ moves from $a(0) = b(0)$ to $\sigma$ monotonically. Since $\dot{a} \ge \dot{b}$ at every $a = b$, $a$ will always be closer to $\sigma$ than $b$ does. We can explicitly solve for $b$, which yields

$b(t) = \sigma + (b(0) - \sigma)e^{-\frac{t}{\lambda}}$. Then the distance between $b$ and $\sigma$ is $|b - \sigma| = |b(0) - \sigma|e^{-\frac{t}{\lambda}}$. Using $|b - \sigma|$, we can bound $|a - \sigma|$:

$$|a - \sigma| \leq |b - \sigma| = |b(0) - \sigma|e^{-\frac{t}{\lambda}} \tag{78}$$

Therefore,

$$|\sigma_i^X - \sigma_i^Y| \leq |\sigma_i^X(0) - \sigma_i^Y|e^{-\frac{t}{\lambda_i}} \tag{79}$$

$\square$

Since $\lambda$ is a conserved quantity, its value set at initialization remains unchanged throughout the gradient flow. Therefore, we are able to optimize the convergence rate by choosing a favorable value for $\lambda$ at initialization. In this example, smaller $\lambda_i$'s lead to faster convergence.

### G.3 EXPERIMENTS

We compare the convergence rate of two-layer networks initialized with different $Q$ values. We run gradient descent on two-layer networks with whitened input with the following objective

$$\text{argmin}_{U,V}\{\mathcal{L}(U,V) = \|Y - U\sigma(V^T)\|_F^2\} \tag{80}$$

where $\sigma$ is the identity function, ReLU, sigmoid, or tanh. Matrices $Y \in \mathbb{R}^{5 \times 10}$, $U \in \mathbb{R}^{5 \times 50}$ and $V \in \mathbb{R}^{10 \times 50}$ have random Gaussian initialization with zero mean. We repeat the gradient descent with learning rate 0.1, 0.01, and 0.001. The learning rate is set to $10^{-3}$, as we do not observe significant changes in the shape of learning curves at smaller learning rates. $U$ and $V$ are initialized with different variance, which leads to different initial values of $Q$.

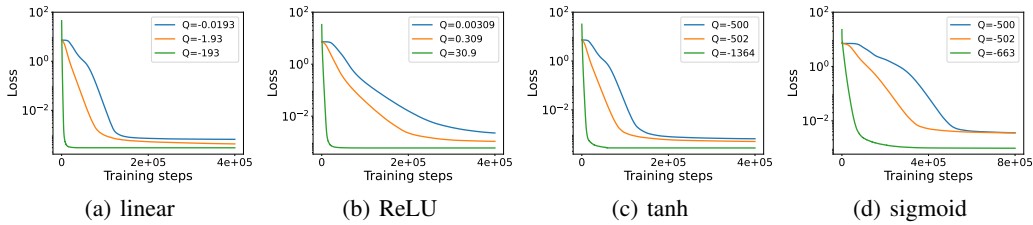

(a) linear       (b) ReLU       (c) tanh       (d) sigmoid

Figure 6: Training curves of two-layer networks initialized with different $Q$. The value of $Q$ affects convergence rate.

As shown in Fig.6, the number of steps required for the loss curves to drop to near convergence level is correlated with $Q$ in both linear and element-wise nonlinear networks. This result provides empirical evidence that initializing parameters with optimal values for $Q$ accelerates convergence.

We then demonstrate the effect of conserved quantity values on the convergence rate of radial neural networks. Fig.7 shows the training curve for loss function defined in Proposition G.2. We initialize parameters $U \in \mathbb{R}^{5 \times 5}$ and $V \in \mathbb{R}^{10 \times 5}$ with 4 different values of $Q$ and the learning rate is set to $10^{-5}$. As predicted in Eq. 75, convergence is faster when $Q = \text{Tr}[U^T U + V^T V]$ is small.

## H CONSERVED QUANTITY AND GENERALIZATION ABILITY

Conserved quantities parameterize the minimum of neural networks and are related to the eigenvalues of the Hessian at minimum. Recent theory and empirical studies suggest that sharp minimum do not generalize well (Hochreiter & Schmidhuber, 1997; Keskar et al., 2017; Petzka et al., 2021). Explicitly searching for flat minimum has been shown to improve generalization bounds and model performance (Chaudhari et al., 2017; Foret et al., 2020; Kim et al., 2022). We derive their relationship for the simplest two-layer network, and show empirically that conserved quantity values affect sharpness. Like convergence rate, a systematic study of the relationship between conserved quantity and generalization ability of the solution is an interesting future direction.

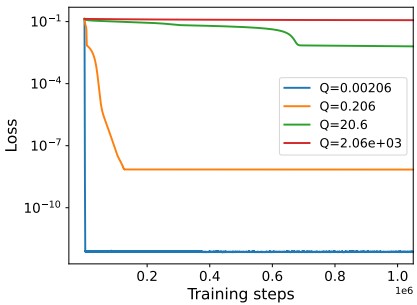

Figure 7: Training curve for the loss function defined in Proposition G.2. Smaller value of $Q = \text{Tr}[U^T U + V^T V]$ at initialization leads to faster convergence.

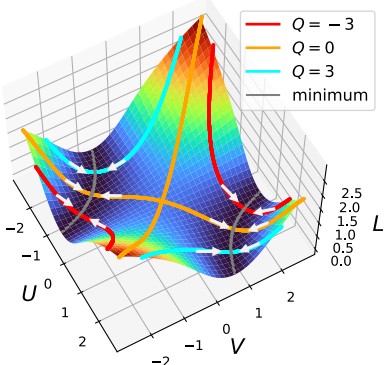

Figure 8: Gradient flow for $L(U, V) = \frac{1}{2}\|Y - UVX\|^2$, where $U, V \in \mathbb{R}$, $Y = 2$, and $X = 1$. Trajectories corresponding to different values of $Q$ intersect the minima at different points.

### H.1 EXAMPLE: TWO-LAYER LINEAR NETWORK WITH 1D PARAMETERS

We again consider the two-layer linear network with loss $\mathcal{L} = \frac{1}{2}\|Y - UVX\|^2$. For simplicity, we work with one dimensional parameters $U, V \in \mathbb{R}$ and assume $X = Y = 1$ in this example. We show that at the point to which the gradient flow converges, the eigenvalues of the Hessian are related to the value of the conserved quantity.

The gradients and Hessian of $\mathcal{L}$ are

$$\nabla\mathcal{L} = \begin{bmatrix} -(Y - UVX)VX \\ -(Y - UVX)UX \end{bmatrix} \qquad \mathcal{H} = \begin{bmatrix} V^2 X^2 & -YX + 2UVX^2 \\ -YX + 2UVX^2 & U^2 X^2 \end{bmatrix} \qquad (81)$$

At the minima, $U, V$ are related by $UVX = Y$. Recall that $Q = U^2 - V^2$ is a conserved quantity. From the above two equations, we can write $U, V$ as functions of $Q$. Taking the solution $U = \sqrt{\frac{1}{2}(Q + \sqrt{Q^2 + 4})}, V = \sqrt{\frac{1}{2}(-Q + \sqrt{Q^2 + 4})}$ and substitute in $X = Y = 1$, we have

$$\mathcal{H} = \begin{bmatrix} \frac{1}{2}(-Q + \sqrt{Q^2 + 4}) & 1 \\ 1 & \frac{1}{2}(Q + \sqrt{Q^2 + 4}) \end{bmatrix}, \qquad (82)$$

and the eigenvalues of $\mathcal{H}$ are

$$\lambda_1 = 0, \qquad\qquad \lambda_2 = 2\sqrt{Q^2 + 4}. \qquad (83)$$

We have shown that $Q$ is related the eigenvalues of the Hessian at the minimum. Since the eigenvalues determines the curvature, $Q$ also determines the sharpness of the minimum, which is believed to related to model's generalization ability. The result in this example can also be observed in Figure 1, where the minimum of the $Q = 0$ trajectory lies at the least sharp point of the loss valley.

## H.2 EXPERIMENTS: TWO-LAYER NETWORKS

The goal of this section is to explore the relation between $Q$ and the sharpness of the trained model. We measure sharpness by the magnitude of the eigenvalues of the Hessian, which are related to the curvature at the minima. We use the same loss function (80) in Section G.3. The parameters are $U \in \mathbb{R}^{10 \times 50}$ and $V \in \mathbb{R}^{5 \times 50}$, each initialized with zero mean and various standard deviations that lead to different $Q$'s. We first train the models using gradient descent. We then use the vectorized parameters in the trained model to compute the eigenvalues of the Hessian.

The linear model extends the example in Section H.1 to higher dimension parameter spaces. 700 out of the 750 eigenvalues are around 0 (with magnitude $\leq 10^{-3}$), which verifies the dimension of the minima in Proposition C.3. After removing the small eigenvalues, the center of the eigenvalue distribution correlates positively with the value of $Q$ (Figure 9(a)). In models with nonlinear activations, $Q$ is still related to eigenvalue distributions, although the relations seem to be more complicated.

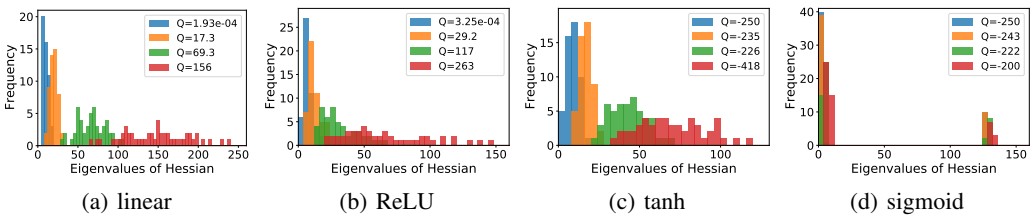

(a) linear      (b) ReLU      (c) tanh      (d) sigmoid

Figure 9: Eigenvalues of the Hessian from trained models initialized with different conserved quantity values ($Q$). The distribution of the eigenvalues and the value of $Q$ appear to be related.

## I ENSEMBLE MODELS

In neural networks, the optima of the loss functions are connected by curves or volumes, on which the loss is almost constant (Freeman & Bruna, 2017; Garipov et al., 2018; Draxler et al., 2018; Benton et al., 2021; Izmailov et al., 2018). Various algorithms have been proposed to find these low-cost curves, which provides a low-cost way to create an ensemble of models from a single trained model. Using our group actions, we propose a new way of constructing models with similar loss values. We show that even with stochasticity in the data, the loss is approximately unchanged under the group action (Appendix I). This provides an efficient alternative to build ensemble models, since the transformation only requires random elements in the symmetry group, without any searching or additional optimization.

We implement our group actions by modifying the activation function between two consecutive layers. Let $H = VX$ be the output of the previous layer. The group action on the weights $U, V$ is

$$g \cdot (U, V) = (U\pi(g, H), gV) \tag{84}$$

where $\pi(g, H) = \sigma(H)\sigma(gH)^\dagger$. The new activation implements the symmetry group action

$$U\sigma(H) \to U\pi(g, H)\sigma(gH) \tag{85}$$

by wrapping the transformations around an activation function $\sigma'(x) = \pi(g, x)\sigma(gx)$, so that $U\sigma'(H) = U\pi(g, H)\sigma(gH)$.

We test the group action on CIFAR-10. The model contains a convolution layer with kernel size 3, followed by a max pooling, a fully connected layer, a leaky ReLU activation, and another fully connected layer. The group action is on the last two fully connected layers. After training a single model, we create transformed models using $g = I + \varepsilon M$, where $M \in \mathbb{R}^{32 \times 32}$ is a random matrix and $\varepsilon$ controls the magnitude of movement in the parameter space. We then use the mode of the transformed models' prediction as the final output.

We compare the ensemble formed by group actions to four ensembles formed by various random transformation. Let $g = I + \varepsilon M$. The random baselines are:

- 'group': $(U, V) \mapsto (U\pi(g, H), gV)$. This is the model created by group actions.

- '$g^{-1}$': $(U, V) \mapsto (Ug^{-1}, gV)$.
- 'random': $(U, V) \mapsto (Ug', gV)$, where $g' = I + \varepsilon D$ and $D$ is a random diagonal matrix.
- 'shuffle': $(U, V) \mapsto (U\pi'(g, H), gV)$, where $\pi'(g, H)$ is constructed by randomly shuffling $\pi(g, H)$.
- 'interpolated permute' or 'perm_interp': $(U, V) \mapsto (U\left(\frac{I+\frac{\varepsilon}{2}(I+S)}{I+\varepsilon}\right)^{-1}, \frac{I+\frac{\varepsilon}{2}(I+S)}{I+\varepsilon}V)$, where $S \in \mathbb{R}^{32 \times 32}$ is a random permutation matrix.

Figure 10 shows the accuracy of the ensembles compared to single models. The ensemble formed by group actions preserves the model accuracy for small $\varepsilon$ and has smaller accuracy drop at larger $\varepsilon$. The ensemble model also improves robustness against Fast Gradient Signed Method (FGSM) attacks (Figure 11). Under FGSM attacks with various strength, the ensemble model created using group actions consistently performs better than the baselines with random transformations. However, the same improvement is not observed under Projected Gradient Descent (PGD) attacks.

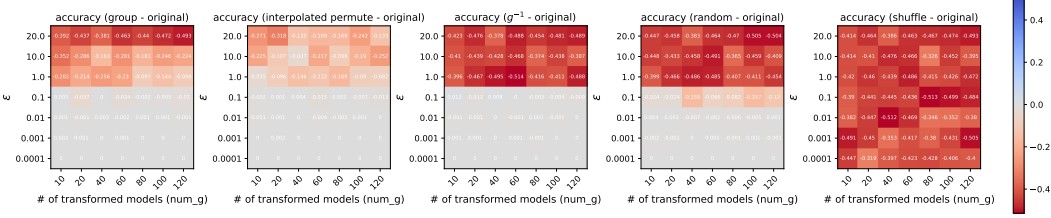

Figure 10: Change in accuracy compared to the original single model when using the ensemble model and 4 baselines. The red color indicates degradation in model performance. The ensemble created by group actions has similar loss values when $\varepsilon$ is small.

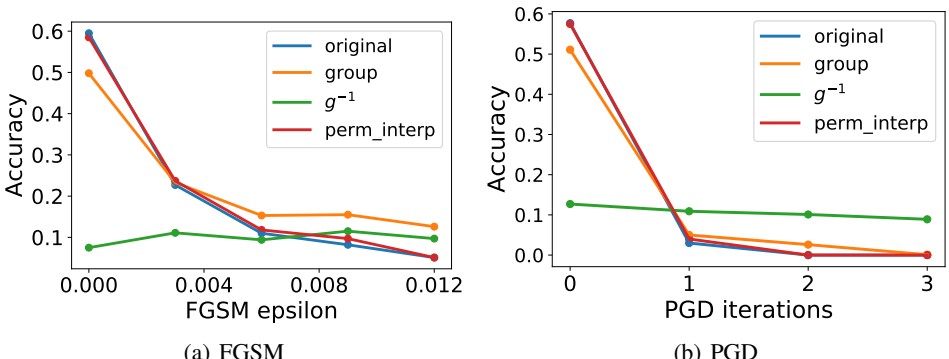

Figure 11: Adversarial attacks on the original model and the ensemble models with various strengths. In FGSM, the group ensemble model improves robustness. In PGD, the ensemble has negligible effects on robustness.

