# OpenReview forum: "Symmetries, Flat Minima, and the Conserved Quantities of Gradient Flow"
_ICLR.cc/2023/Conference — ICLR 2023 poster_

### Official Review · Reviewer_M5tH · 2022-10-19

**Confidence:** 4
**Correctness:** 3
**Technical Novelty And Significance:** 3
**Empirical Novelty And Significance:** Not applicable
**Recommendation:** 6

**Clarity, Quality, Novelty And Reproducibility:**

I already tackled these questions earlier. Let me ask a question here.

In 2-layer diagonal linear networks: that is 2-layers linear neural networks with only diagonal connexion between matrices, it is known that the gradient flow has a nice non-trivial invariance property: it can be recasted as a mirror flow, where the mirror map $\nabla \psi: \mathbb{R}^d \to \mathbb{R}^d$ is some explicit function. Say the coordinate-wise logarithm (it is almost true that it is the case). Then, it can be show that $ \forall t>0, \ \nabla \psi(\theta_t) \in \textrm{span} (X^\top), $ where $\theta_t$ is as in the article the dynamics of the gradient flow at time $t$ and $X$ is the design matrix (See for a reference http://proceedings.mlr.press/v125/woodworth20a/woodworth20a.pdf). Hence,  $\nabla \psi(\theta_t)$ is not a *conserved*  quantity strictly speaking but a nice *geometric invariant of the dynamics*. Is this possible to derive the point of view on group invariance to unveil this without resorting to what is now to me more like a "miracle" ?

**Strength And Weaknesses:**

## **Strength**

The first real strength of the paper is that, despite the technicality of the theory of invariance, the paper is pretty clear and easy to follow (despite some paragraphs and typos -- see *Weaknesses*). Overall, at least at the beginning of the paper, there is a real care about pedagogy, and, being personally not familiar with this theory, I felt that the paper is really understandable from this perspective.

Furthermore, beyond this point of view, two results appeared particularly interesting, at least to me:
- They manage to recover, with their general framework, standard invariant already derived in precedent works that were simply seen as "tricks"
- The fact that the authors try to tackle data-dependant symmetries

---------------------------------------------

## **Weaknesses**

Here is a list of potential weaknesses that I personally felt when reading the paper:

- First, even if the authors try to tackle the case of potential data-dependant non-linear symmetries, they do not find any application of this perpective for neural networks. In fact, the only conclusive result they have on these is the known balance property, but this *only* comes from a function symmetry and is not data-related. Do the authors have in mind an example where some data invariance is of paramount importance ? (I have a more detailed question about this in the next box).
- In fact, going further, the different paragraphs of Subsection 1.5 are really fuzzy and there is a lack of details in the calculations. Propositions and facts are inline without emphasis between them so that it is very diffiicult to follow. Furthermore, it appeared to me (maybe I am wrong), that some of the derivations are very fast and this prevents for a proper easy reading of this *very technical* section. Examples of these are e.g.: (i) second line of the **General equivariance** paragraph $c(g_1, g_2 Z) c(g_1, g_2 Z) = c(g_1g_2, Z)$, what is a typical $c$? What does it mean "with physical interpretation"? why is this property called "equivariance"? This paragraph lacks of lot of details (both in the interpretation and in calculation) to be followed. (ii) Second line of **Using nonlinear transformed [...]** paragraph: do really the authors want to write: $g \cdot (U, V, X) = (\tilde{U} (g, VX), gV, X)$, and in this case, what is $\tilde{U}$? In this paragraph, it is concluded that at permutation + small deformation of the data set the function should be almost the same: why should we care about this perturbative results? I have in mind that the assumption on the data for generalisation is more that they come iid from the same distribution not that they are close to each other in distance? (iii) The last paragraph of this section on **adapting models** is very hard to follow, notations are not properly introduced, e.g. in $Z' = gZP$, what is $g$? Why should we look for $U'\sigma(VX') = U\sigma(VX)P$? I am sure there are some calculations that permit to go from one line to the others and things are clear in the minds of the authors, but I felt in was impossible to follow this paragraph. Maybe a real mathematical Proposition (proved in Appendix) can clarify everything here: e.g. *Assume X' is such that $\exists g \in G$ such that for all $V$, [...]*.
- As far as I understood, the authors are not the first to consider symmetries to explain invariants of the dynamics. Comparison with results of previous works would be welcome to understand how it articulates with theses previous works. I am sure that all the propositions of Section 1 are well-known facts for example.
- Minor issues: Eq (6), the gradient is seen as a linear form in this article it seems because operations with matrices are left operations. It can be troubling so it needs to be said. Typo page 6, paragraph 3: $\sigma(a) - \sigma(b)$. Paragraph **need for data-dependant symmetries**, isnt this $\mathcal{L}(\theta', X') = \mathcal{L}(\theta, X') $ (and not $\mathcal{L}(\theta, X)$). After proposition 1.6, $F(x)$ should be $U\sigma (Vx)$. After formula (10), the infinitesimal form of $\sigma(gz)$ ... Like previously the formula (11) is weird as $\langle Mz, d\sigma_z \rangle$ is a vector and the $\langle \cdot, \cdot \rangle$ usually indicate a scalar product. As far as I understand, the authors may come from a theoretical physics community where such notations may be usual.

**Summary Of The Paper:**

The paper provides an analysis of symmetries inherent to the architecture of certain models (overall the study is centred in neural-networks-like predictors), as well as the loss and data. Link to the set of local minimizers is also made. Finally, it is also shown that these symmetries have a corresponding echo in conserved quantities during gradient flow training.

**Summary Of The Review:**

Overall, even if the paper could be very nice I think that I feel disappointed when it comes to summarize the results. Beyond learning a nice theory of Lie invariant, I do not know what I have really learnt after reading this. The novelty and promising application to data-dependant symmetry, in particular, is a bit disappointing.

I will be very glad to largely increase my score if the authors answer my concerns and questions.

---

> ### Author Response · Authors · 2022-11-15
> **Part 1**
>
>
> Thank you, we really appreciate your detailed and constructive comments.
>
>
> > First, even if the authors try to tackle the case of potential data-dependant non-linear symmetries, they do not find any application of this perpective for neural networks.
>
> We are still trying to understand the implications, and limitations, of this new class symmetries. As our preliminary experimental results suggest, they might be useful for improving convergence rate, or robustness, or any task requiring models with similar performance on different input data (e.g. fairness).
> We admit that we still exploring the possible applications, but believe the broader ML community would be better suited for this purpose.
> Hence, instead choosing an artificial use-case, we would like to share these findings with the community.
>
> > In fact, the only conclusive result they have on these is the known balance property, but this only comes from a function symmetry and is not data-related.
>
> Regarding conserved quantities derived from symmetries, we have two main results.
> First, is a general derivation of conserved $Q$, which includes the imbalance you mention. But more importantly, we find that this approach __does not yield conserved $Q$ for rotational symmetry__.
> To our knowledge, for rotation symmetry, our result regarding cancellation of angular momentum between layers is novel.
> We believe this to be of equal importance to the imbalance properties.
> It also highlights the limitation of previous approaches,  which missed this failure case.
>
> Additionally, our derivation of the dimensions of flat minima, and showing conserved $Q$ can parametrize them are other concrete results of this work.
>
> > Do the authors have in mind an example where some data invariance is of paramount importance? (I have a more detailed question about this in the next box).
> > In fact, going further, the different paragraphs of Subsection 1.5 are really fuzzy and there is a lack of details in the calculations. Propositions and facts are inline without emphasis between them so that it is very diffiicult to follow. Furthermore, it appeared to me (maybe I am wrong), that some of the derivations are very fast and this prevents for a proper easy reading of this very technical section.
>
>
> Thank you for raising this issue. We have completely rewritten 1.5 (now 1.4) with a step-by-step derivation, starting from why data-dependnece is key.
> We begin from parameter transformations that can make a model perform well on labeled out of distribution (OOD) data.
> We show that these transformations are precisely an example of general equivariance.
> So rather than defining general equivariance first, we now derive it as a solution to this OOD setting and then discuss its more abstract mathematical properties.
>
> > Examples of these are e.g.:
> (i) second line of the __General equivariance__ paragraph $c(g_2,g_1Z)c(g_1,Z) = c(g_2g_1, Z) $, what is a typical $c$? What does it mean "with physical interpretation"? why is this property called "equivariance"? This paragraph lacks of lot of details (both in the interpretation and in calculation) to be followed.
>
>
> The main point of __general equivariance__  is that it generalizes $\sigma(gZ) = \pi(g)\sigma(Z)$ to the _data-dependent_ transformation $\sigma(gZ) = c(g,Z)\sigma(Z)$.
> Applying two $g_1,g_2$, we find two equations: 1) considering $g=g_2g_1$ as one transformation we have
> $$ \sigma(g_2g_1Z) = c(g_2g_1,Z) \sigma(Z)$$
> 2) breaking $g_2g_1$ into first acting with $g_2$ on $g_1Z$, we have
> $$ \sigma(g_2g_1Z) =  c(g_2,g_1Z)\sigma(g_1Z) = c(g_2,g_1Z)c(g_1,Z)\sigma(Z) $$
> Hence, a consequence of the equivariance $\sigma(gZ)= c(g,Z)\sigma(Z)$ is that $c(g_2g_1,Z)= c(g_2,g_1Z)c(g_1,Z)$.
> We now have a step-by-step derivation of these properties using the OOD example.
>
> > what is a good $c$?
>
> An example is eq. 13, our data-dependent transformation $c(g,Z)=\sigma(Z)\sigma(gZ)^{-1}$.
> Also the simple data-independent equivariance $c(g,Z)=\pi(g)$ is a special case of this.
>
>
> However, we note that this choice of $c$ may not work for arbitrary $g\in GL_h$ as it relies on $\sigma(gZ)$ being invertible.
> For many $\sigma$ it may work for a much small subgroup of $GL_h$.
> When $\sigma(Z)$ is invertible, this $c(g,Z)$ trivially works for exact  symmetries of $\sigma$, since then we have $\sigma(gZ)^{-1} = [\pi(g)\sigma(Z)]^{-1} = \sigma(Z)^{-1} \pi(g)^{-1}$, and so $c(g,Z) = \pi(g)^{-1}$.
> Our current results show that equivariance to full $GL_h$ may only work for batches $Z$ containing 1 datapoint, and the explicit form of $c$ may be very different from the $c(g,Z)$ above.
> We elaborate on this in the new version.

---

> > ### Author Response · Authors · 2022-11-15
> > **Part 2**
> >
> >
> > > (ii) Second line of Using nonlinear transformed [...] paragraph: do really the authors want to write:
> > $g \cdot (U,V,X) = (\tilde{U}(g,VX),gV, X) $, and in this case, what is $\tilde{U}$?
> >
> > We have rewritten that paragraph. We apologize for the confusion. Here $\tilde{U} = U c(g, g^{-1}VX) $.
> >
> >
> > >In this paragraph, it is concluded that at permutation + small deformation of the data set the function should be almost the same: why should we care about this perturbative results? I have in mind that the assumption on the data for generalisation is more that they come iid from the same distribution not that they are close to each other in distance?
> >
> > The closeness of data is more subtle and we are adding explanation about this.
> > Note that $U\sigma(VX)$ does not need to mean a two layer network; it may indicate upper layers in a deep network with $U=W_{i+1}$, $V= W_i$ and $X$ being a latent embedding.
> > When the network is trained, $X$ and $X'$ which are close in the latent space share important features.
> > When the latent embedding has a high quality, closeness maybe related to distribution similarity (e.g. samples from the same class will be close).
> >
> >
> > > (iii) The last paragraph of this section on __adapting models__ is very hard to follow, notations are not properly introduced, e.g. in $Z'=gZP$, what is $g$?
> >
> > Sorry for that. We hope sec. 1.4 in the new version is clearer (posted by Tuesday night). Here $g\in GL_h$ is atransformation relating original data $Z=VX$ to new data $Z'=VX'$.
> >
> >
> > > As far as I understood, the authors are not the first to consider symmetries to explain invariants of the dynamics. Comparison with results of previous works would be welcome to understand how it articulates with theses previous works.
> >
> > We have added clarifications (in blue) in various parts of the paper highlighting the diifference between our work and exisiting work.
> > For instance, Neural mechanics https://arxiv.org/pdf/2012.04728.pdf focused on one-parameter groups, while our derivation is for general groups.
> > Besides, our results regarding the failure of this method for rotations and finding the conservation law for rotations (end of sec. 2) is novel, to our knowledge.
> >
> >
> > > I am sure that all the propositions of Section 1 are well-known facts for example.
> >
> > We have now shortened sec. 1 and go straight to equivariance, which is the novel angle that we believe was not discussed in prior work to derive the symmetries in neural networks.
> >
> >
> > > Minor issues: Eq (6), the gradient is seen as a linear form in this article it seems because operations with matrices are left operations. It can be troubling so it needs to be said.
> >
> > Correct. The paragraph before eq. (6) derives the group action on the gradient.  We hope that suffices. Appendix F (Now C.1) derives the gradient and its group action in more detail. We also considered defining a left action with $g^T$ and eventually chose the current form.
> >
> > > Typo page 6, paragraph 3: $\sigma(a)-\sigma(b)$.
> > > Paragraph need for data-dependant symmetries, isnt this
> > $\mathcal{L}(\theta',X') = \mathcal{L}(\theta,X') $ (and not $\mathcal{L}(\theta,X) $ ).
> > > After proposition 1.6,  should be . After formula (10), the infinitesimal form of  ... Like previously the formula (11) is weird as  is a vector and the  usually indicate a scalar product. As far as I understand, the authors may come from a theoretical physics community where such notations may be usual.
> >
> > We appreciate the careful reading. We have fixed them.
> > For the infinitesimal action, we have the version with explicit indices right below the equation. We can replace the brackets with the indexed version.
> >
> >
> > __Mirror Flow:__
> > We are working on your question regarding mirror flows and will let you know if we have any insights. In the mean time, can you please elaborate on what nice geometric properties $\nabla\psi$ has which could relate it to conserved quantities? Thank you.

---

> > > ### Comment · Reviewer_M5tH · 2022-11-15
> > > **After rebuttal**
> > >
> > > Thank you very much for the detailed answers, I will raise my score by one. I hope that the authors will properly correct the many typos and/or imprecisions of their first exposal.
> > > For the mirror: I have in mind not a *conserved* quantity but a invariant space. At the end of the day, consider even, the least square problem with data matrix $X$, then even when the problem is overparametrised, we can show that at any time $t>0$, the iterates belong to the affine space given by initialisation $+ span({X^\top})$. This is not a conserved quantity but still a nice invariance! Can the authors comment on this ?

---

> > > > ### Author Response · Authors · 2022-11-19
> > > > **Mirror descent conservation law from symmetry**
> > > >
> > > > We have some ideas about your question, though we haven't fully solved it. We think $\nabla \psi \in \mathrm{Span}(X)$ may be expressible as a conservation law.
> > > > The associated symmetry may be linear combinations of samples which keep both the loss as well as $\mathrm{Span}(X)$ invariant. The subtlety is that here the symmetry acts on __data__ $X,Y$, whereas in our paper we only consider parameter space symmetries. Thus, by slightly generalizing our method and allowing symmetries acting on data, we think a procedure similar to sec. 2 can be used for this problem.
> > > >
> > > > We want to relate conservation laws $dQ/dt = \nabla_w Q \dot{w} = -\nabla_w Q \epsilon \nabla \mathcal{L}=0$ to invariance of loss under symmetry $ M \cdot \mathcal{L} =0$.
> > > >
> > > > To verify our understanding of the question, we will first formalize the setup and derive the invariant in the least square problem. We will also briefly discuss how the approach can be extended to mirror flow. We then provide some speculations of how this geometric invariant may be related to symmetries.
> > > >
> > > > **1. Setup**
> > > >
> > > > Denote the data matrix as $X \in \mathbb{R}^{n \times d}$, parameters as $w \in \mathbb{R}^d$, and labels as $y \in \mathbb{R}^n$.
> > > > The least squares loss is
> > > > $$ L(w) = \frac{1}{2}||y - Xw||_2^2. $$
> > > > The gradient is
> > > > $$ \nabla L(w) = X^T (y - Xw). $$
> > > >
> > > > Therefore, $\nabla L(w)$ is always in span$(X)$, the space spanned by the columns of $X$. In the gradient flow
> > > > $ \frac{dw}{dt} = -\nabla L(w)$,
> > > > the difference between parameters at time $t$ and at initialization is $w(t) - w(0) = -\int_0^t \nabla L(w(t)) dt$, which is also in $\text{span}(X)$.
> > > >
> > > > The geometric invariant here is $\Delta w = w(t) - w(0) \in \text{span}(X), \forall t$. Equivalently, $\Delta w - P(\Delta w) = 0$, where $P$ is an operator that projects $\Delta w$ to  $\text{span}(X)$.
> > > >
> > > > The geometric invariant in mirror flow of a two-layer diagonal network is obtained similarly. The mirror flow is
> > > > $$ \frac{d}{dt} \nabla \psi (\beta(t)) = -\nabla \mathcal{L}(\beta(t)) $$
> > > > where $\nabla \psi: \mathbb{R}^d \xrightarrow{} \mathbb{R}^d$ is the mirror map and $\beta$ are parameters. It can then be shown that $ \nabla \psi (\beta(t))$ is in the span of $X$ (first paragraph in Section 4.1 in https://arxiv.org/pdf/2106.09524.pdf). The proof involves integraing the right hand side of the mirror flow and show that it is in the form of $X^T b$, where $b$ is a vector.
> > > >
> > > >
> > > > **2. Symmetry**
> > > >
> > > > The loss function has an $SO(n)$ symmetry. For $g \in SO(n)$, the following group action leaves the loss unchanged:
> > > > $$g \cdot (X, y, w) = (gX, gy, w).$$
> > > > It is easy to verify that the unit and multiplication axioms are satisfied.
> > > > Additionally, this group action preserves the span of $X$. If a vector $v$ is in span$(X)$, then $v$ is also in span$(gX)$. The converse holds similarly.
> > > > For a proof sketch, if $v$ is in span$(X)$, then there exists a vector $b$ such that $X^T b = v$. For $g \in SO(n)$, $v = X^T g^T g b = (gX)^T (gb)$, so $v$ is also in span$(gX)$.
> > > >
> > > > Note that the symmetry considered here acts on data instead of parameters, which differs from our paper.
> > > >
> > > > **3. Conserved quantity**
> > > >
> > > > For small transformations, $g = I + \theta M$, where $\theta$ is a small number and $M$ is in the Lie algebra of $SO(n)$. The rate of change of $L$ in the symmetry direction is:
> > > > $$ \frac{dL}{d\theta} = \frac{\partial L}{\partial X} \cdot \frac{dX}{d\theta} + \frac{\partial L}{\partial y} \cdot \frac{dy}{d\theta}$$
> > > > where $\cdot$ denotes the inner product calculated by contracting all indices. Writing out the indices,
> > > > $$
> > > > \frac{dL}{d\theta} = -\frac{\partial L}{\partial X_i^a} \cdot \frac{dX_i^a}{d\theta}
> > > > +\frac{\partial L}{\partial y_i} \cdot \frac{dy_i}{d\theta}
> > > > $$
> > > > $$
> > > > =-w_a (y^{Ti} - w^{^Tc} X_c^{Ti}) M_i^j X_j^a  + (w^{^Tc} X_c^{Ti}) M_i^j y_j $$
> > > > $$=Tr[- (y - Xw)^T MXw + (y - Xw)^T My]$$
> > > > $$=Tr[(y - Xw)^T M (y - Xw)]
> > > > $$
> > > >
> > > > By definition, the change of $L$ in the symmetry direction is 0. The equation above confirms this: since $M$ is anti-symmetric, $\frac{dL}{d\theta}= 0$ is always satisfied.
> > > >
> > > > We want to see if the conserved quantity is related to a symmetry. In other words, we want to derive $\Delta w - P(\Delta w) = 0$ from $\frac{dL}{d\theta} = Tr[(y - Xw)^T M (y - Xw)] = 0$.
> > > >
> > > > The gradient of loss with respect to $w$ is
> > > > $$\delta w =- \frac{\partial L}{\partial w} = -X^T (y - Xw)$$
> > > >
> > > > If we rewrite the conservation law $dL/d\theta=0$, which is now in terms of $\partial L/\partial X$,
> > > > in terms of $\partial L/\partial w$, then we get a condition between $\delta \psi \sim \Delta w $ and $X$.
> > > > When the number of samples $n$ in $X$ is less than features $d$, We can use a right pseudo-inverse $XX^+=I_n$ to write
> > > >
> > > > $$\frac{dL}{d\theta}= Tr[\delta w^T X^+ M (y-XW)]=0$$
> > > >
> > > > Recall that for parallel vectors $v'=cv$, we have $ v^TMv'= cv^T Mv=0$.
> > > > We suspect that $\nabla \psi \sim \delta w \in \mathrm{Span}(X)$ is analogous to parallel vectors $v^T M v'=0$, and some expression like $\delta w M X$ must vanish, but we don't have a proof yet.

---

> > > > > ### Comment · Reviewer_M5tH · 2022-11-22
> > > > > **Detailed answerr**
> > > > >
> > > > > Thank you for this. This was simply food for further thoughts and I appreciate that the authors tried to dig into this!

---

### Official Review · Reviewer_FC6G · 2022-10-24

**Confidence:** 2
**Correctness:** 4
**Technical Novelty And Significance:** 3
**Empirical Novelty And Significance:** 3
**Recommendation:** 8

**Clarity, Quality, Novelty And Reproducibility:**

**Clarity**

The work is clear and well written, although there are some typos and broken links.

p.5 “LeakyReLU This is a special case of homoegeneous activation”, misspelling of homogeneous.

Appendix:
p. 37 broken link in last paragraph.  Equation overfull into margin

p. 38  Broken link at the end of Proposition I.2 and in the statement of Proposition I.2

P. 39 in Appendix, broken link in statement of Lemma I.5

**Quality**

I believe the work is of high quality as it offers a useful formalism and an extensive tutorial in the appendix.

**Reproducibility**

Since this work is primarily theoretical and provides proofs of its theorems I don't think reproducibility is applicable here.

**Questions/Comments**

p.5 “Suppose the activation $\sigma: \mathbb{R}^h \rightarrow \mathbb{R}^h$ is homogeneous, so that $\sigma$ is applied pointwise in the standard basis. Assume also that there exists $\alpha > 0$ such that $\sigma(c z) = c^\alpha \sigma(z)$ for all $c \in \mathbb{R}_{>0}$ and $z \in \mathbb{R}^h$.”  I’m confused.  Isn’t the second condition simply stating the activation is $\alpha$ positive homogeneous.  So what additional condition are you trying to specify with the first sentence?





**Strength And Weaknesses:**

**Strengths**

The paper introduces a general framework to relate symmetries of parameter space to conserved quantities of gradient flow, which is something I was hoping to see.  They make extensive effort in the appendix to catalogue applications of their theory to common activation functions and give an extended tutorial of their theory.

**Weaknesses**

I think the data dependent symmetries are the most interesting component, and the work could be enhanced by more experiments or discussion that give insight into what these data dependent symmetries look like in some settings.

**Summary Of The Paper:**

This paper studies the parameter space symmetries induced by equivariances in the activation function as well as data dependent nonlinear symmetries.  Furthermore they formalize how these symmetries lead to conserved quantities during gradient flow.

**Summary Of The Review:**

**Summary**

This paper offers a useful formalization of how equivariances lead to parameter space symmetries which in turn lead to conserved quantities of gradient flow.  They offer an extensive tutorial in the appendix.  I believe the theoretical community will appreciate this work.

---

> ### Author Response · Authors · 2022-11-15
> **Clarifications**
>
> Thank you for your comments and positive feedback! We have fixed the typos and broken links in the updated version.
>
> > the work could be enhanced by more experiments or discussion that give insight into what these data dependent symmetries look like in some settings.
>
> On the theoretical side, we have provided the explicit expression for the nonlinear group actions that are data dependent. We are __significantly rewriting__  section 1.5 (sec. 1.4 in the next version) on nonlinear action to provide additional intuition of where the data dependent symmetries come from. On the application side, we have several examples of how these symmetries affect the trained model and learning dynamics.
> For example, we ran experiments that use data-dependent symmetries to transform trained models on CIFAR-10, and show that the transformations leave the loss approximately invariant.
> We welcome additional suggestions on better explaining data dependent symmetries in specific settings.
>
>
> > p.5 ... Isn’t the second condition simply stating the activation is $\alpha$ positive homogeneous. So what additional condition are you trying to specify with the first sentence?
>
> We have fixed the explanation of a homogeneous activation function by rewording those two sentences into the following: "Suppose the activation $\sigma : \mathbb{R}^h \to \mathbb{R}^h$ is  _homogeneous_, meaning that (1) $\sigma$ is applied pointwise in the standard basis and (2) there exists $\alpha>0$ such that  $\sigma(c z) = c^\alpha \sigma(z)$ for all $c\in \mathbb{R}_{>0}$ and $z \in \mathbb{R}^h$."

---

> > ### Comment · Reviewer_FC6G · 2022-11-17
> > **Response to authors**
> >
> > I thank the authors for their response and the clarifications.  I have no further questions at this time.

---

### Official Review · Reviewer_z99y · 2022-11-04

**Confidence:** 2
**Correctness:** 4
**Technical Novelty And Significance:** 3
**Empirical Novelty And Significance:** 2
**Recommendation:** 6

**Clarity, Quality, Novelty And Reproducibility:**

As the authors describe, there are many works characterizing the loss landscape of neural networks but relatively little on the origin of low-loss valleys. The need for data-dependent symmetries is well-motivated and the authors build upon previous work relating conserved qualities to symmetries.

The writing is clear and quality of the analysis appear sound.

Minor:
- Figure 1 caption typo "minum"
- Figure 2 does not appear to be referenced in the text.


**Strength And Weaknesses:**

### Strengths
- The characterization of symmetries is a novel perspective and sheds some light on a deep learning phenomena of interest (e.g. ensembles formed along flat minima have been shown to generalize better). The paper describes detailed analysis wider class of data-dependent symmetries and how they relate to conserved qualities in gradient flow.
- The empirical results are interesting and support the theory. The conserved quality Q affects the convergence rate and distribution of the eigenvalues of the Hessian. The group action allows for an ensemble without retraining, which has potential in improving robustness to adversarial attacks.
- Related work is discussed well throughout the paper.

### Weaknesses
- It would support the theory to run experiments on a non-toy dataset. Currently, they are done with small randomly generated matrices. If the same conclusions can hold for MNIST/Cifar, these results would be more relevant for practitioners. I believe experiments relating symmetries to OOD data would also strengthen the discussion in section 1.5.
- The paper is long (56 pages with appendix) and has many different examples/derivations. I'm not sure a conference format such as ICLR is the correct venue for such a submission.


**Summary Of The Paper:**

This paper seeks to better understand theoretically symmetries in the loss landscape of neural networks. It first uses group theory to formulate a more general notion of parameter space symmetries, beyond permutations. The authors then motivate and present a class of nonlinear, data-dependent symmetries and a procedure for deriving conserved qualities associated with the symmetries. They show that these conserved qualities can be used to parametrize symmetric flat directions and show numerical results on two layer neural networks.

**Summary Of The Review:**

The paper offers an interesting perspective on understanding neural network loss landscapes through ideas from group theory and physics. It would be strengthened if the empirical results could be reproduced beyond a toy dataset.

context for review: I'm familiar with/written empirical deep learning papers but less capable of assessing the significance of the theoretical contributions in this work. I skimmed the appendix/some related works in writing this review.

---

> ### Author Response · Authors · 2022-11-15
> **Shortening derivations and comment about experiments**
>
> Thank you for your comments and insightful perspectives!
>
> > It would support the theory to run experiments on a non-toy dataset. Currently, they are done with small randomly generated matrices. If the same conclusions can hold for MNIST/Cifar, these results would be more relevant for practitioners.
>
> We agree that empirical results on non-toy datasets would be more convincing to practitioners.
> For the ensemble experiments, we used models trained on CIFAR-10 and showed that our symmetries can be used to improve robustness to adversarial attacks (setup described in Appendix O in old version, Appendix L in updated paper)).
> Since our main contribution is theoretical, the rest of the experiments serve to confirm and extend theoretical examples. Using artificial datasets gives us better control of the problem settings and help us demonstrate the theories more clearly.
>
>
> > The paper is long (56 pages with appendix) and has many different examples/derivations. I'm not sure a conference format such as ICLR is the correct venue for such a submission.
>
> We hope that the main text (9 pages) includes all main results, while being self-contained and accessible to a broad audience. The appendix, which contains technical proofs and many examples, are only intended for interested readers.
> As an effort to further improve readability, we have shortened and reorganized some of the technical, mathematical sections in the main text to make the results more accessible. We plan to do so for the appendices as well.

---

> > ### Comment · Reviewer_z99y · 2022-11-30
> > **Response**
> >
> > Thank you for the response! I have read through the other discussions and appreciate the improvements to clarity and presentation, but still have some concerns about the applicability of the results to modern neural networks. I will maintain my score.

---

### Official Review · Reviewer_J3kj · 2022-11-04

**Confidence:** 2
**Correctness:** 3
**Technical Novelty And Significance:** 3
**Empirical Novelty And Significance:** 2
**Recommendation:** 6

**Clarity, Quality, Novelty And Reproducibility:**

Due to a lack of clarity, it is hard to assess the quality and the novelty of the work.


**Strength And Weaknesses:**


I had a very hard time reading the paper. I find the organization
of the paper quite confusing. A lot of mathematical formalism
is squeezed in every page while giving away very little in terms of
explanations and intuions. Much of the paper revists classical concepts from group theory and their physical implications through Noether's theorem,
with  novel contributions seemingly contained in Section 1.5
and possibly Section 2. It is hard to asses the relevance of such contributions.

The equivariance property they propose in eq. 12 and 13 seems useful
to produce a new set of parameters with approximately the same loss as the original ones
when observing a new batch of data that is close (up to a permutation) to the original data. In practice though I'm not sure how useful this is, and how "close" the data should be, since it has not been experimentally tested in the manuscript.

The analysis in Section 2 of the conserved quantities along the dynamics is also interesting, but it's not clear how much of it is original and how much is relevant. The experiments are not properly explained in the main text.

It could be the case that I didn't understand the main points of the paper,
which in any case calls for a much better rewrite since I can easily read through
similar papers such as e.g. "Noether’s Learning Dynamics" arxiv:2105.02716.


**Summary Of The Paper:**

The paper investigates the role of symmetries in multi-layer perceptrons,
putting forward the idea of data-dependent symmetries.
Conservation laws stemming from Noether's theorem are discussed.


**Summary Of The Review:**

The paper should be mostly rewritten and provide a more clear focus on the results and their implications.

---

> ### Author Response · Authors · 2022-11-15
> **improving flow and contrast with existing work**
>
> Thank you for your comments! We have significantly modified the writing to improve organization and to highlight our contributions. We provide details below on how we addressed specific comments in the new version (will be posted Tuesday end of day).
>
> > I find the organization of the paper quite confusing. ... It is hard to asses the relevance of such contributions.
>
> We are reorganizing much of the paper in order to provide a clearer focus on the results (edits in blue).
> We have made the contributions clearer now to contrast with existing work.
> Ex: while deriving conserved quantities from symmetries was done before, it was done for one-parameter groups. We extended that to general continuous groups and also __identified a major limitation__ of that approach: it __fails to find $Q$ for rotation symmetries__.
>
> Our main contribution is the symmetry-based insights on the loss landscape.
> The main novel theoretical results are (1) the computation of the dimensions of generic orbits (and hence flat local minima) in the case of linear networks, networks with homogeneous activations such as Step-ReLU, and networks with radial rescaling activations; (2) a non-linear group action that is data-dependent; (3) a version of conservation of angular momentum for networks with radial rescaling activations.
> We also discuss potential applications related to optimization, generalization, and robustness.
>
>
> > The equivariance property they propose in eq. 12 and 13 seems useful to produce a new set of parameters with approximately the same loss as the original ones when observing a new batch of data that is close (up to a permutation) to the original data. In practice though I'm not sure how useful this is, and how "close" the data should be, since it has not been experimentally tested in the manuscript.
>
> Empirically, we verify on CIFAR-10 that the model performance is approximately the same after transforming the parameters using our group actions (Section 3, "Ensemble models"). More details on the setup can be found in Appendix L. We provide theoretically analysis on how "close" the data should be in Appendix E.
>
>
> > The analysis in Section 2 of the conserved quantities along the dynamics is also interesting, but it's not clear how much of it is original and how much is relevant.
>
> Two new results in Section 2 are conservation of angular momentum and discussion of using conserved quantities as coordinates for flat minima. We have modified the text to clarify novelty.
>
> > The experiments are not properly explained in the main text.
>
> Due to space constraints and our focus on theoretical contributions, the application section in the main text is meant to be a short summary of potential use cases.
> We do have an extensive discussion of empirical observations in Appendix H, I, J, K, and L.

---

> > ### Comment · Reviewer_J3kj · 2022-11-30
> > **score updated**
> >
> > Thanks, for the reply. Since the revised paper is definitely more readable and the results are now clearer I increased by 1 my score.

---

### Author Response · Authors · 2022-11-15
**Updates to the paper**


We thank the reviewers for very helpful and insightful comments.
We are encouraged that they find our data-dependent symmetry novel, interesting, and well-motivated.
We have added individual response for each reviewer and updated the paper.
We are glad that reviewers z99y, FC6G, and M5tH find our writing clear and M5tH states "despite the technicality of the theory of invariance, the paper is pretty clear and easy to follow... there is a real care about pedagogy".
Nevertheless, we regret that J3kj found it confusing and we are __making extensive edits__ (blue text) to clarify certain concepts, improve organization, highlight novelty, and further improve readability. We have __significantly reduced the clutter__ by moving known propositions and many proofs to the appendix.

We have updated the contributions to contrast this work with other works relating symmetries and conserved quantities.
__contributions:__

1. A general framework based on __equivariance__ for finding symmetries in NN loss landscapes.
2. A derivation of the __dimensions of flat minima__ induced by symmetries.
3. A new class of __nonlinear, data-dependent symmetries__ of NN parameter spaces.
4. Expanding prior work on __deriving conserved quantities__ (CQ) associated with symmetries and discussing its failure for rotation symmetries.
5. Finding a __cancellation of angular momenta__ between layers for rotation symmetries.
6. A __parameterization__ of symmetry-induced flat minima via  the associated CQ.

We are finishing the edits and will post a revised version of the paper late Tuesay,  Nov 15.

---

> ### Author Response · Authors · 2022-11-16
> **Updated pdf uploaded**
>
> Please find the new version of the paper, with significant changes highlighted in blue. We will continue to iterate over and polish the text. Some of the proofs are still being written in the appendix, but the main text should be self-contained.
> Important change log:
> 1. We highlight the contrast between our work and existing work in various parts.
> 1. Nonlinear action (now sec. 2) has been completely rewritten.
> 2. We have added more intuitive steps motivating the need for data-dependent action.
> 3. We provide an algorithm for constructing equivariant maps used in the nonlinear action.
> 4. The Lipshitz bounds on the effect of nonlinear action are in a separate theorem, with explanation about how this implies limits on the effect of group action on latent embedding of data.

---

### Decision · Program_Chairs · 2023-01-20

**Decision:**

Accept: poster

**Justification For Why Not Higher Score:**

As one of the reviewers pointed out, the paper would be strengthened if the empirical results could be reproduced beyond a toy dataset.

While some reviewers find the paper clear, there are reviewers who have raised clarity issues and suggest rewriting. Authors did make an effort to significantly change the papers adopting reviewers requests.


**Justification For Why Not Lower Score:**

This paper offers a useful formalization of how equivariances lead to parameter space symmetries and then to conserved quantities of gradient flow. The paper offers an extensive tutorial in the appendix. Theoretical community at ICLR  would appreciate this work.

Data-dependent symmetry is a well motivated and interesting concept introduced by the authors.


**Metareview: Summary, Strengths And Weaknesses:**

The paper analyzes parameter space symmetries in neural networks induced by equivariance in the activation function + data dependent non-linear symmetries. The paper goes beyond conventional permutation symmetries. Authors formalize how these symmetries lead to conserved quantities during gradient flow.  With these conserved quantities, authors could parametrize symmetric flat directions and show numerical results on two layer neural networks.

Strength
- Introduces general framework to related symmetries of parameter spaces to conserved quantities of gradient flow
- High quality and offer useful formalism and extensive tutorial
- Characterization of symmetry is novel perspective and shed light on deep learning phenomena of interest
- Empirical results are interesting and supports theoretical claim: authors show conserved quantity affects convergence rate and distribution of eigenvalues of Hessian
- Authors also show group action allows for an ensemble without retraining which could be useful for robustness

Weakness
- While data dependent symmetry is the interesting component, experiments and discussion on it is not there to fully explain the significance
- Would be better to have a non-toy dataset beyond randomly generated matrices. Similar analysis on MNIST / CIFAR-10 tasks would shed much more light and practical impact for practitioners.


**Note From Pc:**

if the above contains the word "oral" or "spotlight" please see: "oral" presentation means -> notable-top-5% and "spotlight" means -> notable-top-25%. As stated in our emails, we are disassociating presentation type from AC recommendations